health and disease and epidemiology/theoretical biology

livestock, spread of disease, cattle shipment, movement network, expert data

**Author for correspondence:**
Tom Lindström
e-mail: tom.lindstrom@liu.se

# Assessing intrastate shipments from interstate data and expert opinion

Peter Brommesson[1], Stefan Sellman[1],
Lindsay Beck-Johnson[2], Clayton Hallman[2],
Deedra Murrieta[2], Colleen T. Webb[2], Ryan S. Miller[3],
Katie Portacci[3] and Tom Lindström[1]

[1]Department of Physics, Chemistry and Biology, Division of Theoretical Biology, Linköping University, 58183 Linköping, Sweden
[2]Department of Biology, Colorado State University, Fort Collins, CO 80523, USA
[3]Center for Epidemiology and Animal Health, United States Department of Agriculture-Veterinary Services, Fort Collins, CO 80526, USA

PB, 0000-0003-4941-1313; LB-J, 0000-0001-7247-0309;
TL, 0000-0001-7856-2925

Live animal shipments are a potential route for transmitting animal diseases between holdings and are crucial when modelling spread of infectious diseases. Yet, complete contact networks are not available in all countries, including the USA. Here, we considered a 10% sample of Interstate Certificate of Veterinary Inspections from 1 year (2009). We focused on distance dependence in contacts and investigated how different functional forms affect estimates of unobserved intrastate shipments. To further enhance our predictions, we included responses from an expert elicitation survey about the proportion of shipments moving intrastate. We used hierarchical Bayesian modelling to estimate parameters describing the kernel and effects of expert data. We considered three functional forms of spatial kernels and the inclusion or exclusion of expert data. The resulting six models were ranked by widely applicable information criterion (WAIC) and deviance information criterion (DIC) and evaluated through within- and out-of-sample validation. We showed that predictions of intrastate shipments were mildly influenced by the functional form of the spatial kernel but kernel shapes that permitted a fat tail at large distances while maintaining a plateau-shaped behaviour at short distances better were preferred. Furthermore, our study showed that expert data may not guarantee enhanced predictions when expert estimates are disparate.

# 1. Introduction

Transboundary animal diseases (TADs) pose a global threat to food security, and outbreak events are a major concern for animal health. Outbreaks may cause national emergencies, with huge costs to the livestock sector due to disruption of production and export. As such, TADs are a primary concern for food security [1]. Additionally, several TADs have painful symptoms, making them an animal welfare concern. In the face of an outbreak, policymakers must often make high-stakes decisions with limited information. Therefore, outbreak preparedness is important for facilitating a swift and efficient response, which is essential to disease control [2]. A detailed understanding of the potential contacts between premises that could mediate transmission in a potential outbreak is an important aspect of outbreak preparedness.

Animal shipments between farms and other agricultural premises are of particular concern for disease spread because of their potential to introduce infected animals into susceptible herds or flocks [3]. Contacts often occur over large distances and can precipitate geographically widespread epidemics [4]. For example, nine of the twelve spatial clusters of the 2001 foot and mouth disease (FMD) outbreak in the UK were initiated by live animal shipments [5]. Early detection of shipment contacts may promote a reduced outbreak.

Cattle production is an important part of the US meat animal industry accounting for US$67.3 billion in total production value year 2017 [6] and is second only to poultry in the total pounds of product produced [7]. The US cattle industry accounts for approximately 900 000 premises and 103 million animals [8,9]. An outbreak of a TAD such as FMD is expected to have severe impact on the US economy, with economic losses predicted to be at least US$14 billion in the first year of an outbreak as a result of control and disruption to international trade. This number corresponds to 9.5% of the USA farm income and for the live cattle and beef meat sector the losses in gross revenues were estimated at 17% and 20%, respectively [10]. Economic losses due to spread of disease among cattle are not unique to TADs. Attempts to control bovine tuberculosis (bTB), which is endemic in Michigan, cost $200 million over 15 years [11].

Shipment restrictions that minimize the risk of pathogen spread without interrupting production are essential for cost-efficient disease control. However, modelling efforts to investigate the efficiency of control options are challenged by frequently limited data on locations, sizes and types of premises, in particular in Africa and Southeast Asia [12]. By contrast, countries of the European Union are legally required to collect and store data of all live animal shipments [13]. In the USA, premises-level data describing location, premises type and animal inventory is not uniformly collected for domestic animal industries due to stakeholder concerns regarding cost, confidentiality and security of collected information [14]. Survey data have previously been used to identify interstate shipping patterns using Interstate Certificates of Veterinary Inspection (ICVI) issued by animal health authorities when animals are shipped across state boundaries [15]. However, shipping patterns in countries with complete data typically show high frequency of short-distance shipments [16], and a similar pattern in the USA would result in a large number of intrastate shipments. As such, there is a need for methods to extrapolate from existing data to predict complete shipment patterns. Lindström *et al.* [17] proposed an approach to address this need using a kernel function to model distance dependence. These Bayesian predictions have been used for FMD outbreak modelling [18] and making recommendations for bTB surveillance [4]. However, there is a need for more detailed methods, particularly regarding the potential sensitivity of within-state shipments to the choice of kernel function. Therefore, there is a need for investigating different kernels to understand their impact on the predictions.

Empirical data are not the only source of information to inform models. Other sources, such as the knowledge of experts, can be used instead when developing and using models, e.g. for estimating presence and risk of infection of animal diseases [19,20]. Expert data can also facilitate prior elicitation for sought parameters [21] and have been used in addition to empirical data, for instance in ecology [22–24]. Furthermore, expert data can be used to inform within-state shipments in the USA. Here, state veterinarians, cooperative extension professors and other experts with extensive knowledge about the US cattle industry may offer important insight. This extra information, can be used together with empirical data to provide better understanding of cattle shipment patterns.

The Bayesian paradigm is ideally suited to incorporate expert opinion when these can be expressed as prior distributions. However, there are several issues when converting the answers of expert surveys into priors. Answers might not be expressed explicitly for the parameters we wish to elicit priors for, particularly when the survey is not tailored specifically for the statistical model. Expert data may also be missing if there is a lack of expertise in some geographical areas or if some targeted experts choose not to participate. Furthermore, expert answers are typically provided as point estimates rather than a range. Instead of using expert data to inform priors, we propose a statistical model whereby expert answers are treated as data in a hierarchical Bayesian framework.

Based on ICVI and expert data, we developed a modelling framework for continental-scale cattle shipments and investigated three functional forms of the spatial kernel, taking into account that the livestock production system varies across the USA. For instance, feedlots are primarily located in the central states, whereas dairy production is most intense in coastal states such as California and New York and in the Midwest [9]. We therefore propose a model, denoted the United States animal movement model (USAMM), that accounts for the heterogeneity of the system. Our aims are to (i) estimate the proportion of intrastate shipments for each state, (ii) clarify how sensitive this estimate is to the choice of kernel function, (iii) improve estimates of shipment rates across the USA at state and county levels, and (iv) investigate the value of including expert data in the analysis.

# 2. Material and methods

## 2.1. Overview

We modelled beef and dairy shipments as two separate networks consisting of all counties of the contiguous USA (i.e. excluding Hawaii and Alaska) as nodes and shipments between them as links. Models were parametrized from shipment data extracted from Interstate Certificates of Veterinary Inspection (ICVIs) and county information, such as county centroid coordinates and number of animals in the county. Furthermore, we incorporated expert estimates of the proportion of interstate shipments to enhance our model.

We used Bayesian modelling to estimate parameters describing the underlying processes that produce the observed networks. We also performed model selection to investigate which candidate model better fits the data and implemented a series of model validations to determine predictive accuracy. Finally, we conducted sensitivity analysis to investigate the robustness of our model, including investigation of biases that could have been imposed by the presence of temporal variations. The mathematical notation used and its interpretation, can be found in table S1 in the electronic supplementary material.

## 2.2. Data

The shipment data used in this study consists of ICVI records, which are official documents issued by an accredited veterinarian or an official state or federal veterinarian. The primary purpose of ICVIs is to prevent potential disease spread by ensuring that shipped animals are apparently healthy and show no visible signs of communicable disease. ICVIs are also used as one source of information to support traceability of animals in the event of a disease outbreak [25]. ICVIs are required for most interstate shipments, except for shipments going directly to slaughter, and contain information that includes origin, destination, date of the shipment and characteristics of the shipped animals.

We used the ICVI dataset from 2009, which is described in detail in Buhnerkempe et al. [15]. It consists of systematic 10% samples of ICVIs from all contiguous US states except New Jersey (did not participate), and was provided by the origin states. ICVIs were classified as beef or dairy by a classification tree analysis [15], and the data contained after curation 15 725 and 2814 beef and dairy shipments, respectively. The number of premises per county were estimated as the mean number of premises from 10 realizations of the cattle version of the Farm Location and Animal Population Simulator (FLAPS) [26]. FLAPS disaggregates county level National Agricultural Statistics Service (NASS) estimates of the number of premises in each county in 2012. The mismatch with the ICVI data (which is from 2009) is a caveat; however, the total number of cattle premises varied by only 6.4% from 2007 to 2017 [27] and the benefit of imputed premises obtained from FLAPS outweighs potential effects of demographic changes over time. Out of 3108 contiguous counties, 3046 and 2499 counties had at least one premises with production of beef and dairy, respectively. Though FLAPS predicts exact locations of premises within counties, these are different for each realization and there is no data available to connect ICVI data to characteristics of specific premises. Thus, we are confined to county-level modelling of shipments.

For eight states (California, Iowa, Minnesota, New York, North Carolina, Tennessee, Texas and Wisconsin), we also had access to ICVI data for 2010 and 2011, which we used for validation. For a detailed description of these data, see Gorsich et al. [4].

We also used responses from an expert elicitation survey by Beck-Johnson et al. [28]. Here, experts across multiple US states were asked about the number of shipments that cross state borders. Questions in the survey were commodity specific (beef or dairy) and named specific origin or destination premises types; the specificity in the survey allowed for inference about different cattle industry sectors and about the proportion of interstate shipments at a state level. The expert elicitation survey provided data

for 17 contiguous US states, including California, Colorado, Idaho, Iowa, Minnesota, Montana, Mississippi, Nebraska, Nevada, New York, North Carolina, Oklahoma, Pennsylvania, Tennessee, Texas, Virginia and Wisconsin. The data from specific survey questions were selected to ensure that the survey data was compatible with the information that is captured in the ICVI data. Specifically, all the survey questions that were used in this study asked about shipments originating at herds of specific sizes or at different types of premises (i.e. market, seed-stock operation) (see survey questions 7a–d, 8a–d, f and 12a–d for beef and 15a–c, 16a–c, 18a–b and 19a–c in Beck-Johnson *et al.* [28]). These survey questions were selected because they dealt with the origin of shipments just as the ICVI data used in USAMM do. Expert survey questions regarding destination premises type and those dealing with slaughter shipments were excluded from this study because the ICVI data used in USAMM is origin data and does not include slaughter shipments. Individual expert survey data was provided for the selected questions and was processed according to the methods described in Beck-Johnson *et al.* [28]. The question-level expert estimates were then combined into commodity-specific, state-level estimates for each individual expert by taking the mean over the expert responses to the selected survey questions. Two of the states in this dataset, Montana and Mississippi, did not have expert data in one of the commodity types—Montana data was available for beef but not dairy, and Mississippi data was available for dairy but not beef—and the expert data for these states only included one commodity.

For a detailed description of the survey and results, see Beck-Johnson *et al.* [28].

## 2.3. Model definition

We defined a statistical model for the probability of observing a set of shipments $\mathbf{T}$, here the shipments in the ICVI data. Information about individual premises was not available from the data, and we focused on county-level prediction, yet defined the model structure based on expectations about premises. We assumed that shipments from each premises in state $S \in \mathbf{U}$, where $\mathbf{U}$ denotes the set of contiguous US states, arise by a Poisson process with state-specific rate $\lambda_S$ (shipment·day$^{-1}$). Consequently, the rate of shipments originating from all $n_\omega$ number of premises in county $\omega \in \mathbf{E}_S$ is $\lambda_S n_\omega$, where $\mathbf{E}_S$ denotes the set of counties in state $S$. The ICVI data only include a proportion $\eta_S = 10\%$ of all interstate shipments of one year for all considered states except for New Jersey, where $\eta_S = 0\%$. We therefore introduce $\hat{\lambda}_S = \eta_S \lambda_S$ as the rate of shipments observed in the ICVI data.

Additionally, the complete probability of a transport includes $p_{\delta|\omega}$, the probability of county $\delta$ being the destination, conditional on its origin county $\omega$. The probability $p_{\delta|\omega}$ depends on the characteristics of the destination county and distance from origin county and is further derived in §2.4. Denoting the subset of shipments from county $\omega \in \mathbf{E}_s$ to county $\delta$ on day $\tau$ as $\mathbf{T}_{\omega,\delta}^{(\tau)} \subset \mathbf{T}$, the probability of observing $|\mathbf{T}_{\omega,\delta}^{(\tau)}|$ number of shipments was modelled as

$$|\mathbf{T}_{\omega,\delta}^{(\tau)}| \sim \mathrm{Poisson}(|\mathbf{T}_{\omega,\delta}^{(\tau)}| | \hat{\lambda}_S n_\omega\, p_{\delta|\omega}). \tag{2.1}$$

For New Jersey, where $\eta_S = 0\%$, the Poisson distribution is not defined and instead we used a degenerate distribution with all probability mass located at $|\mathbf{T}_{\omega,\delta}^{(\tau)}| = 0$. Using equation (2.1), the probability of observing all transports ($\mathbf{T}$) can be written as a product of the probability of the subsets $\mathbf{T}_{\omega,\delta}^{(\tau)}$, as is elaborated on in §§2.4, 2.5 and 2.6.

## 2.4. Spatial kernels

Given that a shipment originates in county $\omega \in \mathbf{E}_S$, we assumed three factors determine the probability $p_{\delta|\omega}$ of county $\delta$ being the destination: the number of premises in $\delta$, the distance $D_{\omega,\delta}$ from $\omega \in \mathbf{E}_S$ to $\delta$, and state-level differences in infrastructure. As with the origin of shipments, we assumed the probability of destination county is proportional to the number of premises $\hat{n}_\delta$, which is equal to $n_\delta - 1$ if $\delta = \omega$ and otherwise equal to $n_\delta$. Thereby, we corrected our model to exclude the possibility of a premises shipping to itself. Distance dependence was modelled with a spatial kernel $K_k$, where the subscript $k$ denotes a functional form, as elaborated on below. Finally, we account for state-level differences in import rates through parameter $w_\delta = W_S$ for county $\delta$ in state $S$. The probability of $\delta$ given $\omega \in \mathbf{E}_s$ for kernel type $k$ is given by

$$p_{\delta|\omega}^{(k)} = \frac{K_k(D_{\omega,\delta}|d_S, R_S) w_\delta \hat{n}_\delta}{\sum_{j \in \mathbf{C}} K_k(D_{\omega,j}|d_S, R_S) w_j \hat{n}_j}, \tag{2.2}$$

for $\delta \in \mathbf{C}$, where $\mathbf{C}$ denotes the set of all counties in contiguous USA. That is, the relative probability of a shipment to county $\delta$, conditional on origin county $\omega$, is normalized over all possible destination

counties. Random variables $d_S$ and $R_S$ are the kernel scale and shape parameters, respectively, as further elaborated on below.

The random variable $W_S$ can be interpreted as how likely a premises in a state is to be the destination of a shipment relative to other premises at the same distance from $\omega \in \mathbf{E}_S$. Because it is a relative measure, we set $W_{\mathrm{Missouri}} = 1$ to ensure an identifiable model. The choice of state with $W_S = 1$ is arbitrary and does not affect the results, and we choose Missouri because it had a large number of incoming shipments, thereby improving computational efficiency by avoiding a substantial uncertainty in the parameter that all other $W_S$ are referenced against.

The shape of the spatial kernel ($K_k$ in equation (2.2)) is essential for the focus of this study because the kernel characteristics at short distances has large effect on the proportion of within state shipments. We implemented three functional forms for the kernel

$$
\left.\begin{aligned}
K_1(D_{\omega,\delta}|a, b) &= \mathrm{e}^{-(D_{\omega,\delta}/a)^b} & a, b > 0 \\
K_2(D_{\omega,\delta}|a, b) &= 1 - \mathrm{e}^{-(D_{\omega,\delta}/a)^b} & a > 0, b < 0 \\
K_3(D_{\omega,\delta}|a, b) &= \frac{1}{1 + (D_{\omega,\delta}/a)^b} & a, b > 0.
\end{aligned}\right\}
\tag{2.3}
$$

$K_1$ has the form of a generalized normal distribution and includes as special cases well-known distributions such as the Gaussian normal ($b = 2$), Laplace ($b = 1$) and, as a limiting case, the uniform distribution as $b$ approaches infinity [29]. This kernel has been used in previous studies of US cattle shipments [17,18]. For low values of $b$, $K_1$ has a steep slope at short distances and a fat tail. Conversely, for high $b$, $K_1$ takes on a plateau shape at short distances with a steep decline of the tail. Hence, the behaviour at large distances restricts the kernel characteristics at short distances. Therefore, we introduce alternative kernels of the form that allow for different shapes at both short and long distances (i.e. kernels that allow for plateau shape at short distances but also fat tails describing high probability of long-distance shipments). Furthermore, these kernels have closed form solutions to a reparametrization that can be used instead of parameters $a$ and $b$ which hold no readily interpretable information. Previous studies have reparametrized $K_1$ by moment statistics [16,17], but here we use a different approach for two main reasons. First, these moment statistics provide little intuitive understanding of the behaviour of the kernels and are hence only marginally more informative than $a$ and $b$. Secondly, $K_2$ and $K_3$ include shapes that lack finite moments and it is therefore not possible to define finite quantities for all possible shapes. Instead, we employ an approach where we define the state-specific kernel scale by parameters $d_S$, defined as the distance where the kernel has dropped to half its original value, i.e. where a premises is half as likely to be the destination compared with an immediate neighbouring premises. The use of half of its initial value is used to give an intuitive understanding of the scale of the kernel. We further define kernel shape $R_S$ as the ratio between $d_S$ and the distance where the kernel value reaches some lower value $u$, here set to 5%. While the value of $u$ is somewhat arbitrary, it corresponds to a value with substantially lower kernel value compared with half of its value that used to define $d_S$. This reparametrization makes the kernels easy to visualize and express the kernels on equivalent statistics. Parameter $d_s$ has a distance unit (here km) and $R_S$ is a scale-free measure of shape. We considered only monotonously decreasing kernel functions and therefore put the restrictions $d_S \in (0, \infty)$ and $R_S \in (1, \infty)$ on the parameters. Figure 1 shows examples of shapes of the three kernel functions for equivalent parametrization.

By multiplying equation (2.1) over all possible counties not in the origin state, we obtain the total probability of observing the interstate shipments from this county at a certain day $\tau$. That is, we model the probability of the set of shipments $T_\omega^{(\tau)} = \bigcup_{\delta \in \mathbf{C} \setminus \mathbf{E}_S} \mathbf{T}_{\omega,\delta}^{(\tau)}$. For simplified notation, we introduce $\mathbf{W} = (W_1, \ldots, W_{|\mathbf{U}|})$ and $\boldsymbol{\Theta} = (\boldsymbol{\Theta}_1, \ldots, \boldsymbol{\Theta}_{|\mathbf{U}|})$, where $\boldsymbol{\Theta}_S = (d_S, R_S, \lambda_s)$. The indices of $W_S$ and $\boldsymbol{\Theta}_S$ indicate that these are state-specific parameters. With this notation, the probability of the set $T_\omega^{(\tau)}$ of shipments from county $\omega$ at time $\tau$ given kernel functional form $k$ is

$$
P_\omega^{(\tau,k)}(\mathbf{T}_\omega^{(\tau)}|\boldsymbol{\Theta}_S, \mathbf{W}) = \prod_{\delta \in \mathbf{C} \setminus \mathbf{E}_S} \mathrm{Poisson}\left(|\mathbf{T}_{\omega,\delta}^{(\tau)}||\hat{\lambda}_S n_\omega p_{\delta|\omega}^{(\tau,k)}\right).
\tag{2.4}
$$

We expanded the model to include all days and origin counties in state $S$, yielding

$$
P_{S,\mathbf{T}}^{(k)}(\mathbf{T}_S|\boldsymbol{\Theta}_S, \mathbf{W}) = \prod_{\omega \in \mathbf{E}_S} \prod_{\tau \in \boldsymbol{\tau}} P_\omega^{(\tau,k)}(\mathbf{T}_\omega^{(\tau)}|\boldsymbol{\Theta}_S, \mathbf{W}).
\tag{2.5}
$$

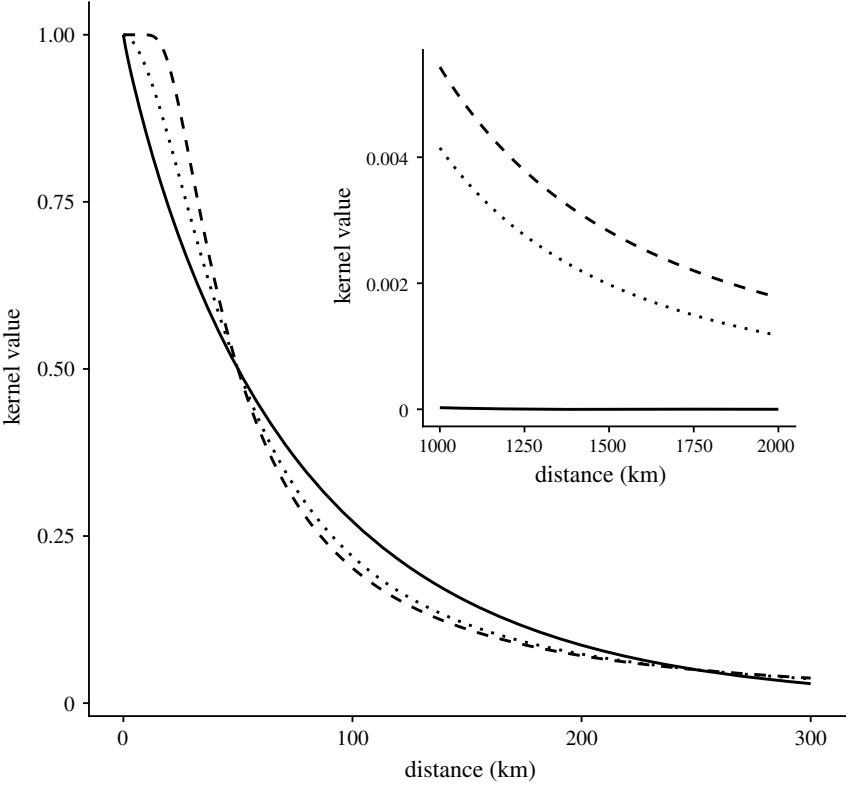

**Figure 1.** Examples of the three kernels under equivalent parametrizations ($d = 50$, $R = 5$). Inset panel shows the kernel values for larger distances.

Here, $\mathbf{T}_S = \bigcup_{\omega \in \mathbf{E}_S} \bigcup_{\tau \in \boldsymbol{\tau}} \mathbf{T}_\omega^{(\tau)}$ is the set of all shipments originating in state $S$ and $\boldsymbol{\tau}$ is the set of all days in year 2009.

Seasonality is not the focus of this study, but to investigate if seasonal trends could influence the results, we included a version of the model that also accounted for differences between the four quarters of the year. For this purpose, we let the parameters $\boldsymbol{\Theta}$ and $\mathbf{W}$ not only to be state-specific but also quarter specific, and denote the four quarters of the year as $(Q_1, Q_2, Q_3, Q_4) = \mathbf{Q}$, where $Q_1$ is the set of days in quarter one, and so on. We will for convenience drop the subscript when denoting an arbitrary quarter, and we denote parameters as $\boldsymbol{\Theta}_S^{(Q)}$ and $W_S^{(Q)}$, where the superscript $Q$ indicates that seasonality is taken into account. We also define the set of shipments from state $S$ in quarter $Q$ as $\mathbf{T}_S^{(Q)} = \bigcup_{\omega \in \mathbf{E}_S} \bigcup_{\tau \in Q} \mathbf{T}_\omega^{(\tau)}$. Similar to equation (2.5), we write the model considering quarter $Q \in \mathbf{Q}$ as

$$P_{S,\mathbf{T}}^{(k,Q)}(\mathbf{T}_S | \boldsymbol{\Theta}_S^{(Q)}, \mathbf{W}^{(Q)}) = \prod_{\omega \in \mathbf{E}_S} \prod_{\tau \in Q} P_\omega^{(\tau,k)}(\mathbf{T}_\omega^{(\tau)} | \boldsymbol{\Theta}_S^{(Q)}, \mathbf{W}^{(Q)}). \qquad (2.6)$$

We implemented the seasonality model only for kernel functional form 3. Further details on how equations (2.4), (2.5) and (2.6) are rewritten for computational convenience and implemented in practice can be found in electronic supplementary material, S1.2 and S1.4.

## 2.5. Modelling expert data

In addition to exploring different kernel shapes, we investigated if inclusion of expert opinions could further improve estimation of the proportion of interstate shipments. Bayesian analysis is well suited for this, and typically experts are used to elicit informative priors [21]. However, because of the model structure and the available expert estimates, this is not straightforward in our analysis. The parameter we wish to inform is the expected proportion of interstate shipments from a given state, here denoted $\hat{z}_S^{(k)}$, and is a function of several random variables and data. $\hat{z}_S^{(k)}$ is estimated via $p_{\delta|\omega}^{(\tau,k)}$ and is hence an implicit function of the other parameters ($d_S$, $R_S$, $\mathbf{W}$). The details of its definition and how this variable is derived can be found in electronic supplementary material, S1.2. As such, there are no

convenient means to transform available expert opinions to priors for $d_S$, $R_S$ and **W**. Furthermore, several states have a single expert respondent, providing a point estimate about expected proportions of interstate shipments. Thus, even if we could transform the questionnaire responses into an estimate about model parameters, it would not be straightforward to specify a distribution from this estimate. Instead, we expanded our model to include information from expert questionnaire responses as data. We denote the state-specific expert response from respondent $r_S \in \mathbf{r}_S$ as $\rho_{r,S}$, where $\mathbf{r}_S$ denotes the set of respondents in state $S$, and assume $\rho_{r,S}$ are distributed around the true (unknown) value $\hat{z}_S^{(k)}$, which is a function of the underlying process, quantified by model parameters $d_S$, $R_S$ and **W**. The expert data are only available at an annual level and can therefore not be used to inform parameters including seasonal variations. Hence, we omitted expert data when we used seasonal variations in our predictions.

Because $\rho_{r,S}$ is confined on the range (0,1), we modelled the expert data as logit-normally distributed around $\hat{z}_S^{(k)}$ as

$$P_{S,\boldsymbol{\rho}}^{(k)}(\boldsymbol{\rho}_S | d_S, R_S, \mathbf{W}, \xi_S) = \prod_{r \in \mathbf{r}_S} \text{logit-normal}(\rho_{r,S} | \hat{z}_S^{(k)}, \xi_S^{-1}), \tag{2.7}$$

where $\xi_S$ quantifies the precision of experts. This may be interpreted such that for large $\xi_S$, experts have an exact opinion about $\hat{z}_S^{(k)}$, close to the true value. For low $\xi_S$, the experts have only a vague idea about $\hat{z}_S^{(k)}$. The subscript $S$ expresses that $\xi_S$ is a state-specific measure, indicating that experts in some states may be more precise estimators about the proportion of shipment moving interstate than in others. The parameter $\xi_S$ is only defined for states where expert opinions are available (i.e. $\mathbf{r}_S \neq \emptyset$) and we denote this set of parameters as $\boldsymbol{\xi}$. For the states where $\mathbf{r}_S = \emptyset$, we simply define $P_{S,\boldsymbol{\rho}}^{(k)} = 1$.

## 2.6. Likelihoods

We considered likelihoods for four assemblies of the data. These include three at annual level: (ICVI only, experts only, and ICVI and experts combined), and one for the case with seasonality, for which expert data was not available. The full likelihood for ICVI only for all states (i.e. the set **T**) is defined via equation (2.5) as

$$L_{\mathbf{T}}^{(k)}(\mathbf{T} | \boldsymbol{\Theta}, \mathbf{W}) = \prod_{S \in \mathbf{U}} P_{S,\mathbf{T}}^{(k)}(\mathbf{T}_S | \boldsymbol{\Theta}_S, \mathbf{W}). \tag{2.8}$$

When using expert data only, we defined the full likelihood in terms of the random variable $v_S$, defined as logit of the estimated proportion of interstate shipments, rather than $\hat{z}_S^{(k)}$, which has the same definition (with the difference of not being logit transformed). However, $\hat{z}_S^{(k)}$ is a function of several random variables ($d_S$, $R_S$ and **W**), and not meaningful in the absence of the ICVI data. Based on equation (2.7), the likelihood for experts only is defined as

$$L_{\boldsymbol{\rho}}^{(k)}(\boldsymbol{\rho} | \boldsymbol{v}, \boldsymbol{\xi}) = \prod_{S \in \mathbf{U}} P_{S,\boldsymbol{\rho}}^{(k)}(\boldsymbol{\rho}_S | v_S, \xi_S), \tag{2.9}$$

with state-specific $v_S \in \boldsymbol{v}$, $\xi_S \in \boldsymbol{\xi}$ and $\rho_S \in \boldsymbol{\rho}$, where $\boldsymbol{v}$, $\boldsymbol{\xi}$ and $\boldsymbol{\rho}$ are the set of logit-mean of expert opinions, the set of precision parameters, and the full set of expert data, respectively. Analogously, the full likelihood for ICVI and expert data combined is defined as

$$L_{\mathbf{T},\boldsymbol{\rho}}^{(k)}(\mathbf{T}, \boldsymbol{\rho} | \boldsymbol{\Theta}, \mathbf{W}, \boldsymbol{\xi}) = \prod_{S \in \mathbf{U}} [P_{S,\mathbf{T}}^{(k)}(\mathbf{T}_S | d_S, R_S, \lambda_S, \mathbf{W}) P_{S,\boldsymbol{\rho}}^{(k)}(\rho_{S,\mathbf{r}} | d_S, R_S, \mathbf{W}, \xi_S)]. \tag{2.10}$$

The full likelihood $L_{\mathbf{T},\boldsymbol{\rho}}^{(k)}$ (equation (2.10)), including ICVI and expert data, is merely a product of the two likelihoods $L_{\mathbf{T}}^{(k)}$ and $L_{\boldsymbol{\rho}}^{(k)}$.

Finally, when ICVI data are considered at seasonal quarter level, the full likelihood is written as

$$L_{\mathbf{T}}^{(k,\mathbf{Q})}(\mathbf{T} | \boldsymbol{\Theta}^{(\mathbf{Q})}, \mathbf{W}^{(\mathbf{Q})}) = \prod_{S \in \mathbf{U}} \prod_{Q \in \mathbf{Q}} P_{S,\mathbf{T}}^{(k,Q)}(\mathbf{T}_S | \boldsymbol{\Theta}_S^{(Q)}, \mathbf{W}^{(Q)}). \tag{2.11}$$

In total, we implemented eight different likelihood functions: six for the ICVI data using the three different kernels in the absence or presence of expert data, one for the expert data only and one using ICVI data at quarter level focusing on kernel three only. We analysed beef and dairy shipments separately, since the two production types have different farming practices and consequently are likely to differ in their parameter estimates.

## 2.7. Hierarchical Bayesian model

We implemented a hierarchical Bayesian model for parameter estimation. This approach provides intelligible estimates regarding parameter uncertainty, which may be incorporated when the models are used for prediction. For $d_S$, $R_S$, $\lambda_S$ and $\xi_s$ as well as their quarter-specific equivalents, we implemented

$$
\left.
\begin{aligned}
d_S &\sim \text{lognormal}(m_{\mathbf{d}}, \kappa_{\mathbf{d}}) \\
R_S &\sim \text{lognormal}_{m-1}(m_{\mathbf{R}}, \kappa_{\mathbf{R}}) \\
\lambda_S &\sim \text{gamma}(m_{\boldsymbol{\lambda}}, \kappa_{\boldsymbol{\lambda}}) \\
\xi_S &\sim \text{gamma}(m_{\boldsymbol{\xi}}, \kappa_{\boldsymbol{\xi}}),
\end{aligned}
\right\}
\tag{2.12}
$$

where subscript $m-1$ indicates that the prior for $R_S$ is shifted one unit to the right since $R_S$ is defined on the interval $(1, \infty)$. Here, $m$ and $\kappa$, denote the mean and coefficient of variation, respectively, and their relationship to the standard parametrization of the lognormal distribution is $m = e^{\mu - (\sigma^2/2)}$ and $\kappa = (e^{\sigma^2} - 1)^{1/2}$ where $\mu$ and $\sigma$ are the mean and standard deviation, respectively, of the logarithm of the variable. The lognormal and shifted lognormal distribution differ in their relationship to standard parametrization in the way the mean is calculated. For the shifted distribution, the mean is expressed as $m_{\mathbf{R}} = e^{\mu - (\sigma^2/2)} + 1$, whereas the calculation of $\kappa$ is unchanged. For the gamma distribution, the mean and coefficient of variation are expressed as $m = \alpha\beta^{-1}$ and $\kappa = \alpha^{-(1/2)}$, respectively, where $\alpha$ and $\beta$ denote shape and rate in the standard parametrization, respectively. Using a hierarchical model structure, $m$ and $\kappa$ parameters were treated as random variables, and the alternative parametrization facilitates cognizant hyperprior elicitation as specified in §2.8. Denoting the prior distribution for $x_S \in \{d_S, R_S, \lambda_S, \xi_S\}$ as $\Phi_x(x_S \mid m_x, \kappa_x)$ and corresponding hyperpriors as $\Psi_m^{(x)}(m_x)$ and $\Psi_\kappa^{(x)}(\kappa_x)$, the full Bayesian model is given as

$$
\begin{aligned}
&P_{\mathbf{T},\boldsymbol{\rho}}^{(k)}(\boldsymbol{\Theta}, \mathbf{W}, \boldsymbol{\xi}, m_{\mathbf{d}}, \kappa_{\mathbf{d}}, m_{\mathbf{R}}, \kappa_{\mathbf{R}}, m_{\boldsymbol{\lambda}}, \kappa_{\boldsymbol{\lambda}}, m_{\boldsymbol{\xi}}, \kappa_{\boldsymbol{\xi}} \mid \mathbf{T}, \boldsymbol{\rho}) \\
&\propto L_{\mathbf{T},\boldsymbol{\rho}}^{(k)} \prod_{S \in \mathbf{U}} [\Phi_{\mathbf{d}} \Phi_{\mathbf{R}} \Phi_{\mathbf{W}} \Phi_{\boldsymbol{\lambda}} \Phi_{\boldsymbol{\xi}}] \cdot \Psi_m^{(\mathbf{d})} \Psi_\kappa^{(\mathbf{d})} \Psi_m^{(\mathbf{R})} \Psi_\kappa^{(\mathbf{R})} \Psi_m^{(\boldsymbol{\lambda})} \Psi_\kappa^{(\boldsymbol{\lambda})} \Psi_m^{(\boldsymbol{\xi})} \Psi_\kappa^{(\boldsymbol{\xi})} \\
&= \prod_{s \in \mathbf{U}} [P_{S,\mathbf{T}}^{(k)} P_{S,\boldsymbol{\rho}}^{(k)} \Phi_{\mathbf{d}} \Phi_{\mathbf{R}} \Phi_{\mathbf{W}} \Phi_{\boldsymbol{\lambda}} \Phi_{\boldsymbol{\xi}}] \cdot \Psi_m^{(\mathbf{d})} \Psi_\kappa^{(\mathbf{d})} \Psi_m^{(\mathbf{R})} \Psi_\kappa^{(\mathbf{R})} \Psi_m^{(\boldsymbol{\lambda})} \Psi_\kappa^{(\boldsymbol{\lambda})} \Psi_m^{(\boldsymbol{\xi})} \Psi_\kappa^{(\boldsymbol{\xi})}
\end{aligned}
\tag{2.13}
$$

for ICVI and expert data combined. In equation (2.13), $d_S \in \mathbf{d}$, $R_S \in \mathbf{R}$, $\lambda_S \in \boldsymbol{\lambda}$ and $\xi_s \in \boldsymbol{\xi}$, where bold symbols denote the set of parameters across all included states.

Prior distributions $\Phi_{\mathbf{W}}(W_S)$ for models including ICVI data and $\Phi_{\mathbf{v}}(v_S)$ for models with only experts were included as fixed distributions, without hierarchical structure. Further details of equation (2.13) and definition of models including ICVI or expert data only and model including seasonality can be found in electronic supplementary material, S1.3.

## 2.8. Prior elicitation

Our general approach for eliciting prior and hyperprior distributions was to first identify the range on which the parameters are defined and then choose suitable prior distributions with domains matching that range. To obtain statistics about which we can have at least a minimal intuitive expectation, we expressed priors by their mean ($m_x$) and coefficient of variation ($\kappa_x$) for parameters $x \in (\mathbf{d}, \mathbf{R}, \lambda, \xi)$. For these parameters, we deduced hyperpriors by specifying a range of plausible values in which we, with 95% certainty, believe encapsulates the true parameter value. By choosing the range 95%, we do not exclude more extreme values but consider them less likely. The model's sensitivity to our choices of hyperpriors was evaluated by choosing alternative hyperpriors as described in §2.9.

Because $\mathbf{d}$ is defined on the range $(0, \infty)$, the prior distribution $\Phi_{\mathbf{d}}(d_S \mid m_{\mathbf{d}}, \kappa_{\mathbf{d}})$ was chosen as lognormal. We further specified the hyperpriors for $m_{\mathbf{d}}$ and $\kappa_{\mathbf{d}}$ as lognormal distributions and used our general approach of identifying a range of plausible values. For the hyperprior $\Psi_m^{(\mathbf{d})}(m_{\mathbf{d}})$, we elicited hyperparameters such that the average distance to where destinations are half as likely as an immediate neighbour is within the range of 10 km and 4000 km. We find these to be reasonable values for a vague hyperprior because 10 km is a very short shipment distance considering the spatial scales of the US livestock system and 4000 km is the approximate distance between the east and west coast of the USA, which would be a high value for the average distance to where a destination premises is half as likely as an immediate neighbour. Since $\mathbf{R} \in (1, \infty)$ is not defined on the whole positive real line we implemented a lognormal prior distribution shifted one unit to the right ($\Phi_{\mathbf{R}}(R_S \mid m_{\mathbf{R}}, \kappa_{\mathbf{R}})$). For the hyperprior $\Psi_m^{(\mathbf{R})}(m_{\mathbf{R}})$, we chose prior parameters such that the interpretation is that

we put 95% of the density of the mean of $\mathbf{R}$ between 2 and 1000. This hyperprior allows for a wide range of plausible values of $\mathbf{R}$. A value of 2 corresponds to a steep drop of the kernel between distances $d$ and $2d$ (where the kernel attains 5% of its initial value). Conversely, a value of 1000 corresponds to a flat kernel (the corresponding decrease in kernel value occurs between the distances $d$ and $1000d$). Thus, this range mirrors our vague *a priori* beliefs regarding the distance dependence. To achieve conjugacy, we implemented the gamma distributions as prior for the rate parameters $\lambda$ and expert precisions $\xi$, denoted $\Phi_\lambda(\lambda_S \mid m_\lambda, \kappa_\lambda)$ and $\Phi_\xi(\xi_S \mid m_\xi, \kappa_\xi)$, respectively. Similar to the kernel parameters, we defined the hyperprior in terms of means and coefficient of variation (i.e. $m_\lambda$, $\kappa_\lambda$ and $m_\xi$, $\kappa_\xi$). For hyperprior $\Psi_m^{(\lambda)}(m_\lambda)$, we implemented a lognormal distribution such that the prior distribution of $m_\lambda$ has 95% of the density between 0.00027 and 0.27 shipments per day. These numbers were derived from the vague prior beliefs that 95% of the density of the average rate of yearly shipments per premises in the average state, is between 0.1 and 100 shipments. Furthermore, when expert data were included, the hyperprior was also chosen as the lognormal, and parameters for the mean of the precision parameter were chosen such that 95% of its density lay within the range (0.1, 10). Precision parameters within this interval will allow wide as well as narrow distributions of the expert opinions and therefore constitutes a vaguely informative hyperprior for $m_\xi$.

To define hyperpriors for the coefficients of variation, $\Psi_\kappa^{(d)}(\kappa_d)$, $\Psi_\kappa^{(R)}(\kappa_R)$, $\Psi_\kappa^{(\lambda)}(\kappa_\lambda)$ and $\Psi_\kappa^{(\xi)}(\kappa_\xi)$, we expressed our beliefs in terms of expectations regarding how similar underlying parameters are between states. We therefore used the ratio between the median of corresponding $m_x$ and its 97.5th percentile. For this ratio, we chose the lower limit as two, which corresponds to the case of high similarity between states regarding the underlying parameters $\mathbf{d}$, $\mathbf{R}$, $\lambda$ and $\xi$. Thus, we obtained a ratio of the 97.5th and the 2.5th percentile equal to four. Furthermore, we chose the upper limit such that it corresponds to a ratio of one order of magnitude, i.e. ten. As a consequence, the ratio of the 97.5th and the 2.5th percentile is equal to two orders of magnitude. This ratio corresponds to distributions expressing large differences in the state-specific estimates. From the limits above, we deduced the hyperprior distributions $\Psi_\kappa^{(d)}(\kappa_d)$, $\Psi_\kappa^{(R)}(\kappa_R)$, $\Psi_\kappa^{(\lambda)}(\kappa_\lambda)$ and $\Psi_\kappa^{(\xi)}(\kappa_\xi)$ as lognormal with 95% of its density between 0.3650 and 1.724. We chose the prior distribution of $\mathbf{W}$, $\Phi_{\mathbf{W}}(W_S)$ as a lognormal distribution with parameters such that 95% of the density lay in the range (0.01, 100). That is, we formulated a vague hyperprior expressing that plausible values of the propensity to attract shipments parameter for a premises in a certain state ranges between 0.01 and 100 times the corresponding parameter of a premises in our arbitrarily chosen reference state Missouri. Furthermore, in the analyses of expert data only (equation (2.9)), we implemented $\Phi_\nu(\nu_S) \propto 1/(\nu(1-\nu))$ for $\nu_S \in (0, 1)$, i.e. the prior for $\mathrm{logit}(\nu_S)$ is uniform.

## 2.9. Sensitivity analysis

To investigate the sensitivity of the posterior to our choice of hyperpriors, we conducted a sensitivity analysis to assess the robustness of our results. To identify parameters of potential concern, we used the criterion that marginal posteriors that had more than 1% density of either tail outside of the interval of the hyperprior containing 95% of the density could indicate instances where our choice of hyperpriors restricted the posterior range. Thus, if the first percentile of the marginal posterior distribution was lower than the 2.5th percentile of the hyperprior, we investigated the effect of the elicited hyperprior by decreasing the lower bound of the 95% interval by which it was defined by 50%. Analogously, we adjusted the hyperprior if the 99th percentile of the marginal posterior was greater than the 97.5th percentile by doubling this upper bound. We then re-analysed the data and compared the estimates corresponding to the elicited and alternative hyperpriors.

We also compared the models at annual level with the model where seasonality was included to investigate if seasonal variations could influence our results. Because no seasonal estimates were available in the expert data, we focused on ICVI modelling only.

## 2.10. Computation

None of the Bayesian models (e.g. equation (2.13)) have a standard form, and we therefore relied on numerical algorithms to estimate the posterior (see electronic supplementary material, S1.4 for details). We used Markov chain Monte Carlo (MCMC) methods to approximate the marginal posterior distributions. The idea of the MCMC approach is to simulate a Markov Chain whose limiting state distribution is equal to the posterior and from this obtain samples of the parameters. For most of our

model parameters, the conditional distribution is not of a standard form, and we implemented Metropolis–Hastings updates of these parameters [30].

## 2.11. Model selection and validation

To rank our proposed models by their level of parsimony, we used two types of information criteria: deviance information criterion (DIC) [31] and widely applicable information criterion (WAIC) [32]. Both criteria are derived from estimating the out-of-sample predictive accuracy using within-sample data and a penalty term for the overestimation of the accuracy this leads to [33]. Both criteria have similar interpretation; a lower score indicates the preferred model. DIC estimates the fit from the log-likelihood (or log predictive density) of the data conditional on the posterior means of the parameters, and the penalty term used in DIC is equal to the effective number of parameters [31]. We chose, however, to use the median of the parameters (as proposed in [31]) since it proved to increase numerical stability in our study because posterior densities typically exhibited high skewness. In practice, DIC is easily estimated in MCMC algorithms and is therefore a convenient tool for model selection. However, DIC is not fully Bayesian since the log predictive density conditions on point estimates (in our case the median), and concerns have been raised about a tendency for DIC to favour complex models [34]. We therefore used WAIC as an additional measure for model selection, which is a more fully Bayesian approach. WAIC uses the (computed) log pointwise posterior predictive density to estimate the out-of-sample predictive accuracy. This pointwise approach means that WAIC better captures the posterior uncertainty than DIC. For WAIC, two types of penalty terms have been proposed. We chose to follow the advice of Gelman *et al.* [33] and used the posterior sample variance of the log predictive density. For computational reasons, we used a subsample of the iterations in the MCMC algorithm when calculating WAIC and we chose 20 000 random samples of our parameters from the last 2 000 000 iterations. The measures above provide information on the accuracy of our model's predictive ability and we therefore implemented an additional strategy for validation. We compared predictions of the models to ICVI data, including other years' data where available, by network summary statistics to ensure our model re-captured relative features of the original data. For these purposes, we simulated 1000 networks from each model based on random draws from the posterior distribution. We compared distributions of observed shipment distances in the ICVI data to corresponding distributions based on posterior predictive simulations. For the sake of comparison, the latter were down-sampled to 10% interstate shipments and no intrastate shipments. Furthermore, we calculated for each state the correlation between ICVI data and simulated networks in terms of destination states.

## 3. Results

Our models for the ICVI data included four state-level parameters describing the shipment pattern: one modelling shipment rate describing the rate at which premises in each state generate shipments ($\lambda_S$), one modelling propensity to attract shipments ($\hat{W}_S$), and two kernel parameters modelling how the probability of shipments decay with distance (scale, $d_S$, and shape, $R_S$). The analyses revealed a heterogeneous shipment pattern across the USA, with large variation in all estimated parameters across states and production systems, as exemplified for four selected states in figure 2. Parameters $\lambda_S$ and $\hat{W}_S$ varied among states by more than an order of magnitude, and estimates were similar across the three kernels. The kernel parameters $d_S$ and $R_S$ also varied substantially among states, pinpointing the importance of accounting for state-level heterogeneity when assessing shipment distances. These estimates, however, depended heavily on the choice of functional form. Whereas parameters for kernels two and three showed great similarity, estimates for kernel one consistently differed from the other two. Notably, kernel one estimates for $d_S$, defined as the distance where the probability of destination has dropped to half of the probability of an immediate neighbour, were often remarkably low, including estimates below $10^{-7}$ km. The parameter values were, however, similar when comparing across seasons, which is depicted in electronic supplementary material, figure S1. Estimates for all parameters and kernels at annual level are shown in electronic supplementary material, S1.7.

Estimates for expert precision parameters ($\xi$) were similar for all kernels, and the credibility intervals largely overlapped for all states where expert data were available. Detailed results for $\xi$ are shown in electronic supplementary material, S1.8.

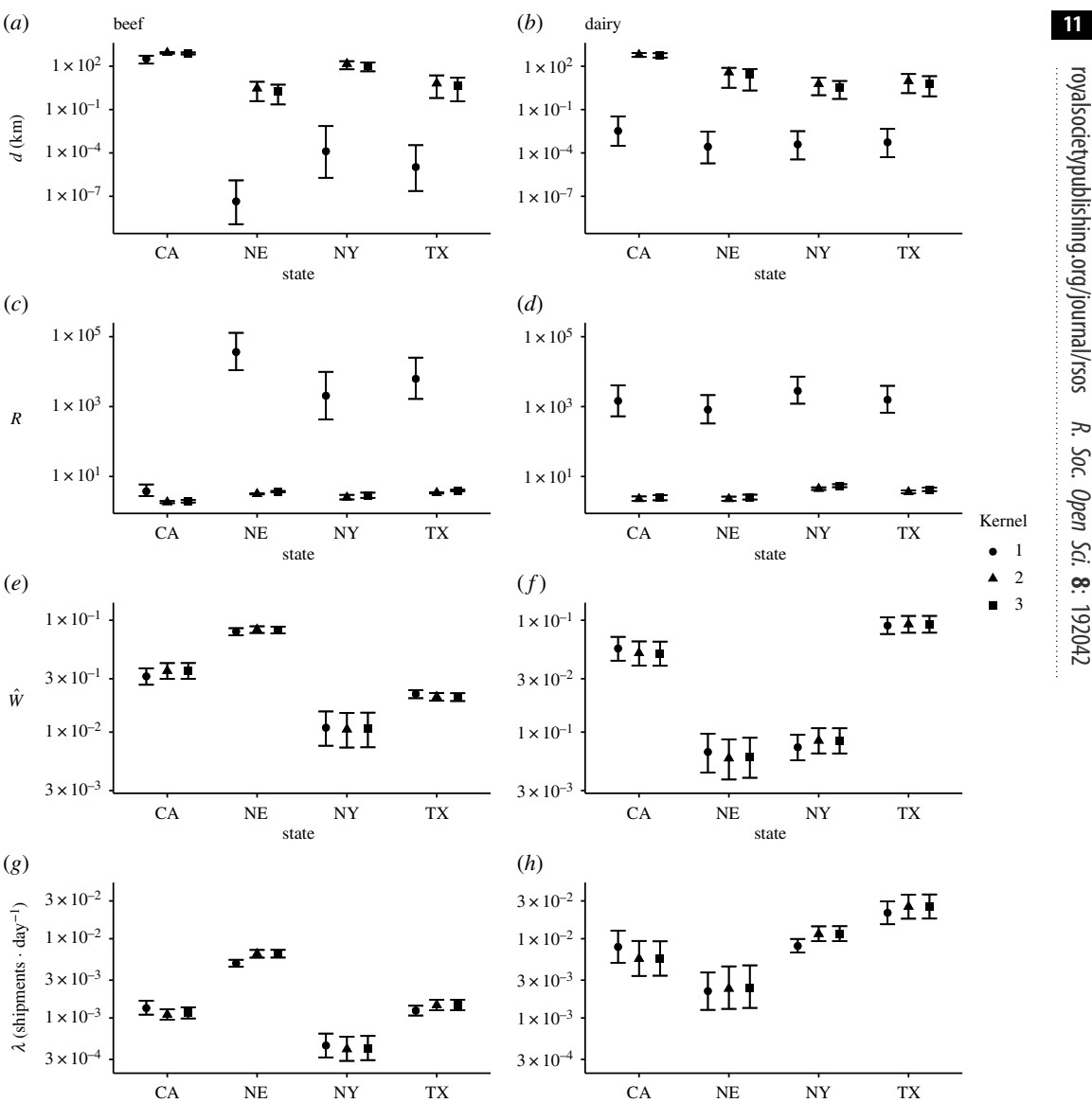

**Figure 2.** 95% credibility intervals for selected states and all implemented kernels. The rows show from top to bottom, **d** (km), shape parameter **R** (unitless), propensity to attract shipments **Ŵ** (unitless) and shipment rate $\lambda \cdot$ (shipment · day$^{-1}$). Left and right columns show credibility intervals for beef and dairy, respectively. Results are shown for analysis including expert data.

The variability in the underlying parameters elicited a heterogeneous pattern in terms of the predicted proportion of shipments moving intrastate (figure 3), and the number of shipments moving in and out of each state varied substantially for both beef and dairy shipment (figure 5). These patterns cannot be explained simply from the proportion of premises located in each state, which is shown in electronic supplementary material, figure S2, and it is clear that distance dependence as well as geographical differences in the propensity to ship and receive shipments are important to capture the heterogeneous shipment pattern. Figures 3 and 4 further show that this heterogeneous pattern is not as prominent when comparing across kernels or seasons.

## 3.1. Model selection and the effect of experts

Independent of selection criteria (table 1), model selection consistently disfavoured the kernel functional form one. The choice between functional forms two and three varied between datasets (beef or dairy), but selection criteria were considerably more similar than either of these two was to functional form one. Furthermore, the posterior predictive distributions of shipment distances showed high similarity between functional forms two and three and fit better with the distances observed in the ICVI data

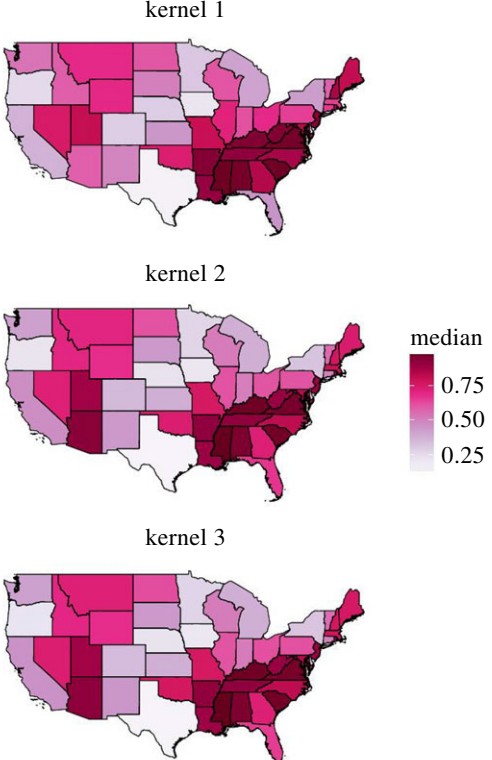

**Figure 3.** Predicted proportion of shipments moving interstate for beef and dairy combined based on three different functional forms of the spatial kernel. Results are shown for analyses where experts were included for available states.

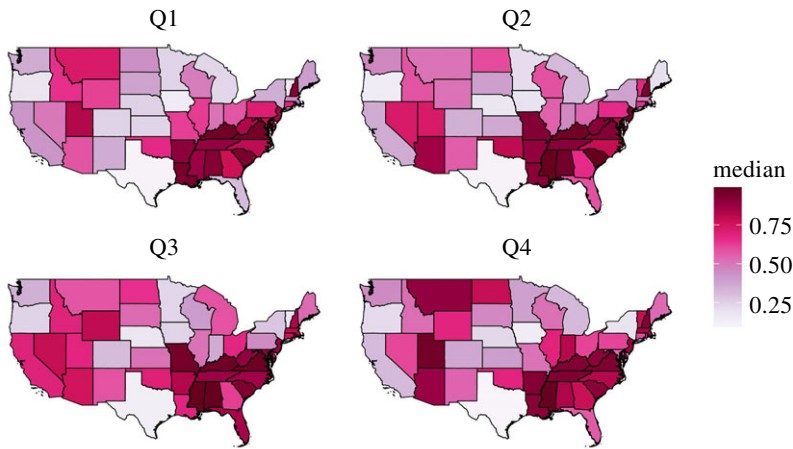

**Figure 4.** Predicted proportion of shipments moving interstate for beef and dairy combined based on kernel three for each quarter.

than did functional form one (figure 6). The choice of kernel functional form only had minor effect on the large-scale contact pattern, providing near identical median estimates of in- and out-degree (figure 5). Thus, we here primarily focus on functional form three and present equivalent analyses for the other kernels in electronic supplementary material, S1.7, S1.8, S1.11, S1.13.

The inclusion of experts did not change ranking of kernels in terms of model selection (table 1). Figure 7 further shows that the posterior predictive estimates of $p$ using experts-only data typically provided wide intervals of expected proportion of shipments moving intrastate. Consequently, their contribution to the analysis of both experts and ICVI data was moderate, shifting only slightly the corresponding distributions of $p$. The experts also had only marginal effect on the models' ability to recapture between-state link strengths (figure 8). Because of the low impact of experts on the estimates, we primarily focus on estimates including expert and provide non-expert analyses in electronic supplementary material, S1.7, S1.9, S1.10, S1.11, S1.13.

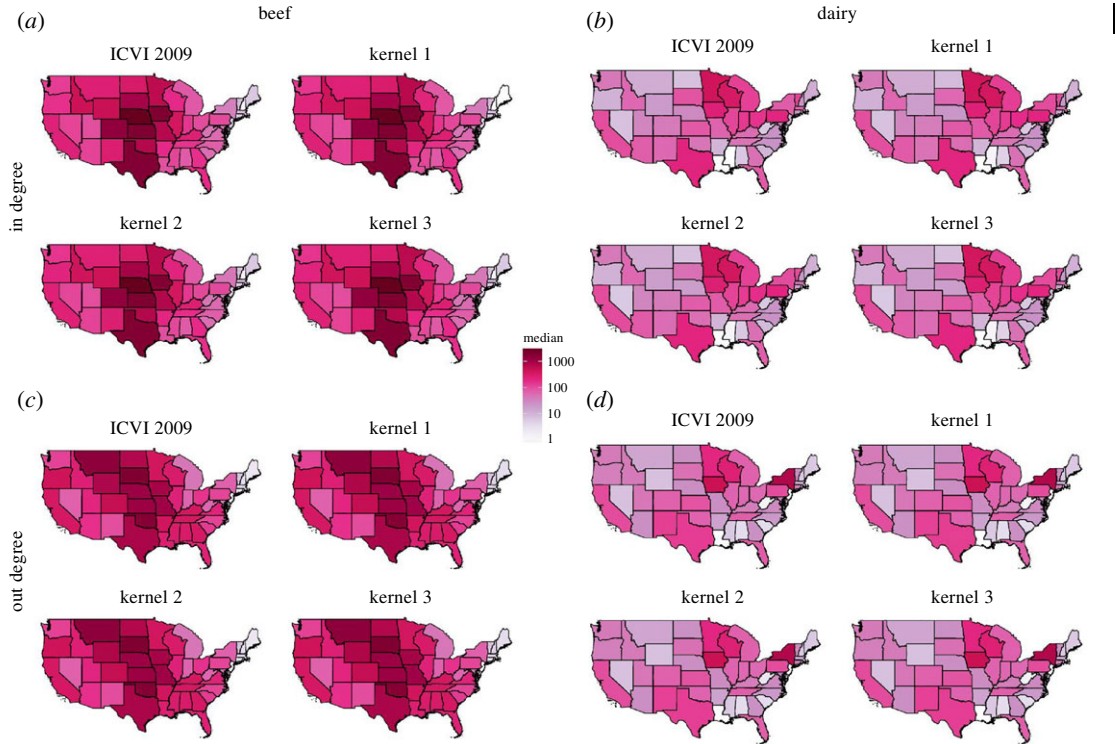

**Figure 5.** Number of incoming ((*a*) beef and (*b*) dairy) and outgoing ((*c*) beef and (*d*) dairy) shipments for US states as given by the ICVI data and the corresponding median prediction from 1000 realizations with each implemented kernel. Simulated shipments were down-sampled by 90% to correspond to the ICVI data. Results are shown for analyses where experts where included for available states.

**Table 1.** Differences from the preferred kernel functional form for two model selection criteria ($\Delta$WAIC and $\Delta$DIC) for each considered dataset. The minimum number of independent samples includes in parenthesis for which (hyper)parameter (and state where applicable) the minimum value was estimated for.

| commodity | data | kernel | $\Delta$WAIC | $\Delta$DIC | minimum number of independent samples |
|---|---|---|---|---|---|
| beef | ICVI | 1 | 625.0 | 630.0 | 6873 ($W$ VT) |
| | | 2 | 14.2 | 11.5 | 2445 ($d$ OH) |
| | | 3 | 0.0 | 0.0 | 3805 ($d$ OH) |
| | ICVI + experts | 1 | 626.4 | 630.1 | 9505 ($W$ OH) |
| | | 2 | 21.0 | 10.4 | 2348 ($\lambda$ OH) |
| | | 3 | 0.0 | 0.0 | 6071 ($R$ FL) |
| dairy | ICVI | 1 | 530.0 | 531.0 | 4724 ($\sigma_d$) |
| | | 2 | 0.0 | 0.0 | 4876 ($W$ IN) |
| | | 3 | 45.2 | 45.6 | 6050 ($W$ IA) |
| | ICVI + experts | 1 | 510.0 | 512.1 | 6631 ($W$ WI) |
| | | 2 | 0.0 | 0.0 | 6792 ($W$ OR) |
| | | 3 | 45.3 | 46.0 | 11 47 ($W$ MT) |

## 3.2. Validation

We performed both within- and out-of-sample validation. The posterior predictions recaptured the relevant large-scale structure of the 2009 ICVI data, with the number of incoming and

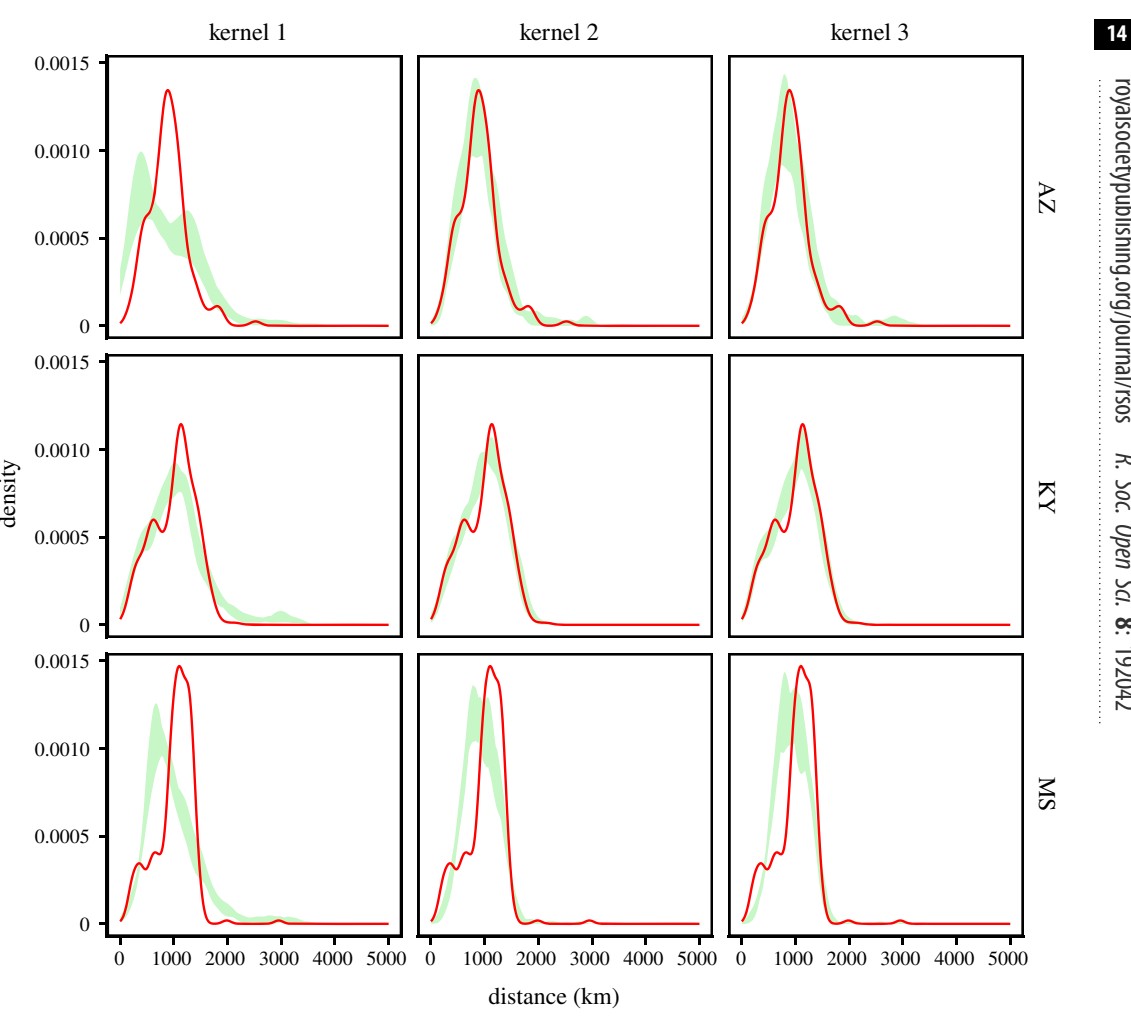

**Figure 6.** Distributions of interstate shipment distances for Arizona, Kentucky and Mississippi, comparing ICVI data (red) to posterior predictions (green) to models with different functional forms for the kernel. Results are shown for dairy and beef shipments combined for analyses where experts where included for available states.

outgoing shipments showing striking similarity to the ICVI data (figure 5). The analysis of between-state link strengths showed a high correlation between posterior predicted networks and the ICVI data (figure 8).

Importantly, we were also able to use additional ICVI data from 2010 and 2011 for selected states to perform out-of-sample validation. Independent of kernel functional form, we found the link strength from these states correlated well with USAMM predictions (figure 9), with the exception of 2010 and 2011 shipments from North Carolina and 2010 shipments from Nebraska. In all instances, correlations of 2010 and 2011 ICVI data with USAMM predictions were comparable to correlations of 2010 and 2011 ICVI data with correlations 2009 ICVI data, indicated as red diamonds in figure 9. This result and the overall high correlation in figure 8 show that regardless of choice of kernel, the model captures the shipping patterns. That is, both within- and out-of-sample validation verifies our models at the state level.

Comparing distance distribution for these states (figure 10) revealed similar shipment distances across years, with the exception of Iowa and Minnesota. The 95% posterior predictive bands typically do not fully envelope the observed distribution curves for all distances, typically if the distributions have more than one mode. Still, USAMM captured the broad pattern of shipment distances. The sensitivity analysis showed that using a wider hyperprior only had a minor influence on parameter estimates and did not change ranking of models. A comprehensive presentation of the results of the sensitivity analysis is provided in electronic supplementary material, S1.12.

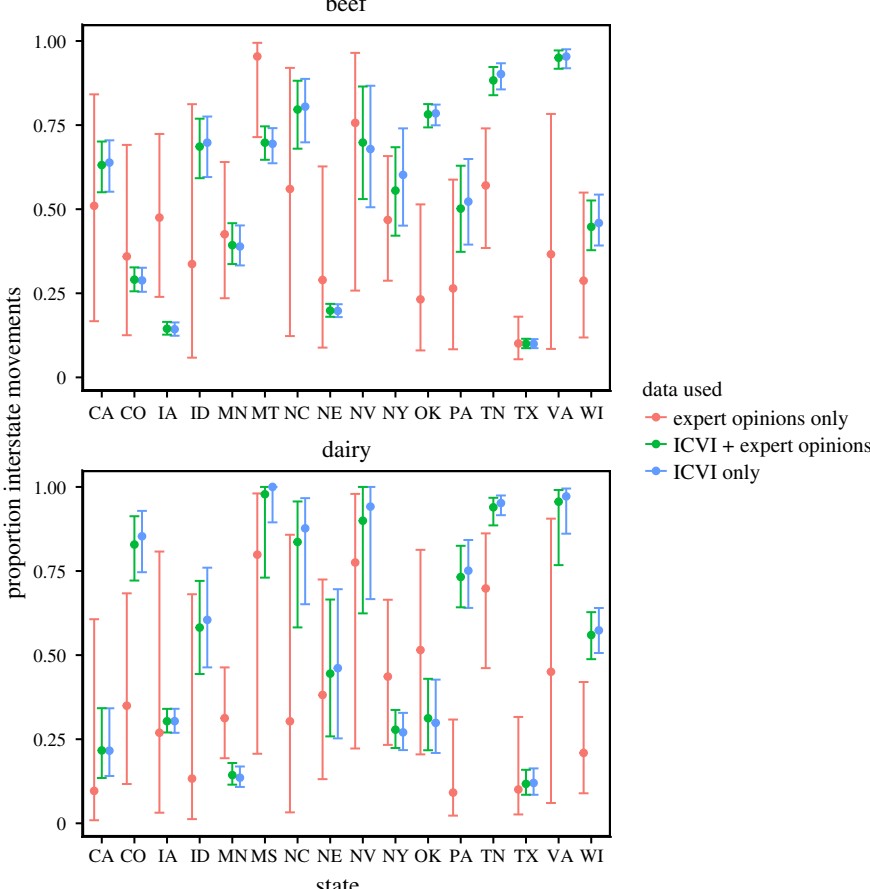

**Figure 7.** Median and 95% credibility interval of posterior distributions of proportion of movements moving interstate for analysis including only experts (red), ICVI and experts together (green) and ICVI data only (blue). Results are shown for analyses with kernel functional form three.

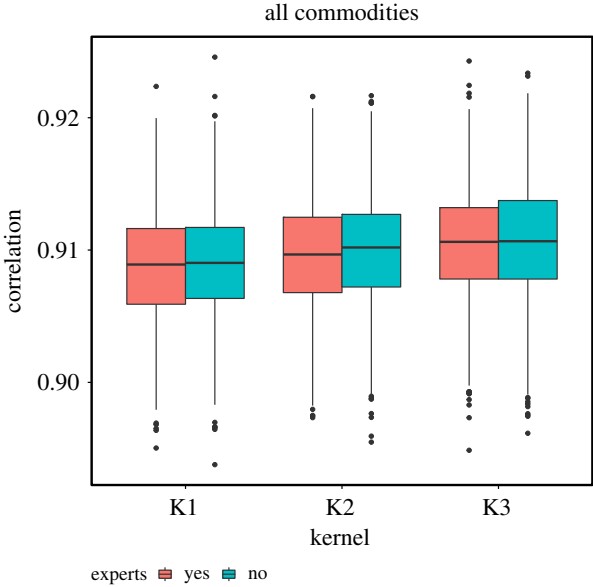

**Figure 8.** Boxplots of correlations between the 2009 ICVI data and 1000 predicted networks with three different kernel functional forms, with and without expert data, in terms of number of shipments between all contiguous states. Boxes show inter-quartile range and horizontal lines within boxes indicate median. Whiskers show highest and lowest values within 1.5 times inter-quartile range and dots represents outliers. Results are shown for networks with beef and dairy shipments combined.

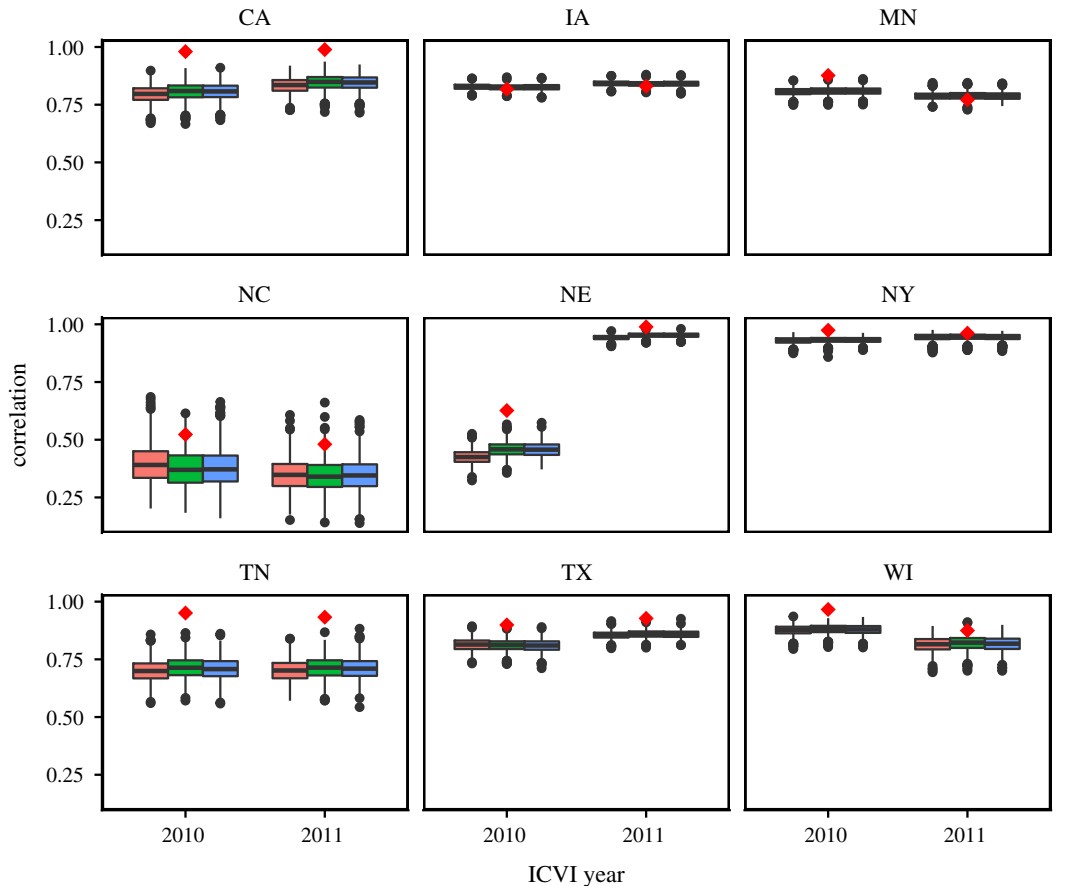

**Figure 9.** Pearson correlations for number of interstate shipments between networks (1000 realizations, boxplots) generated with USAMM with different kernel functional forms, fit to ICVI data from 2009, and ICVI data for 2010 and 2011. Corresponding correlations between the 2009 ICVI data and ICVI data for 2010 and 2011 are indicated as red diamonds. Beef and dairy combined.

## 3.3. Computation

The potential scale reduction factor (PSRF, [35]) investigates convergence for MCMC chains, and a value close to one is expected if all chains have converged. For the models considered here, all PSRF were less than 1.001, indicating sufficient convergence. We further estimated the number of independent samples and table 1 shows that the lowest value across models and data was estimated at 2348 for models excluding seasonality. For models including seasonality, the lowest number of independent samples was estimated at 1146.

## 4. Discussion

Understanding between premises contact patterns is essential for epidemiological modelling of domestic animal diseases [12,36] and can be used to inform preparedness [37,38] and surveillance [4,39]. Animal shipments are of particular importance because of their high risk [3] and potentially large spatial scales [18,40]. However, information on complete shipments is not always accessible, necessitating methods that scale up from available information.

The version of USAMM presented here includes several important improvements from previous versions. We focused on improving state-level estimates, specifically targeting the proportion of shipments moving within-state. Building on the framework of Lindström *et al.* [17], we made three important improvements for predicting inter- and intrastate animal shipments. First, we estimated the

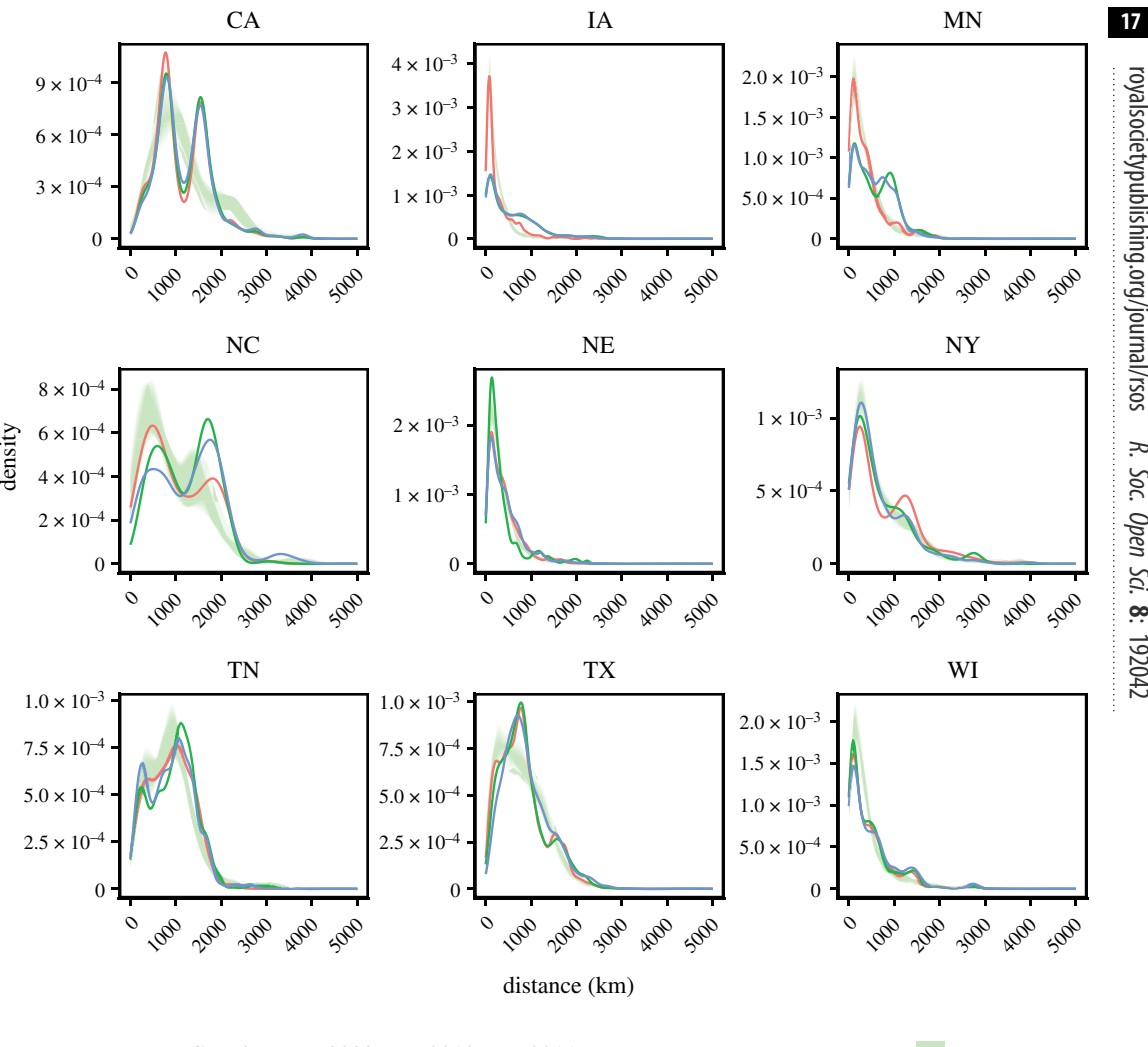

**Figure 10.** Distance distributions of movement distances in ICVI data of 2009, 2010 and 2011 for nine states and 95% predictive band of model of movement distances predicted by USAMM (kernel three, including experts).

propensity to attract shipments (**W**) in the model, rather than *a priori* specifying it as a fixed constant, as was done by Lindström *et al*. This advancement improved the correlation in between-state link strength to greater than 0.9 (figure 8) compared with approximately 0.8 in the model of Lindström *et al*., indicating the models presented here capture the overall state pattern better than previous versions of USAMM.

Secondly, we changed the model structure to model rate of shipments per premises ($\lambda$) rather than number of shipments per state. This change offers a more intuitive sense of the parameter and is more likely to be reported in other studies since it is interpreted as premises-level rate rather than a state-level quantity, which depends on the number of premises in the state. For the same reasons, it also facilitated prior elicitation and fits better in a hierarchical model structure. Furthermore, the changes improved computation since the rate parameter could be Gibbs-sampled through the conditional distribution, given a conjugate gamma prior. Importantly, it also allowed for inclusion of shipment rate in model selection based on WAIC.

Thirdly, we implemented three different functional forms for the spatial kernel that model how the probability of destination decays with distance. In countries where complete data is available, shipments typically occur at shorter distances [16,41,42], and it is therefore important when modelling outbreaks of infectious diseases to have accurate estimations of the kernel's behaviour at distances where the majority of shipments occurs. Yet, it is equally important to accurately predict shipments at long distances since these can spread pathogens to previously uninfected areas and spark new local outbreaks [3,43]. Kernels two and three take on a plateau shape at short distances, even when the estimated shape encompasses a fat tail. Conversely, kernel one exhibits a peaked behaviour at short distances when the kernel tail is fat. It is particularly essential to investigate the effect of these

differences since the behaviour at short distances determines how many shipments are estimated to move within-state. The model selection consistently disfavoured kernel one (table 1), which has the functional form used by Lindström *et al.* [17]. This illustrates that this kernel did worse than the other two in representing shipment distances at both short and long distances. Furthermore, kernel one estimates of scale parameter $d$ included values below $10^{-7}$ km (figure 2). These are exceptionally short distances considering the large spatial scales of the USA; most of the drop in probability occurs at shorter distances than possible to observe in the data. As such, the functional form of kernel one is unsuitable for the system. Furthermore, the large differences in kernel parameter estimates between states and kernels show the importance of considering different kernels and accounting for spatial heterogeneity when modelling the distance dependence, especially since animal shipments can play an important role in spreading infectious diseases [3].

The predictions of shipments given by Lindström *et al.* [17] have been used to investigate efficiency of potential surveillance strategies of bTB [4,44] and control strategies of FMD [18]. Here, we show that the functional form used by Lindström *et al.* is the least preferred. This raises an important question: are conclusions based on previous shipment predictions unreliable? Large-scale predictions were shown here to be similar across all kernel functional forms in terms of the proportion of shipment moving intrastate (figure 3) as well as number of incoming and outgoing shipments per state (figure 5). Figure 6 further shows that even though kernels two and three are better predictors of observed shipment distances than kernel one, simulated shipment distances are typically similar. Thus, we argue that studies based on previous versions are not invalid. However, the improvements made in this study are substantial, and we advise future studies that rely on estimates of US cattle shipments to use these updated predictions. For this purpose, we provide 1000 realizations of shipments with kernel three in the electronic supplementary material.

To further inform the choice of kernel functional form, we included expert opinion data in the analysis for states where these were available. Expert elicitation questions must be formulated so that they are clear to all the targeted experts. In the survey used here, questions were designed to promote inclusion of experts with varying levels of statistical expertise, and it was not possible to elicit information about the four state-level parameters (rate $\lambda_S$, propensity to attract shipments $W_S$, scale $d_S$ and shape $R_S$). Instead, we used responses about quantities experts could have an intuitive sense about and recalculated responses into quantities that could be included in the statistical model, specifically the proportion of shipments moving interstate. Expert responses have previously been used successfully in Bayesian modelling (e.g. [22–24]). The inclusion of experts in our analysis did, however, not alter the ranking of kernel functional forms (table 1) and had minimal effect on estimates (figures 7 and 8). The reason is a mixture of two properties of the data. First, the expert responses were disparate, leading to low precision ($\xi_s$) of the logit-normal distribution modelling expert responses. Consequently, the contribution of the experts to the posterior was low. Secondly, in instances where expert estimates were less disparate, such as for Texas beef shipments, the estimates coincided with the estimates from the ICVI data only. Thus, we can at least conclude that the estimates of the statistical model for ICVI data does not contradict the collective expectations of the experts. The disparate answers from the included experts further demonstrate that intuitive expectations about basic shipment patterns in the USA are challenging, even for insightful practitioners. These findings are important since they show that the inclusion of expert opinion, carefully designed questionnaires and rigorous mathematical models do not serve as a guarantee for enhanced predictions by including experts. This also put emphasis on the need for quantitative approaches such as those presented in this study.

We also investigated the presence of seasonal variation. Overall, patterns were consistent over the year, with parameter estimates largely overlapping (electronic supplementary material, figure S1). Also, considering the median estimates of proportion of interstate shipments, the geographical patterns were highly similar (figure 4). Yet, some variability can be observed, with e.g. quarter four exhibiting a higher proportion of interstate shipments for Montana and Utah and lower proportion for New York compared with previous quarters. Prominent seasonal variability may influence experts' perception of shipment patterns by making it harder to provide an unbiased yearly estimate. It could therefore to some degree explain the disparate estimates offered by experts in Beck-Johnson *et al.* [28] and consequently the low contribution of experts to the estimates of this study. However, given that patterns are largely consistent over the year, it is unlikely to be the major factor. The consistent pattern also suggests that, for large-scale modelling purposes, it is probably often sufficient to rely on annual estimates. Yet, it may have some bearing for individual states, and we here also provide seasonal network simulations such that they may be used in models where seasonality is an important feature. With the improvements to the existing framework, the analysis recaptured essential state-level features of the training data (figures 5, 6 and 8) as well as validation data (figure 9). Figures 6 and 10 further illustrate that USAMM predictions

produce shipment distances comparable to the observed shipments. However, there are instances where the posterior predictive bands do not fully overlap the observed distribution curves, especially for bi- or multi-modal distribution curves. The underlying reason for this behaviour in the observed shipment distances is unclear, but it is probably due to the presence of infrastructure or premises characteristics that promotes a high number of shipments between specific county pairs. We suggest further developments of the modelling framework should aim to identify the industry structures that produce these patterns and thereby improve estimates at the county scale. We also suggest that future studies could focus on identifying deviations from the assumption of a Poisson process in movements, which can effect outbreak dynamics, particularly for rapidly transmitting diseases [45,46].

Nevertheless, we believe the spatially explicit model structure of our model is an appropriate framework to build on. Other methods of inferring links in incomplete networks typically rely on expectations of the nodes [47] or network structure [48,49], neither of which are readily available for the considered system. The spatial component is essential for most topics where estimates of cattle shipments are of interest, such as investigations of surveillance, control strategies and disease preparedness.

The advancement here focused on state-level differences in shipment patterns and revealed substantial differences in shipment patterns across the USA median estimates of per premises rate of shipments ($\lambda$) and propensity to receive shipments ($W$) varied by more than an order of magnitude between states (figure 2), a pattern that was consistent across models and data. Similarly, the kernel parameter varied substantially among states, indicating that a single set of parameters is insufficient to recapture the shipment pattern of a heterogeneous livestock industry. This is consistent with the findings of Brommesson *et al.* [16], who found that parameters of the spatial kernel describing Swedish cattle shipments varied geographically. These considerations are of even greater importance when considering the highly heterogeneous structure of the US cattle industry.

The estimates provided by this study currently offer the most reliable depiction of national-scale cattle shipments in the USA, accounting for state-level differences in the industry structure. For the first time, we were able to validate the predictions with out-of-sample data from additional years and found that the broad predictions fit well with the observations. To make our findings available for the broad range of research that relies on estimates of cattle shipments in the USA, we make realizations of county-level shipment networks available (see electronic supplementary material). We expect these will benefit future agency planning efforts and studies of infectious livestock diseases where shipment of animals constitutes an important route of transmission.

Data accessibility. We make available the code to allow a replication of our study, as well as indicate in the manuscript sources of supporting data (for instance, expert opinion data [28]). In addition to the code, the study required the use of three types of data/input; namely, Interstate Certificate of Veterinary Inspection (ICVI) data, FLAPS model predictions of farm densities based on data from the USDA National Agricultural Statistics Service, and expert opinion estimates. The underlying data for county-level densities is accessible via the NASS homepage, and we share the FLAPS predictions for county-level totals in the cases where NASS has not published data, and the aforementioned expert opinion data. These come from a previously published survey study with experts from several US states. We use all respondent data, but do not disclose individual expert responses for states with two or less respondents. This is because the premise of the survey was that individual experts' responses would not be disclosed. To the extent that they can be shared, data are already available. The ICVI datasets analysed during this study are not publicly available. Requests for these data may be available from USDA authors, Dr Katie Portacci (Katie.Portacci@aphis.usda.gov) and Dr Ryan Miller (Ryan.S.Miller@aphis.usda.gov) upon reasonable request, in compliance with Federal regulations, and under agreements with the US Department of Agriculture.
Authors' contributions. Conceived and designed the experiments: P.B., C.T.W. and T.L. Collection and curation of data: L.B.-J., C.H., D.M., R.S.M. and K.P. Implementation: P.B. and S.S. Visualization: P.B. and C.H. Analysis: P.B., S.S., L.B.-J., C.H., D.M., R.S.M., C.T.W. and T.L. All authors contributed in writing the paper and gave final approval for publication.
Competing interests. We declare we have no competing interests.
Funding. This work was supported by funding provided by the US Department of Homeland Security Science and Technology Directorate under contract no. HSHQDC-13-C-B0028.
Acknowledgements. All analyses were performed on resources provided by the Swedish National Infrastructure for Computing (SNIC).

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
