## [Peer Review File · Royal Society Open Science]

Review History

RSOS-192042.R0 (Original submission)

Review form: Reviewer 1

Is the manuscript scientifically sound in its present form?

Yes

Are the interpretations and conclusions justified by the results?

Yes

Is the language acceptable?

No

Do you have any ethical concerns with this paper?

No

Have you any concerns about statistical analyses in this paper?

No

Recommendation?

Major revision is needed (please make suggestions in comments)

Comments to the Author(s)

I found this manuscript very difficult to read. This is in part because the statistical model is quite complicated, but I think much more could be done to help the reader.

At a trivial level, there are a few notational errors or inconsistencies that need to be fixed. With a model as complicated as this, the notation should be carefully designed to assist understanding; having the same symbol used for different things is a bad idea (for example, ω is used for two quite different things), and notational errors are confusing and frustrating.

At a more structural level, my view is that the paper needs some fundamental restructuring to make it easier on the reader.

While having a table like Table 1 to define notation is a good idea, putting it up front rather than in an appendix is a bad idea in my view. Without knowing the model intimately, the reader can't possibly absorb everything in the table. By the time I had been through the table I was a bit dazed, knowing I had not absorbed all of it, and wondering how I was possibly going to follow what came next without holding the whole table in my head.

Here's a suggestion for an alternative structure, in which the central components of the model are presented to give a broad-brush picture of the model, followed by the details of each component later, by which time the reader has the big picture and the context:

1. Put Eqn (2.2) front and centre, at the start of the model description. Say what each of its components is but don't give the detail of $p_{\{\Delta | \omega\}}$, just forward-reference the subsection that gives this detail (which should also contain details of the kernels, since these appear only as a component of $p_{\{\Delta | \omega\}}$).
2. Replace lines 50-60 on page 9, and lines 1-16 on page 10 with the statement "We factorise Eqn (2.2) into the probability of total shipments $|T^{\{\tau\}}_{\omega}|$ leaving county ω , and the probabilities of the observed incoming shipments at each destination county outside the state, conditional on this total shipment". Then give the maths of Eqn (2.4) - but try to do it in a simpler way! Then just say that the probability for all days and all origin counties is just the product over counties and days of this probability. I struggle to see the purpose of Eqn (2.5) beyond this simple statement, and as it stands (2.5) and all the text and math around it just muddies the waters (even if it is mathematically rigorous - if you really want it, put it in an appendix).

The above then specifies the model for the observed data, in a much simpler and much more understandable way than currently.

3. Put all the detail of the hierarchical Bayes model (in particular, lines 20 to 43 on page 12) into an appendix.
4. Put as much of section J "Computation" as possible into an appendix.

Finally, the writing style is really poor throughout. Sentences are often too long, there are too many parenthetical clauses, and a mixture of mathematics and English that makes them very difficult to read. Use shorter, simpler sentences and separate the maths from the text. To give a random example, you say this :

"we modeled the total probability of observing $|T^{\{\tau\}}_{\omega\Delta}|$ number of interstate shipments from county $\omega \in E_S$ to all possible interstate destination counties

$\Delta_{inC \setminus E \setminus S}$ (i.e. all contiguous counties not in S) on day τ , as a product of their individual probabilities."

This is shorter, simpler and clearer:

"we modeled the probability of the total observed interstate shipments on a day, from one county to any other outside its state, as a product of their individual probabilities." Then give the maths.

Much easier to follow.

There are a handful of minor comments on notation and such, in the attached pdf file (Appendix A).

Review form: Reviewer 2

Is the manuscript scientifically sound in its present form?

Yes

Are the interpretations and conclusions justified by the results?

Yes

Is the language acceptable?

Yes

Do you have any ethical concerns with this paper?

No

Have you any concerns about statistical analyses in this paper?

No

Recommendation?

Accept with minor revision (please list in comments)

Comments to the Author(s)

The combination of elicited expert opinion and quantitative data is an interesting and potentially very useful approach to reconstructing networks. I will not profess high levels of expertise in understanding how to interpret expert opinion data such as these, but the approach appears sensible. It is of course impossible to directly evaluate the computational aspects of the model but the approach to obtaining the posteriors is sensible and careful and would seem to be appropriate.

I have a few general comments. First of all, the authors largely seem to ignore the temporal aspects of the livestock network, in particular whether or not there is variation in the tempo particularly when comparing across states. The assumption of Poisson distributed frequencies may be poor, and this is likely to be a critical issue for the transmission of some disease, in particular where disease transmission is close to the threshold for persistence. It would be useful to be given some confidence that temporal variation (be it weekly, monthly or seasonal) will not impact the predicted results.

Examining the maps comparing the distribution (by state) of proportions of interstate inward and outward movements (Figures 3 and 4) there seems to be little difference amongst the 3 kernels, and it is difficult to discern how much is being added to the information by the processes generated the 3 fitted outcomes, as there is no comparison to a null model. This comparison is essential to determining the value of the fit. I would suggest taking a very simple, easy to

calculate model as a point of comparison to determine how much value is being added by the authors' methods.

Also the expert opinion information seems to add little to the model. For example in terms of the model selection in Table 2, or predicting the proportion of interstate movements in Fig. 6, or in Fig 7. The authors are transparent about this and the reasons why. However, the paper is written as if this inclusion is a major aim of the study, and yet there is little here to validate the approach they have developed. It would be helpful to have some better evidence that the methods they have proposed are really worth the effort (which appears to be considerable). At the very least some insight as to what conditions would one expect these opinions to matter would be very useful to have.

In figure 9, there are some strong bi-modalities in the CVI data for some states, which are not present in the USAMM predictions - this could do with some explanation.

Decision letter (RSOS-192042.R0)

09-Mar-2020

Dear Mr Brommesson,

The editors assigned to your paper ("Assessing intrastate cattle shipments from interstate data and expert opinion") have now received comments from reviewers. We would like you to revise your paper in accordance with the referee and Associate Editor suggestions which can be found below (not including confidential reports to the Editor). Please note this decision does not guarantee eventual acceptance.

Please submit a copy of your revised paper before 01-Apr-2020. Please note that the revision deadline will expire at 00.00am on this date. If we do not hear from you within this time then it will be assumed that the paper has been withdrawn. In exceptional circumstances, extensions may be possible if agreed with the Editorial Office in advance. We do not allow multiple rounds of revision so we urge you to make every effort to fully address all of the comments at this stage. If deemed necessary by the Editors, your manuscript will be sent back to one or more of the original reviewers for assessment. If the original reviewers are not available, we may invite new reviewers.

- Data accessibility

<http://datadryad.org/submit?journalID=RSOS&manu=RSOS-192042>

- Competing interests

- Authors' contributions

- Acknowledgements

- Funding statement

on behalf of the Associate Editor, and Professor Kevin Padian (Subject Editor)
 openscience@royalsociety.org

Associate Editor's comments:

Thank you for submitting this piece to RSOS. We have received two thorough reviewer reports, with both reports detailing many suggestions and comments which need to be addressed in your revision. It would be very helpful if you could please provide a clean version, and a separate tracked changes version of your manuscript within your revised submission. We appreciate that the comments regarding restructuring your manuscript may take additional time, so please contact the editorial office if you require an extension. Best of luck with your revisions

Reviewers' Comments to Author:

Reviewer: 1
 Comments to the Author(s)

I found this manuscript very difficult to read. This is in part because the statistical model is quite complicated, but I think much more could be done to help the reader.

At a trivial level, there are a few notational errors or inconsistencies that need to be fixed. With a model as complicated as this, the notation should be carefully designed to assist understanding; having the same symbol used for different things is a bad idea (for example, ω is used for two quite different things), and notational errors are confusing and frustrating.

At a more structural level, my view is that the paper needs some fundamental restructuring to make it easier on the reader.

While having a table like Table 1 to define notation is a good idea, putting it up front rather than in an appendix is a bad idea in my view. Without knowing the model intimately, the reader can't possibly absorb everything in the table. By the time I had been through the table I was a bit dazed, knowing I had not absorbed all of it, and wondering how I was possibly going to follow what came next without holding the whole table in my head.

Here's a suggestion for an alternative structure, in which the central components of the model are presented to give a broad-brush picture of the model, followed by the details of each component later, by which time the reader has the big picture and the context:

1. Put Eqn (2.2) front and centre, at the start of the model description. Say what each of its components is but don't give the detail of $p_{\{\delta | \omega\}}$, just forward-reference the subsection that gives this detail (which should also contain details of the kernels, since these appear only as a component of $p_{\{\delta | \omega\}}$).

2. Replace lines 50-60 on page 9, and lines 1-16 on page 10 with the statement "We factorise Eqn (2.2) into the probability of total shipments $|T\&\{\tau\}_\omega|$ leaving county ω , and

the probabilities of the observed incoming shipments at each destination county outside the state, conditional on this total shipment". Then give the maths of Eqn (2.4) - but try to do it in a simpler way! Then just say that the probability for all days and all origin counties is just the product over counties and days of this probability. I struggle to see the purpose of Eqn (2.5) beyond this simple statement, and as it stands (2.5) and all the text and math around it just muddies the waters (even if it is mathematically rigorous - if you really want it, put it in an appendix).

The above then specifies the model for the observed data, in a much simpler and much more understandable way than currently.

3. Put all the detail of the hierarchical Bayes model (in particular, lines 20 to 43 on page 12) into an appendix.

4. Put as much of section J "Computation" as possible into an appendix.

Finally, the writing style is really poor throughout. Sentences are often too long, there are too many parenthetical clauses, and a mixture of mathematics and English that makes them very difficult to read. Use shorter, simpler sentences and separate the maths from the text. To give a random example, you say this :

"we modeled the total probability of observing $|T^{\tau}_{\omega\delta}|$ number of interstate shipments from county $\omega \in E_S$ to all possible interstate destination counties $\delta \in C \setminus E_S$ (i.e. all contiguous counties not in S) on day τ , as a product of their individual probabilities."

This is shorter, simpler and clearer:

"we modeled the probability of the total observed interstate shipments on a day, from one county to any other outside its state, as a product of their individual probabilities." Then give the maths. Much easier to follow.

There are a handful of minor comments on notation and such, in the attached pdf file.

Reviewer: 2

Comments to the Author(s)

The combination of elicited expert opinion and quantitative data is an interesting and potentially very useful approach to reconstructing networks. I will not profess high levels of expertise in understanding how to interpret expert opinion data such as these, but the approach appears sensible. It is of course impossible to directly evaluate the computational aspects of the model but the approach to obtaining the posteriors is sensible and careful and would seem to be appropriate.

I have a few general comments. First of all, the authors largely seem to ignore the temporal aspects of the livestock network, in particular whether or not there is variation in the tempo particularly when comparing across states. The assumption of Poisson distributed frequencies may be poor, and this is likely to be a critical issue for the transmission of some disease, in particular where disease transmission is close to the threshold for persistence. It would be useful to be given some confidence that temporal variation (be it weekly, monthly or seasonal) will not impact the predicted results.

Examining the maps comparing the distribution (by state) of proportions of interstate inward and outward movements (Figures 3 and 4) there seems to be little difference amongst the 3 kernels, and it is difficult to discern how much is being added to the information by the processes generated the 3 fitted outcomes, as there is no comparison to a null model. This comparison is

essential to determining the value of the fit. I would suggest taking a very simple, easy to calculate model as a point of comparison to determine how much value is being added by the authors' methods.

Also the expert opinion information seems to add little to the model. For example in terms of the model selection in Table 2, or predicting the proportion of interstate movements in Fig. 6, or in Fig 7. The authors are transparent about this and the reasons why. However, the paper is written as if this inclusion is a major aim of the study, and yet there is little here to validate the approach they have developed. It would be helpful to have some better evidence that the methods they have proposed are really worth the effort (which appears to be considerable). At the very least some insight as to what conditions would one expect these opinions to matter would be very useful to have.

In figure 9, there are some strong bi-modalities in the CVI data for some states, which are not present in the USAMM predictions - this could do with some explanation.

Author's Response to Decision Letter for (RSOS-192042.R0)

See Appendix B.

RSOS-192042.R1 (Revision)

Review form: Reviewer 2

Is the manuscript scientifically sound in its present form?

Yes

Are the interpretations and conclusions justified by the results?

Yes

Is the language acceptable?

Yes

Do you have any ethical concerns with this paper?

No

Have you any concerns about statistical analyses in this paper?

No

Recommendation?

Accept with minor revision (please list in comments)

Comments to the Author(s)

The authors have largely addressed my comments and those of the other reviewer. I would note that they have misunderstood my query regarding temporality of movements. While it is true that seasonality may play a role, and the additional analysis presented is useful, I was referring to the role that the Poisson distribution may play in allowing for the transmission of short infectious

period diseases. Two papers that examine this question are Nickbakhsh et al (doi: 10.1016/j.epidem.2013.03.001), and Colman et al. (doi: 10.1186/s12879-018-3117-6). For slowly transmitting diseases this won't matter, but a Poisson distribution assumption is likely to make a rapidly transmitting disease spread more easily - therefore influencing the epidemic thresholds. I don't think its necessary that the authors do more work to address this in this manuscript - they have done plenty for a single publication already. It is worth noting though, that the assumption they've made (and given that livestock movements are a commercial enterprise, they almost certainly have much more regularly scheduled tempos) could have an influence on epidemic thresholds.

A minor point (which I did not pick up on before) is that the authors state on line 55 onwards of page 23 that "Further, kernel one estimates of scale parameter d included values below $10E-7$ km (Figure 2). These are exceptionally short distances considering the large spatial scales of the U.S." This is true - and indeed is very short on any national scale. However, in practice, one would imagine that anyone using the model to assign movements to farms would simply devise some method to handle this - e.g. assign all probability below a certain threshold to either a non-movement, or to a local movement to a near neighbour. The important question is whether or not the probability of moving these very short distances (once the issue has been handled by some such method) seriously impacts the predictive power of the model in terms of disease transmission - if this is the case then the authors are correct that it detracts from the model in a practical sense - but otherwise it just necessitates an approximation.

Decision letter (RSOS-192042.R1)

Dear Mr Brommesson

On behalf of the Editors, we are pleased to inform you that your Manuscript RSOS-192042.R1 "Assessing intrastate cattle shipments from interstate data and expert opinion" has been accepted for publication in Royal Society Open Science subject to minor revision in accordance with the referees' reports. Please find the referees' comments along with any feedback from the Editors below my signature.

Please submit your revised manuscript and required files (see below) no later than 7 days from today's (ie 25-Jan-2021) date. Note: the ScholarOne system will 'lock' if submission of the revision is attempted 7 or more days after the deadline. If you do not think you will be able to meet this deadline please contact the editorial office immediately.

Please note article processing charges apply to papers accepted for publication in Royal Society Open Science (<https://royalsocietypublishing.org/rsos/charges>). Charges will also apply to papers transferred to the journal from other Royal Society Publishing journals, as well as papers submitted as part of our collaboration with the Royal Society of Chemistry

(<https://royalsocietypublishing.org/rsos/chemistry>). Fee waivers are available but must be requested when you submit your revision (<https://royalsocietypublishing.org/rsos/waivers>).

on behalf of the Associate Editor and Professor Kevin Padian (Subject Editor)
openscience@royalsociety.org

Associate Editor Comments to Author:

Please ensure that you address the referee's remaining minor comments accordingly.

Reviewer comments to Author:

Reviewer: 2

Comments to the Author(s)

The authors have largely addressed my comments and those of the other reviewer. I would note that they have misunderstood my query regarding temporality of movements. While it is true that seasonality may play a role, and the additional analysis presented is useful, I was referring to the role that the Poisson distribution may play in allowing for the transmission of short infectious period diseases. Two papers that examine this question are Nickbakhsh et al (doi: 10.1016/j.epidem.2013.03.001), and Colman et al. (doi: 10.1186/s12879-018-3117-6). For slowly transmitting diseases this won't matter, but a Poisson distribution assumption is likely to make a rapidly transmitting disease spread more easily - therefore influencing the epidemic thresholds. I don't think its necessary that the authors do more work to address this in this manuscript - they have done plenty for a single publication already. It is worth noting though, that the assumption they've made (and given that livestock movements are a commercial enterprise, they almost certainly have much more regularly scheduled tempos) could have an influence on epidemic thresholds.

A minor point (which I did not pick up on before) is that the authors state on line 55 onwards of page 23 that "Further, kernel one estimates of scale parameter d included values below $10E-7$ km (Figure 2). These are exceptionally short distances considering the large spatial scales of the U.S." This is true - and indeed is very short on any national scale. However, in practice, one would imagine that anyone using the model to assign movements to farms would simply devise some method to handle this - e.g. assign all probability below a certain threshold to either a non-movement, or to a local movement to a near neighbour. The important question is whether or not the probability of moving these very short distances (once the issue has been handled by some such method) seriously impacts the predictive power of the model in terms of disease transmission - if this is the case then the authors are correct that it detracts from the model in a practical sense - but otherwise it just necessitates an approximation.

===PREPARING YOUR MANUSCRIPT===

===PREPARING YOUR REVISION IN SCHOLARONE===

- Any electronic supplementary material (ESM).
- If you are requesting a discretionary waiver for the article processing charge, the waiver form must be included at this step.
- If you are providing image files for potential cover images, please upload these at this step, and inform the editorial office you have done so. You must hold the copyright to any image provided.
- A copy of your point-by-point response to referees and Editors. This will expedite the preparation of your proof.

- Ensure that your data access statement meets the requirements at <https://royalsociety.org/journals/authors/author-guidelines/#data>. You should ensure that you cite the dataset in your reference list. If you have deposited data etc in the Dryad repository, please only include the 'For publication' link at this stage. You should remove the 'For review' link.
- If you are requesting an article processing charge waiver, you must select the relevant waiver option (if requesting a discretionary waiver, the form should have been uploaded at Step 3 'File upload' above).
- If you have uploaded ESM files, please ensure you follow the guidance at <https://royalsociety.org/journals/authors/author-guidelines/#supplementary-material> to include a suitable title and informative caption. An example of appropriate titling and captioning may be found at https://figshare.com/articles/Table_S2_from_Is_there_a_trade-off_between_peak_performance_and_performance_breadth_across_temperatures_for_aerobic_scope_in_teleost_fishes_/3843624.

Author's Response to Decision Letter for (RSOS-192042.R1)

See Appendix C.

Decision letter (RSOS-192042.R2)

Dear Mr Brommesson,

It is a pleasure to accept your manuscript entitled "Assessing intrastate cattle shipments from interstate data and expert opinion" in its current form for publication in Royal Society Open Science.

You can expect to receive a proof of your article in the near future. Please contact the editorial office (openscience@royalsociety.org) and the production office (openscience_proofs@royalsociety.org) to let us know if you are likely to be away from e-mail

contact – if you are going to be away, please nominate a co-author (if available) to manage the proofing process, and ensure they are copied into your email to the journal.

on behalf of Prof Kevin Padian (Subject Editor)
openscience@royalsociety.org

Appendix A**ROYAL SOCIETY
OPEN SCIENCE****Assessing intrastate cattle shipments from interstate data
and expert opinion**

Journal:	Royal Society Open Science
Manuscript ID	RSOS-192042
Article Type:	Research
Date Submitted by the Author:	22-Nov-2019
Complete List of Authors:	Brommesson, Peter; Linköping University, Department of Physics, Chemistry and Biology Sellman, Stefan; Linköping University, Department of Physics, Chemistry and Biology Beck-Johnson, Lindsay; Colorado State University, Biology Hallman, Clayton N; Colorado State University, Biology Murieta, Deedra; Colorado State University, Biology Webb, Colleen; Colorado State University, Biology Miller, Ryan; United States Department of Agriculture, Centers for Epidemiology and Animal Health Portacci, Katie; United States Department of Agriculture, Centers for Epidemiology and Animal Health Lindström, Tom; Linköping University, Department of Physics, Chemistry and Biology
Subject:	health and disease and epidemiology < BIOLOGY
Keywords:	Livestock, Spread of disease, Cattle shipment, Movement network, Expert data
Subject Category:	Biology (whole organism)

Author-supplied statements

Relevant information will appear here if provided.

Ethics

Does your article include research that required ethical approval or permits?:

This article does not present research with ethical considerations

Statement (if applicable):

CUST_IF_YES_ETHICS :No data available.

Data

It is a condition of publication that data, code and materials supporting your paper are made publicly available. Does your paper present new data?:

Yes

Statement (if applicable):

We make available the code to allow a replication of our study, as well as indicate in the manuscript sources of supporting data (for instance, expert opinion (Beck-Johnson et al. 2019).

In addition to the code, the study required the use of three types of data/input; namely, Interstate Certificate of Veterinary Inspection (ICVI) data, FLAPS model predictions of farm densities based on data from the USDA National Agricultural Statistics Service, and expert opinion estimates.

The underlying data for county-level densities is accessible via the NASS homepage, and we share the FLAPS predictions for county-level totals in the cases where NASS has not published data, and the aforementioned expert opinion data.

These come from a previously published survey study with experts from several US states.

We use all respondent data, but do not disclose individual expert responses for states with two or less respondents.

This is because the premise of the survey was that individual experts' responses wouldn't be disclosed.

To the extent that they can be shared, data is already available.

The ICVI datasets analyzed during this study are not publically available.

Requests for these data may be available from USDA authors, Dr. Katie Portacci (Katie.Portacci@aphis.usda.gov) and Dr. Ryan Miller (Ryan.S.Miller@aphis.usda.gov) upon reasonable request, in compliance with Federal regulations, and under agreements with the United States Department of Agriculture

Conflict of interest

I/We declare we have no competing interests

Statement (if applicable):

CUST_STATE_CONFLICT :No data available.

Authors' contributions

This paper has multiple authors and our individual contributions were as below

Statement (if applicable):

1
2
3 Conceived and designed the experiments: P.B, C.T.W., T.L.
4 Collection and curation of data: L.B.-J., C.H, D.M., R.S.M, K.P.
5 Implementation: P.B, S.S.
6 Visualization: P.B., C.H.
7 Analysis: P.B, S.S, L.B.-J., C.H., D.M., R.S.M, C.T.W., T.L.
8 All authors contributed in writing the paper and gave final approval for publication.
9
10
11
12
13
14
15
16
17
18
19
20
21
22
23
24
25
26
27
28
29
30
31
32
33
34
35
36
37
38
39
40
41
42
43
44
45
46
47
48
49
50
51
52
53
54
55
56
57
58
59
60

1
2
3
4
5
6
7
8
9
10
11
12
13
14
15
16
17
18
19
20
21
22
23
24
25
26
27
28
29
30
31
32
33
34
35
36
37
38
39
40
41
42
43
44
45
46
47
48
49
50
51
52
53
54
55
56
57
58
59
60

**ROYAL SOCIETY
OPEN SCIENCE**

rsos.royalsocietypublishing.org

Research

Article submitted to journal

Subject Areas:

Theoretical biology

Keywords:

Livestock, Spread of disease, Cattle shipment, Movement network, Expert data

Author for correspondence:

Tom Lindström

e-mail: tom.lindstrom@liu.se

Assessing intrastate shipments from interstate data and expert opinion.

Peter Brommesson¹, Stefan Sellman¹

Lindsay Beck-Johnson², Clayton Hallman²,

Deedra Murrieta², Colleen T Webb², Ryan

S Miller³, Katie Portacci³, Tom Lindström¹

¹Department of Physics, Chemistry and Biology, Division of Theoretical Biology, Linköping University, 58183 Linköping, Sweden.

²Department of Biology, Colorado State University, Fort Collins, CO 80523

³Center for Epidemiology and Animal Health, United States Department of Agriculture-Veterinary Services, Fort Collins, CO 80526

Live animal shipments are a potential route for transmitting animal diseases between holdings and are crucial when modeling spread of infectious diseases. Yet, complete contact networks are not available in all countries, including the United States. Here, we considered a 10% sample of Interstate Certificate of Veterinary Inspections from one year (2009). We focused on distance dependence in contacts and investigated how different functional forms affect estimates of unobserved intrastate shipments. To further enhance our predictions, we included responses from an expert elicitation survey about the proportion of shipments moving intrastate. We used hierarchical Bayesian modeling to estimate parameters describing the kernel and effects of expert data. We considered three functional forms of spatial kernels and the inclusion or exclusion of expert data. The resulting six models were ranked by WAIC and DIC and evaluated through within- and out-of-sample validation. We showed that predictions of intrastate shipments were mildly influenced by the functional form of the spatial kernel but kernel shapes that permitted a fat tail at large distances while maintaining a plateau shaped behavior at short distances better were preferred. Further, our study showed that expert data may not guarantee enhanced predictions when expert estimate are disparate.

© 2014 The Authors. Published by the Royal Society under the terms of the Creative Commons Attribution License <http://creativecommons.org/licenses/by/4.0/>, which permits unrestricted use, provided the original author and source are credited.

**THE ROYAL SOCIETY
PUBLISHING**

1. Introduction

Transboundary animal diseases (TADs) pose a global threat to food security, and outbreak events are a major concern for animal health. Outbreaks may cause national emergencies, with huge costs to the livestock sector due to disruption of production and export. As such, TADs are a primary concern for food security [1]. Additionally, several TADs have painful symptoms, making them an animal welfare concern. In the face of an outbreak, policy makers must often make high stakes decisions with limited information. Therefore, outbreak preparedness is important for facilitating a swift and efficient response, which is essential to disease control [2]. A detailed understanding of the potential contacts between premises that could mediate transmission in a potential outbreak is an important aspect of outbreak preparedness.

Animal shipments between farms and other agricultural premises are of particular concern for disease spread because of their potential to introduce infected animals into susceptible herds or flocks [3]. Contacts often occur over large distances and can precipitate geographically widespread epidemics [4]. For example, nine of the twelve spatial clusters of the 2001 Foot and Mouth Disease (FMD) outbreak in the United Kingdom (UK) were initiated by live animal shipments [5]. Early detection of shipment contacts may promote a reduced outbreak.

Cattle production is an important part of the United States (U.S.) meat animal industry accounting for US\$67.3 billion in total production value year 2017 [6] and is second only to poultry in the total pounds of product produced [7]. The U.S. cattle industry accounts for approximately 900,000 premises and 103 million animals [8,9]. An outbreak of a TAD such as FMD is expected to have severe impact on the U.S. economy, with economic losses predicted to be at least US\$14 billion in the first year of an outbreak as a result of control and disruption to international trade. This number corresponds to 9.5% of the U.S. farm income and for the live cattle and beef meat sector the losses in gross revenues were estimated at 17% and 20%, respectively [10]. Economic losses due to spread of disease among cattle are not unique to TADs. Attempts to control Bovine Tuberculosis (bTB), which is endemic in Michigan, costed \$200 million over 15 years [11].

Shipment restrictions that minimize the risk of pathogen spread without interrupting production are essential for cost efficient disease control. However, modeling efforts to investigate the efficiency of control options are challenged by frequently limited data on locations, sizes and types of premises, in particular in Africa and South-East Asia [12]. In contrast, countries of the European Union are legally required to collect and store data of all live animal shipments [13]. In the U.S. premises-level data describing location, premises type, and animal inventory is not uniformly collected for domestic animal industries due to stakeholder concerns regarding cost, confidentiality, and security of collected information [14]. Survey data has previously been used to identify interstate shipping patterns using Interstate Certificates of Veterinary Inspection (ICVI) issued by animal health authorities when animals are shipped across state boundaries [15]. However, shipping patterns in countries with complete data typically show high frequency of short distance shipments [16], and a similar pattern in the U.S. would result in a large number of intrastate shipments. As such, there is a need for methods to extrapolate from existing data to predict complete shipment patterns. Lindström et al. [17] proposed an approach to address this need using a kernel function to model distance dependence. These Bayesian predictions have been used for FMD outbreak modeling [18] and making recommendations for bTB surveillance [4]. However, there is a need for more detailed methods, particularly regarding the potential sensitivity of within-state shipments to the choice of kernel function. Therefore, there is a need for investigating different kernels to understand their impact on the predictions.

Empirical data is not the only source of information to inform models. Other sources, such as the knowledge of experts can be used instead when developing and using models, e.g. for estimating presence and risk of infection of animal diseases [19,20]. Expert data can also facilitate prior elicitation for sought parameters [21] and have been used in addition to empirical data, for instance in ecology [22–24]. Further, expert data can be used to inform within-state shipments in the U.S. Here, state veterinarians, cooperative extension professors and other experts with

extensive knowledge about the U.S. cattle industry may offer important insight. This extra information, can be used together with empirical data to provide better understanding of cattle shipment patterns.

The Bayesian paradigm is ideally suited to incorporate expert opinion when these can be expressed as prior distributions. However, there are several issues when converting the answers of expert surveys into priors. Answers might not be expressed explicitly for the parameters we wish to elicit priors for, particularly when the survey is not tailored specifically for the statistical model. Expert data may also be missing if there is a lack of expertise in some geographical areas or some targeted experts choose not to participate. Further, expert answers are typically provided as point estimates rather than a range. Instead of using expert data to inform priors, we propose a statistical model whereby expert answers are treated as data in a hierarchical Bayesian framework.

Based on ICVI and expert data, we developed a modeling framework for continental scale cattle shipments and investigated three functional forms of the spatial kernel, taking into account that the livestock production system varies across the U.S. For instance, feedlots are primarily located in the central states, whereas dairy production is most intense in coastal states such as California and New York and in the Midwest [9]. We therefore propose a model, denoted the United States Animal Movement Model (USAMM), that accounts for the heterogeneity of the system. Our aims are to (i) estimate the proportion of intrastate shipments for each state, (ii) clarify how sensitive this estimate is to the choice of kernel function, (iii) improve estimates of shipment rates across the U.S. at state and county levels, and (iv) investigate the value of including expert data in the analysis.

2. Materials and Methods

(a) Overview

We modeled beef and dairy shipments as two separate networks consisting of all counties of the contiguous U.S. (i.e. excluding Hawaii and Alaska) as nodes and shipments between them as links. Models were parameterized from shipment data extracted from Interstate Certificate of Veterinary Inspections (ICVIs) and county information, such as county centroid coordinates and number of animals in the county. Further, we incorporated expert estimates of the proportion of interstate shipments to enhance our model.

We used Bayesian modeling to estimate parameters describing the underlying processes that produce the observed networks. We also performed model selection to investigate which candidate model better fits the data and implemented a series of model validations to determine predictive accuracy. Finally, we conducted sensitivity analysis to investigate the robustness of our model.

(b) Data

The shipment data used in this study consists of ICVI records, which are official documents issued by an accredited veterinarian or an official state or federal veterinarian. The primary purpose of ICVIs is to prevent potential disease spread by ensuring that shipped animals are apparently healthy and show no visible signs of communicable disease. ICVIs are also used as one source of information to support traceability of animals in the event of a disease outbreak [25]. ICVIs are required for most interstate shipments, except for shipments going directly to slaughter, and contains information that include origin, destination, date of the shipment and characteristics of the shipped animals.

We used the ICVI dataset from 2009 which is described in detail in Buhnerkempe et al. [15]. It consists of systematic 10% samples, one from each set of ICVIs, provided by all origin states of the contiguous U.S. except for New Jersey (did not participate). ICVIs were classified as beef or dairy by a classification tree analysis [15], and the data contained after curation 15,725 and 2,814

beef and dairy shipments, respectively. The number of premises per county were estimated as the mean number of premises from ten realizations of the cattle version of the Farm Location and Animal Population Simulator (FLAPS) [26]. FLAPS disaggregates county level National Agricultural Statistics Service (NASS) estimates of the number of premises in each county in 2012. The mismatch with the ICVI data (which is from 2009) is a caveat, however the total number of cattle premises varied by only 6.4% from 2007 to 2017 [27] and the benefit of imputed premises obtained from FLAPS outweighs potential effects of demographic changes over time. Out of 3,108 contiguous counties, 3,046 and 2,499 counties had at least one premises with production of beef and dairy, respectively. Though FLAPS predicts exact locations of premises within counties, these are different for each realization and there is no data available to connect ICVI data to characteristics of specific premises. Thus, we are confined to county-level modeling of shipments.

For eight states (California, Iowa, Minnesota, New York, North Carolina, Tennessee, Texas, and Wisconsin), we also had access to ICVI data for 2010 and 2011, which we used for validation. For a detailed description of these data, see Gorsich et al. [4].

We also used responses from an expert elicitation survey by Beck-Johnson et al. [28]. Here, experts across multiple U.S. states were asked about the number of shipments that cross state borders. Questions in the survey were commodity specific (beef or dairy) and named specific origin or destination premises types; the specificity in the survey allowed for inference about different cattle industry sectors and about the proportion of interstate shipments at a state level. The expert elicitation survey provided data for 17 contiguous U.S. states including, California, Colorado, Idaho, Iowa, Minnesota, Montana, Mississippi, Nebraska, Nevada, New York, North Carolina, Oklahoma, Pennsylvania, Tennessee, Texas, Virginia and Wisconsin. The data from specific survey questions were selected so that the survey data was compatible with the information that is captured in the ICVI data. Specifically, all the survey questions that were used in this study asked about shipments originating at herds of specific sizes or at different types of premises (i.e. market, seed-stock operation) (see survey questions 7a-d, 8a-d, f, and 12a-d for beef and 15a-c, 16a-c, 18a-b, and 19a-c in Beck-Johnson et al. [28]). These survey questions were selected because they dealt with the origin of shipments just as the ICVI data used in USAMM do. Expert survey questions regarding destination premises type and those dealing with slaughter shipments were excluded from this study because the ICVI data used in USAMM is origin data and does not include slaughter shipments. Individual expert survey data was provided for the selected questions and was processed according to the methods described in Beck-Johnson et al. [28]. The question-level expert estimates were then combined into commodity-specific, state-level estimates for each individual expert by taking the mean over the expert responses to the selected survey questions. Two of the states in this data set, Montana and Mississippi, did not have expert data in one of the commodity types, so the data for these states only includes one commodity. Specifically, Montana data was available for beef but not dairy, and Mississippi data was available for dairy but not beef. For a detailed description of the survey and results, see Beck-Johnson et al. [28].

(c) Model definition

We defined a statistical model for the probability of observing a set of shipments \mathbf{T} , here the shipments in the ICVI data. Information about individual premises was not available from the data, and we focused on county level prediction, yet defined the model structure based on expectations about premises. We assumed that shipments from each premises in state $S \in \mathbf{U}$, where \mathbf{U} denotes the set of contiguous U.S. states, arise by a Poisson process with state specific rate λ_s (shipment \cdot day⁻¹). Consequently the rate of shipments originating from all n_ω premises in county $\omega \in \mathbf{E}_s$, where \mathbf{E}_s denotes the set of counties in state S , is n_ω .

Given that a shipment originates in county $\omega \in \mathbf{E}_s$, we assumed three factors determine the probability $p_{\delta|\omega}$ of destination county $\delta \in \mathbf{C}$, where \mathbf{C} denotes the set of all counties in contiguous US: the number of premises in δ , the distance $D_{\omega,\delta}$ from $\omega \in \mathbf{E}_s$, and state level differences in infrastructure. As with the origin of shipments, we assumed the probability of destination county

Notation	Interpretation / Note
\mathbf{C}	Set of counties in contiguous U.S.
\mathbf{E}_s	Set of counties in state S .
\mathbf{U}	Set of States in the contiguous U.S.
$\mathbf{T}_{\omega,\delta}$	Set of observed shipments from county ω to δ . ICVI data.
ρ	Expert data.
\mathbf{r}_S	Set of respondents in state S .
$D_{\omega,\delta}$	Distance between centroids of counties ω and δ .
n_δ	Number of premises in county δ .
η_S	Proportion data available for state S . 0% for NJ, 10% otherwise.
d_S	Scale measure, defined as the distance where the kernel takes 50% of its initial value. State specific random variable.
R_S	The ratio between d_s and the distance where the kernel takes 5% of its initial value. State specific random variable.
λ_S	Rate with which a premises in state S sends shipments per day. State specific random variable.
w_δ, W_S	$w_\delta =$ Propensity for a premises in county δ to attract shipments. $w_\delta = W_S, \forall \delta \in S$. State specific random variable. Value is relative to W_S for Missouri which is fixed to one.
\hat{W}	Normalized propensity to attract shipments. $\sum_{S \in \mathbf{U}} \hat{W}_s = 1$
ξ_S	Precision of experts in state S . Random variable estimated for states where expert responses were available.
ν_S	Logit of estimated proportion of interstate shipment for state S . State specific parameter to be estimated in absence of ICVI data.
$\Theta = (\Theta_1, \dots, \Theta_{ \mathbf{U} })$ $\Theta_S = (d_S, R_S, \lambda_S)$ $\mathbf{W} = (W_1, \dots, W_{ \mathbf{U} })$ $\hat{\mathbf{W}} = (\hat{W}_1, \dots, \hat{W}_{ \mathbf{U} })$ $\xi = (\xi_1, \dots)$	Vectorized notation introduced for convenience. ξ only contains parameters for states where expert data are present.
a, b	Kernel parameters. Calculated from d_S and R_S .
K_k	Kernel of functional form k .
\hat{z}_S	Proportion of interstate shipments from state S . Function of d_S, R_S and \mathbf{W} .
$p_{\delta \omega}$	Probability of shipment from county ω having destination δ .
\mathbf{L}	Likelihood.
$\Phi_x(x_S m_x, \kappa_x)$	Prior density function for parameter $x_S \in \{d_S, R_S, \lambda_S, \xi_S\}$.
m_x, κ_x	Mean and coefficient of variation of parameter x_S . Hyperparameters to be estimated.
$\Phi_{\mathbf{W}}(W_S), \Phi_\nu(\nu_S)$	Fix prior density functions for W_S and ν_S , respectively.
$\Psi_m^{(x)}(m_x), \Psi_\kappa^{(x)}(\kappa_x)$	Hyperprior density functions for prior parameters m_x and κ_x .

Table 1. Notation used and its interpretation.

is proportional to the number of premises \hat{n}_δ , which is equal to $n_\delta - 1$ if $\delta = \omega$ and otherwise equal to n_δ . Thereby, we corrected our model to exclude the possibility of a premises shipping animals to itself. Distance dependence was modeled with a spatial kernel K_k , where the subscript k indicates a functional form, as elaborated on in section (d). Finally, we account for state level differences in import rates through parameter $w_\delta = W_S$ for county δ in state S . The probability of δ given $\omega \in \mathbf{E}_s$ for kernel type k is given by

$$p_{\delta|\omega}^{(k)} = \frac{K_k(D_{\omega,\delta}|d_S, R_S)w_\delta\hat{n}_\delta}{\sum_{j \in \mathbf{C}} K_k(D_{\omega,j}|d_S, R_S)w_j\hat{n}_j} \tag{2.1}$$

That is, we normalize the relative probability of a shipment to county δ , conditional on origin county ω , over all possible destination counties. Random variables d_S and R_S are the kernel scale and shape parameters, respectively, as further elaborated on in (d) below. The random variable W_S can be interpreted as how likely a premises in a state is to be the destination of a shipment relative to other premises at the same distance from $\omega \in \mathbf{E}_S$. Because it is a relative measure, we set $W_{Missouri} = 1$ to ensure an identifiable model. The choice of Missouri is arbitrary and does not affect the results, and we choose Missouri because it had a large number of incoming shipments, thereby improving computational efficiency by avoiding a substantial uncertainty in the parameter that all other W_S are referenced against. The ICVI data only include a proportion $\eta_S = 10\%$ of all interstate shipments of one year for all considered states except for New Jersey, where $\eta_S = 0\%$. Thus, defining $\mathbf{T}_{\omega,\delta}^\tau$ as the set of shipment from $\omega \in \mathbf{E}_s$ to δ on day τ , the probability of observing $|\mathbf{T}_{\omega,\delta}^\tau|$ number of shipments is

$$|\mathbf{T}_{\omega,\delta}^{(\tau)}| \sim \text{Poisson} \left(|\mathbf{T}_{\omega,\delta}^{(\tau)}| \left| \hat{\lambda}_S n_\omega p_{\delta|\omega} \right. \right), \tag{2.2}$$

where $\hat{\lambda}_S = \eta_S \lambda_S$. For New Jersey, where $\eta_S = 0\%$, the Poisson distribution is not defined and instead we used a degenerate distribution, where all of the probability mass of the distribution is located at $|\mathbf{T}_{\omega,\delta}^{(\tau)}| = 0$.

(d) Spatial kernels

The shape of the spatial kernel is essential for the focus of this study because the kernel behavior at short distances largely determines the proportion of within state shipments. We implemented three functional forms for the kernel:

$$\begin{aligned} K_1(D_{\omega,\delta}|a, b) &= e^{-\left(\frac{D_{\omega,\delta}}{a}\right)^b} & a, b > 0 \\ K_2(D_{\omega,\delta}|a, b) &= 1 - e^{-\left(\frac{D_{\omega,\delta}}{a}\right)^b} & a > 0, b < 0 \\ K_3(D_{\omega,\delta}|a, b) &= \frac{1}{1 + \left(\frac{D_{\omega,\delta}}{a}\right)^b} & a, b > 0. \end{aligned} \tag{2.3}$$

K_1 has the form of a generalized normal distribution and includes for certain parameters well known distributions such as the Gaussian normal ($b = 2$), Laplace ($b = 1$) and as a special case the uniform distribution as b approaches infinity [29]. This kernel has been used in previous studies of U.S. cattle shipments [17,18]. K_1 has a steep slope at short distances and relatively high kernel values at long distances. As alternatives, we choose kernels of the form that allow for different shapes at both short and long distances (e.g. kernels that allow for plateau shape at short distances but also high kernel values at large distances). Further, these kernels have closed form solutions to a reparameterization that can be used instead of parameters a and b which hold no readily interpretable information. Previous studies have reparameterized K_1 by moment statistics [16, 17], but here we use a different approach for two main reasons. First, these moment statistics are only marginally more informative than a and b since they give us no intuitive understanding of the behavior of the kernels. Secondly, K_2 and K_3 include shapes that lack finite moments and it is therefore not possible to define finite quantities for all possible shapes. Instead, we employ an approach where we define the state specific kernel scale by parameters d_S , defined as the distance

Figure 1. Examples of the three kernels under equivalent parameterizations ($d = 50, R = 5$). Inset panel shows the kernel values for larger distances.

where the kernel has dropped to half its original value, i.e. where a premises is half as likely to be the destination compared to an immediate neighboring premises. The use of half of its initial value is used to give an intuitive understanding of the scale of the kernel. We further define kernel shape R_S as the ratio between d_S and the distance where the kernel value reaches some lower value u , here set to 5%. While the value of u is somewhat arbitrary, it corresponds to a value with substantially lower kernel value compared to half of its value that used to define d_S . This reparameterization makes the kernels easy to visualize and express the kernels on equivalent statistics. Parameter d_S has a distance unit (here kilometers) and R_S is a scale free measure of shape. Here, we considered only monotonously decreasing kernel functions and therefore put the restrictions $d_S \in (0, \infty)$ and $R_S \in (1, \infty)$ on the parameters. Figure 1 shows examples of shapes of the three kernel functions for equivalent parameterization.

Through eq. 2.2, we modeled the total probability of observing $|\mathbf{T}_{\omega, \delta}^{(\tau)}|$ number of interstate shipments from county $\omega \in \mathbf{E}_S$ to all possible interstate destination counties $\delta \in \mathbf{C} \setminus \mathbf{E}_S$ (i.e. all contiguous counties not in S) on day τ , as a product of their individual probabilities. We denoted this probability $P_{\omega}^{(\tau, k)}(\mathbf{T}_{\omega}^{(\tau)} | d_S, R_S, \lambda_S, \mathbf{W})$ where $\mathbf{T}_{\omega}^{(\tau)} = \bigcup_{\delta \in \mathbf{C} \setminus \mathbf{E}_S} \mathbf{T}_{\omega, \delta}^{(\tau)}$, i.e. the set of observed interstate shipments from ω on day τ and $\mathbf{W} = (W_1, \dots, W_{|\mathbf{U}|})$. Further, for convenience, we introduce the notation $\Theta = (\Theta_1, \dots, \Theta_{|\mathbf{U}|})$ where $\Theta_S = (d_S, R_S, \lambda_S)$. This allows us to write $P_{\omega}^{(\tau, k)}(\mathbf{T}_{\omega}^{(\tau)} | \Theta_S, \mathbf{W})$, with $q_{\omega}^{(\tau, k)} = \sum_{\delta \in \mathbf{C} \setminus \mathbf{E}_S} p_{\delta | \omega}^{(\tau, k)}$, as

$$P_{\omega}^{(\tau,k)}(\mathbf{T}_{\omega}^{(\tau)}|\Theta_S, \mathbf{W}) = \prod_{\delta \in \mathbf{C} \setminus \mathbf{E}_S} \text{Poisson}(|\mathbf{T}_{\omega,\delta}^{(\tau)}| \hat{\lambda}_S n_{\omega} p_{\delta|\omega}^{(\tau,k)}) =$$

$$\text{Poisson} \left(|\mathbf{T}_{\omega}^{(\tau)}| \hat{\lambda}_S n_{\omega} q_{\omega}^{(\tau,k)} \right) \text{MN} \left(\left(\begin{array}{c} |\mathbf{T}_{\omega,1}^{(\tau)}| \\ \vdots \\ |\mathbf{T}_{\omega,|\mathbf{C} \setminus \mathbf{E}_S|}^{(\tau)}| \end{array} \right) \middle| |\mathbf{T}_{\omega}^{(\tau)}|, \left(\begin{array}{c} p_{w1} |\omega^{(\tau,k)}| / q_{\omega}^{(\tau,k)} \\ \vdots \\ p_{|\mathbf{C} \setminus \mathbf{E}_S| |\omega}^{(\tau,k)} / q_{\omega}^{(\tau,k)} \end{array} \right) \right), \quad (2.4)$$

where MN denotes the multinomial distribution. The variable $q_{\omega}^{(\tau,k)}$ is here defined as the sum of interstate probabilities. That is, $q_{\omega}^{(\tau,k)}$ is the probability that a shipment from county ω occurring on day τ leaves the state. In the same way as with $P_{\omega}^{(\tau,k)}(\mathbf{T}_{\omega}^{(\tau)}|\Theta_S, \mathbf{W})$, we constructed the probability model considering all origin counties in a specific state S for all days $\tau \in \tau$, where τ denotes the set of days in 2009, by multiplication of the individual probabilities in eq. 2.4. We introduce the notation $z_S^{(k)} = \sum_{\omega \in \mathbf{E}_S} n_{\omega} q_{\omega}^{(k)}$, and write the probability as

$$P_S^{(k)}(\mathbf{T}_S|\Theta_S, \mathbf{W}) = \prod_{\omega \in \mathbf{E}_S} \prod_{\tau \in \tau} P_{\omega}^{(\tau,k)}(\mathbf{T}_{\omega}^{(\tau)}|\Theta_S, \mathbf{W}) =$$

$$\text{Poisson} \left(|\mathbf{T}_S| \hat{\lambda}_S |\tau| z_S^{(k)} \right) \text{MN} \left(\left(\begin{array}{c} |\mathbf{T}_{1,1}^{(1)}| \\ \vdots \\ |\mathbf{T}_{|\mathbf{E}_S|,|\mathbf{C} \setminus \mathbf{E}_S|}^{(|\tau|)}| \end{array} \right) \middle| |\mathbf{T}_S|, \left(\begin{array}{c} p_{1,1}^{(1,k)} / (z_S^{(k)}|\tau|) \\ \vdots \\ p_{|\mathbf{E}_S|,|\mathbf{C} \setminus \mathbf{E}_S|}^{(|\tau|,k)} / (z_S^{(k)}|\tau|) \end{array} \right) \right). \quad (2.5)$$

Here, $\mathbf{T}_S = \bigcup_{\omega \in \mathbf{E}_S} \bigcup_{\tau \in \tau} \mathbf{T}_{\omega}^{(\tau)}$ denotes the set of all interstate shipments originating in state S and $q_{\omega}^{(k)} = \sum_{\tau \in \tau} q_{\omega}^{(\tau,k)}$. For convenience, we also define $\hat{z}_S^{(k)} = \frac{z_S^{(k)}}{\sum_{\omega \in \mathbf{E}_S} n_{\omega}}$, which is equal to the expected proportion of interstate shipments from state S .

(e) Modeling expert data

To further improve our estimation of the proportion interstate shipments, besides considering different shapes of the spatial kernel, we included expert opinions in the analysis. Bayesian analysis is well suited for this, and typically experts are used to elicit informative priors [21]. However, this is not straightforward in our analysis, both because of the model structure and the available expert estimates. The parameter we wish to inform, $\hat{z}_S^{(k)}$, is an implicit function of other parameters (d_S, R_S, \mathbf{W}) , and there is no convenient means to transform available expert opinions to priors for these parameters. Further, several states have a single expert respondent, providing a point estimate about expected proportions of interstate shipments. Thus, even if we could transform the questionnaire responses into an estimate about model parameters, it wouldn't be straightforward to specify a distribution from this estimate. Instead, we expanded our model to include information from expert questionnaire responses as data. We denote the state specific expert response from respondent $r_S \in \mathbf{r}_S$ as $\rho_{r,S}$, where \mathbf{r}_S denotes the set of respondents in state S and assume $\rho_{r,S}$ are distributed around the true (unknown) value $\hat{z}_S^{(k)}$, which is a function of the underlying process, quantified by model parameters d_S, R_S and \mathbf{W} . Because $\rho_{r,S}$ is confined on the range (0,1), we model expert data as Logit-Normally distributed around $\hat{z}_S^{(k)}$ as

$$P_{S,\rho}^{(k)}(\rho_S|d_S, R_S, \mathbf{W}, \xi_S) = \prod_{r \in \mathbf{r}_S} \text{Logit-Normal}(\rho_{r,S}|\hat{z}_S^{(k)}, \xi_S^{-1}), \quad (2.6)$$

where ξ_S quantifies the precision of experts. This may be interpreted such that for large ξ_S , experts have an exact opinion about $\hat{z}_S^{(k)}$, close to the true value. For low ξ_S , the experts have only a vague idea about $\hat{z}_S^{(k)}$. The subscript S indicate that ξ_S is a state specific measure, indicating that experts in some states may be more precise estimators about the proportion of shipment moving interstate than in others. The parameter ξ_S is only defined for states where expert opinions are

available (i.e. $r_S \neq \emptyset$) and we denote this set of parameters as ξ . For the states where $r_S = \emptyset$, we simply define $P_{S,\rho}^{(k)} = 1$.

(f) Likelihoods

We considered likelihoods for three assemblies of the data: ICVI only, experts only, and ICVI and experts combined. The full likelihood for ICVI only for all states (i.e. the set \mathbf{T}) is written as

$$L_{\mathbf{T}}^{(k)}(\mathbf{T}|\boldsymbol{\Theta}, \mathbf{W}) = \prod_{s \in \mathbf{U}} P_{s,\mathbf{T}}^{(k)}(\mathbf{T}_s|\boldsymbol{\Theta}_s, \mathbf{W}). \quad (2.7)$$

When using expert data only, we define the full likelihood in terms of the random variable ν_S , defined as logit of the estimated proportion of interstate shipments, rather than $\hat{z}_S^{(k)}$, which has the same definition (with the difference of not being logit transformed). However, $\hat{z}_S^{(k)}$ is a function of several random variables (d_S , R_S and \mathbf{W}), and not meaningful in the absence of the ICVI data. Based on eq. 2.6, the likelihood for experts only is defined as

$$L_{\rho}^{(k)}(\rho|\nu, \xi) = \prod_{S \in \mathbf{U}} P_{S,\rho}^{(k)}(\rho_S|\nu_S, \xi_S), \quad (2.8)$$

with state specific $\nu_S \in \nu$, $\xi_S \in \xi$ and $\rho_S \in \rho$, where ν , ξ and ρ are the set of logit-mean of expert opinions, the set of precision parameters, and the full set of expert data, respectively. Analogously, the full likelihood for ICVI and expert data combined is defined as

$$L_{\mathbf{T},\rho}^{(k)}(\mathbf{T}, \rho|\boldsymbol{\Theta}, \mathbf{W}, \xi) = \prod_{S \in \mathbf{U}} \left[P_{S,\mathbf{T}}^{(k)}(\mathbf{T}_S|d_S, R_S, \lambda_S, \mathbf{W}) P_{S,\rho}^{(k)}(\rho_{S,r}|d_S, R_S, \mathbf{W}, \xi_S) \right]. \quad (2.9)$$

The full likelihood $L_{\mathbf{T}}^{(k)}$ (eq. 2.7) can be interpreted such that we modeled the number of interstate shipments $|\mathbf{T}_S|$ from every state S , to come from a Poisson distribution with a state specific rate. These shipments were in turn considered as multinomially distributed among the origin counties, destination counties and days. Further, the likelihood $L_{\rho}^{(k)}$ consists of the experts' responses as logit-normal distributed data points conditional on mean ν_S and precision ξ_S . In both cases, the full likelihood consists of the product of state specific probabilities, hence the form of eq. 2.7 and 2.8. The full likelihood $L_{\mathbf{T},\rho}^{(k)}$ (eq. 2.9), including ICVI and expert data, is merely a product of the two likelihoods $L_{\mathbf{T}}^{(k)}$ and $L_{\rho}^{(k)}$. In total, we implemented seven different likelihood functions, six for the ICVI data using the three different kernels in the absence or presence of expert data and one for the expert data only. Additionally, we implement a likelihood where we use data from experts only to estimate the proportion of interstate shipments for states where these data are available. We analyzed beef and dairy shipments separately, since the two production types have different farming practices and consequently are likely to differ in their parameter estimates.

(g) Hierarchical Bayesian Model

We implemented a hierarchical Bayesian model for parameter estimation. This approach provides intelligible estimates regarding parameter uncertainty, which may be incorporated when the models are used for prediction. For $d_S \in \mathbf{d}$, $R_S \in \mathbf{R}$, $\lambda_S \in \boldsymbol{\lambda}$ and $\xi_s \in \boldsymbol{\xi}$, where bold symbols denote the set of parameters across all included states, we implemented

$$\begin{aligned} d_S &\sim \text{Log-normal}(m_{\mathbf{d}}, \kappa_{\mathbf{d}}) \\ R_S &\sim \text{Log-normal}_{m-1}(m_{\mathbf{R}}, \kappa_{\mathbf{R}}) \\ \lambda_S &\sim \text{Gamma}(m_{\boldsymbol{\lambda}}, \kappa_{\boldsymbol{\lambda}}) \\ \xi_S &\sim \text{Gamma}(m_{\boldsymbol{\xi}}, \kappa_{\boldsymbol{\xi}}), \end{aligned} \quad (2.10)$$

where subscript $m - 1$ indicates that the prior for R_S is shifted one unit to the right since R_S is defined on the interval $(1, \infty)$. Here m and κ , denote the mean and coefficient of

variation, respectively, and their relationship to the standard parameterization of the log-normal distribution is $m = e^{\mu - \frac{\sigma^2}{2}}$ and $\kappa = (e^{\sigma^2} - 1)^{\frac{1}{2}}$ where μ and σ are the mean and standard deviation, respectively, of the logarithm of the variable. The log-normal and shifted log-normal distribution differ in their relationship to standard parameterization in the way the mean is calculated. For the shifted distribution, the mean is expressed as $m_{\mathbf{R}} = e^{\mu - \frac{\sigma^2}{2}} + 1$, whereas the calculation of κ is unchanged. As for the gamma distribution, the mean and coefficient of variation are expressed as $m = \alpha\beta^{-1}$ and $\kappa = \alpha^{-\frac{1}{2}}$, respectively, where α and β denote shape and rate in the standard parameterization, respectively. Using a hierarchical model structure, m and κ parameters were treated as random variables, and the alternative parameterization facilitates cognizant hyperprior elicitation as specified in section (h). Denoting the prior distribution for parameter $x_S \in \{d_S, R_S, \lambda_S, \xi_S\}$ as $\Phi_x(x_S|m_x, \kappa_x)$ and corresponding hyperpriors as $\Psi_m^{(x)}(m_x)$ and $\Psi_\kappa^{(x)}(\kappa_x)$, the full Bayesian model is given as

$$\begin{aligned}
 & P_{\mathbf{T}}^{(k)}(\boldsymbol{\Theta}, \mathbf{W}, m_{\mathbf{d}}, \kappa_{\mathbf{d}}, m_{\mathbf{R}}, \kappa_{\mathbf{R}}, m_{\boldsymbol{\lambda}}, \kappa_{\boldsymbol{\lambda}} | \mathbf{T}) \propto \\
 & L_{\mathbf{T}}^{(k)} \prod_{S \in \mathbf{U}} [\Phi_{\mathbf{d}}(d_S | m_{\mathbf{d}}, \kappa_{\mathbf{d}}) \Phi_{\mathbf{R}}(R_S | m_{\mathbf{R}}, \kappa_{\mathbf{R}}) \Phi_{\mathbf{W}}(W_S) \Phi_{\boldsymbol{\lambda}}(\lambda_S | m_{\boldsymbol{\lambda}}, \kappa_{\boldsymbol{\lambda}})] \cdot \\
 & \Psi_m^{(\mathbf{d})}(m_{\mathbf{d}}) \Psi_\kappa^{(\mathbf{d})}(\kappa_{\mathbf{d}}) \Psi_m^{(\mathbf{R})}(m_{\mathbf{R}}) \Psi_\kappa^{(\mathbf{R})}(\kappa_{\mathbf{R}}) \Psi_m^{(\boldsymbol{\lambda})}(m_{\boldsymbol{\lambda}}) \Psi_\kappa^{(\boldsymbol{\lambda})}(\kappa_{\boldsymbol{\lambda}}) = \\
 & \prod_{S \in \mathbf{U}} \left[P_{S, \mathbf{T}}^{(k)}(\mathbf{T}_S | \boldsymbol{\Theta}_S, \mathbf{W}) \Phi_{\mathbf{d}}(d_S | m_{\mathbf{d}}, \kappa_{\mathbf{d}}) \Phi_{\mathbf{R}}(R_S | m_{\mathbf{R}}, \kappa_{\mathbf{R}}) \Phi_{\mathbf{W}}(W_S) \Phi_{\boldsymbol{\lambda}}(\lambda_S | m_{\boldsymbol{\lambda}}, \kappa_{\boldsymbol{\lambda}}) \right] \cdot \\
 & \Psi_m^{(\mathbf{d})}(m_{\mathbf{d}}) \Psi_\kappa^{(\mathbf{d})}(\kappa_{\mathbf{d}}) \Psi_m^{(\mathbf{R})}(m_{\mathbf{R}}) \Psi_\kappa^{(\mathbf{R})}(\kappa_{\mathbf{R}}) \Psi_m^{(\boldsymbol{\lambda})}(m_{\boldsymbol{\lambda}}) \Psi_\kappa^{(\boldsymbol{\lambda})}(\kappa_{\boldsymbol{\lambda}})
 \end{aligned} \tag{2.11}$$

for the ICVI data only, as

$$\begin{aligned}
 & P_{\rho}^{(k)}(\nu, \xi, m_{\xi}, \kappa_{\xi} | \rho) \propto L_{\rho}^{(k)} \prod_{S \in \mathbf{U}} [\Phi_{\nu}(\nu_S) \Phi_{\xi}(\xi_S | m_{\xi}, \kappa_{\xi})] \Psi_m^{(\xi)}(m_{\xi}) \Psi_\kappa^{(\xi)}(\kappa_{\xi}) = \\
 & \prod_{S \in \mathbf{U}} \left[P_{S, \rho}^{(k)}(\rho_S | \nu_S, \xi_S) \Phi_{\nu}(\nu_S) \Phi_{\xi}(\xi_S | m_{\xi}, \kappa_{\xi}) \right] \Psi_m^{(\xi)}(m_{\xi}) \Psi_\kappa^{(\xi)}(\kappa_{\xi}).
 \end{aligned} \tag{2.12}$$

for expert data only, and as

$$\begin{aligned}
 & P_{\mathbf{T}, \rho}^{(k)}(\boldsymbol{\Theta}, \mathbf{W}, \xi, m_{\mathbf{d}}, \kappa_{\mathbf{d}}, m_{\mathbf{R}}, \kappa_{\mathbf{R}}, m_{\boldsymbol{\lambda}}, \kappa_{\boldsymbol{\lambda}}, m_{\xi}, \kappa_{\xi} | \mathbf{T}, \rho) \propto \\
 & L_{\mathbf{T}, \rho}^{(k)} \prod_{S \in \mathbf{U}} [\Phi_{\mathbf{d}}(d_S | m_{\mathbf{d}}, \kappa_{\mathbf{d}}) \Phi_{\mathbf{R}}(R_S | m_{\mathbf{R}}, \kappa_{\mathbf{R}}) \Phi_{\mathbf{W}}(W_S) \Phi_{\boldsymbol{\lambda}}(\lambda_S | m_{\boldsymbol{\lambda}}, \kappa_{\boldsymbol{\lambda}}) \Phi_{\xi}(\xi_S | m_{\xi}, \kappa_{\xi})] \cdot \\
 & \Psi_m^{(\mathbf{d})}(m_{\mathbf{d}}) \Psi_\kappa^{(\mathbf{d})}(\kappa_{\mathbf{d}}) \Psi_m^{(\mathbf{R})}(m_{\mathbf{R}}) \Psi_\kappa^{(\mathbf{R})}(\kappa_{\mathbf{R}}) \Psi_m^{(\boldsymbol{\lambda})}(m_{\boldsymbol{\lambda}}) \Psi_\kappa^{(\boldsymbol{\lambda})}(\kappa_{\boldsymbol{\lambda}}) \Psi_m^{(\xi)}(m_{\xi}) \Psi_\kappa^{(\xi)}(\kappa_{\xi}) = \\
 & \prod_{s \in \mathbf{U}} \left[P_{S, \mathbf{T}}^{(k)}(\mathbf{T}_S | d_S, R_S, \lambda_S, \mathbf{W}) P_{S, \rho}^{(k)}(\rho_{S, \mathbf{r}} | d_S, R_S, \mathbf{W}, \xi_S) \cdot \right. \\
 & \left. \Phi_{\mathbf{d}}(d_S | m_{\mathbf{d}}, \kappa_{\mathbf{d}}) \Phi_{\mathbf{R}}(R_S | m_{\mathbf{R}}, \kappa_{\mathbf{R}}) \Phi_{\mathbf{W}}(W_S) \Phi_{\boldsymbol{\lambda}}(\lambda_S | m_{\boldsymbol{\lambda}}, \kappa_{\boldsymbol{\lambda}}) \Phi_{\xi}(\xi_S | m_{\xi}, \kappa_{\xi}) \right] \cdot \\
 & \Psi_m^{(\mathbf{d})}(m_{\mathbf{d}}) \Psi_\kappa^{(\mathbf{d})}(\kappa_{\mathbf{d}}) \Psi_m^{(\mathbf{R})}(m_{\mathbf{R}}) \Psi_\kappa^{(\mathbf{R})}(\kappa_{\mathbf{R}}) \Psi_m^{(\boldsymbol{\lambda})}(m_{\boldsymbol{\lambda}}) \Psi_\kappa^{(\boldsymbol{\lambda})}(\kappa_{\boldsymbol{\lambda}}) \Psi_m^{(\xi)}(m_{\xi}) \Psi_\kappa^{(\xi)}(\kappa_{\xi})
 \end{aligned} \tag{2.13}$$

for ICVI and expert data combined. Prior distributions $\Phi_{\mathbf{W}}(W_S)$ for models including ICVI data and $\Phi_{\nu}(\nu_S)$ for models with only experts were included as fixed distributions, without hierarchical structure.

(h) Prior elicitation

Our general approach for eliciting prior and hyperprior distributions was to first identify the range on which the parameters are defined and then choose suitable prior distributions with domains matching that range. To obtain statistics we can have at least a minimal intuitive expectation about, we expressed the priors on the mean (m_x) and coefficient of variation (κ_x) for parameters $x \in (\mathbf{d}, \mathbf{R}, \boldsymbol{\lambda}, \boldsymbol{\xi})$. For these parameters, we deduced hyperpriors by specifying a range of plausible values in which we, with 95% certainty, believe encapsulates the true parameter value. By choosing 95%, we do not exclude more extreme values but consider them unlikely. The model's sensitivity to our choices of hyperpriors was evaluated by choosing alternative hyperpriors as described in section (i).

Because \mathbf{d} is defined on the range $(0, \infty)$, the prior distribution $\Phi_{\mathbf{d}}(d_S|m_{\mathbf{d}}, \kappa_{\mathbf{d}})$, was chosen as log-normal. We further specified the hyperpriors for $m_{\mathbf{d}}$ and $\kappa_{\mathbf{d}}$ as log-normal distributions and used our general approach of identifying a range of plausible values. For the hyperprior $\Psi_m^{(\mathbf{d})}(m_{\mathbf{d}})$, we elicited hyperparameters such that the average distance to where destinations are half as likely as an immediate neighbor is within the range of 10 km and 4,000 km. We find these to be reasonable values for a vague hyperprior because 10 km would be a very short shipment distance considering the spatial scales of the U.S. livestock system and 4,000 km is the approximate distance between the east- and west coast of the U.S., which would be a high value for the average distance to where a destination premises is half as likely as an immediate neighbor. Since $\mathbf{R} \in (1, \infty)$ is not defined on the whole positive real line we chose a log-normal prior distribution shifted one unit to the right ($\Phi_{\mathbf{R}}(R_S|m_{\mathbf{R}}, \kappa_{\mathbf{R}})$). For the hyperprior $\Psi_m^{(\mathbf{R})}(m_{\mathbf{R}})$, we chose parameters such that the interpretation is that we put 95% of the density of the mean of \mathbf{R} between 2 and 1,000. This hyperprior allow for a wide range of plausible values of \mathbf{R} . A value of 2 would correspond to a steep drop of the kernel between distances d and $2d$ (where the kernel attains 5% of its initial value). Conversely, a value of 1,000 would correspond to a flat kernel (the corresponding decrease in kernel value occurs between the distances d and $1000d$). Thus, this range mirrors our vague a priori beliefs regarding the distance dependence. To achieve conjugacy, we implemented the Gamma distributions as prior for the rate parameters λ and expert precisions ξ , denoted $\Phi_{\lambda}(\lambda_S|m_{\lambda}, \kappa_{\lambda})$ and $\Phi_{\xi}(\xi_S|m_{\xi}, \kappa_{\xi})$, respectively. Similar to the kernel parameters, we defined the hyperprior in terms of means and coefficient of variation (i.e. $m_{\lambda}, \kappa_{\lambda}$ and m_{ξ}, κ_{ξ}). For hyperprior $\Psi_m^{(\lambda)}(m_{\lambda})$, we implemented a log-normal distribution such that the prior distribution of m_{λ} has 95% of the density between 0.00027 and 0.27 shipments per day. These numbers were derived from the vague prior beliefs that 95% of the density of the average rate of yearly shipments per premises in the average state, is between 0.1 and 100 shipments. Further, when expert data were included, the hyperprior was also chosen as log-normal and parameters for the mean of the precision parameter were chosen such that 95% of its density lies within the range $(0.1, 10)$. Precision parameters within this interval will allow wide as well as narrow distributions of the expert opinions and therefore constitutes a vaguely informative hyperprior for m_{ξ} .

To define hyperpriors for the coefficients of variation, $\Psi_{\kappa}^{(\mathbf{d})}(\kappa_{\mathbf{d}})$, $\Psi_{\kappa}^{(\mathbf{R})}(\kappa_{\mathbf{R}})$, $\Psi_{\kappa}^{(\lambda)}(\kappa_{\lambda})$ and $\Psi_{\kappa}^{(\xi)}(\kappa_{\xi})$, we expressed our beliefs in terms of expectations regarding how similar underlying parameters are between states. We therefore used the ratio between the median of corresponding m_x and its 97.5th percentile. For this ratio, we chose the lower limit as two, which corresponds to the case of high similarity between states regarding the underlying parameters \mathbf{d} , \mathbf{R} , λ and ξ . Thus, we obtained a ratio of the 97.5th and the 2.5th percentile equal to four in this case. Further, we chose the upper limit such that it corresponds to a ratio of one order of magnitude, i.e. ten. As a consequence, the ratio of the 97.5th and the 2.5th percentile is equal to two orders of magnitude. This ratio corresponds to distributions expressing large differences in the state specific estimates. From the limits above, we deduced the hyperprior distributions $\Psi_{\kappa}^{(\mathbf{d})}(\kappa_{\mathbf{d}})$, $\Psi_{\kappa}^{(\mathbf{R})}(\kappa_{\mathbf{R}})$, $\Psi_{\kappa}^{(\lambda)}(\kappa_{\lambda})$ and $\Psi_{\kappa}^{(\xi)}(\kappa_{\xi})$, as log-normal with 95% of its density between 0.3650 and 1.724. We chose the prior distribution of \mathbf{W} , $\Phi_{\mathbf{W}}(W_S)$, as a log-normal distribution with parameters such that 95% of the density lies in the range $(0.01, 100)$. That is, we formulate a vague hyperprior expressing that plausible values of the propensity to attract shipments parameter for a premises in a certain state ranges between 0.01 and 100 times the corresponding parameter of a premises in our arbitrary reference state Missouri. Further, in the analysis of expert data only (eq. (2.12)) we implemented $\Phi_{\nu}(\nu_S) \propto \frac{1}{\nu(1-\nu)}$ for $\nu_S \in (0, 1)$, i.e. the prior for $\text{logit}(\nu_S)$ is uniform.

(i) Sensitivity Analysis

To investigate the sensitivity of the posterior to our choice of hyperpriors, we conducted a sensitivity analysis to assess the robustness of our results. To identify parameters of potential concern, we used the criteria that marginal posteriors that had more than 1% density of either tail

outside the interval of the hyperprior containing 95% of the density could indicate that our choice of hyperpriors restricted the posterior range. Thus, if the first percentile of the marginal posterior distribution was lower than the 2.5th percentile of the hyperprior, we investigated the effect of the elicited hyperprior by decreasing the lower bound of the 95% interval by which it was defined by 50%. Analogously, we adjusted the hyperprior if the 99th percentile of the marginal posterior was greater than the 97.5 percentile by doubling this upper bound. We then re-analyzed the data and compared the estimates corresponding to the elicited and alternative hyperpriors.

(j) Computation

None of the Bayesian models (eq. 2.11-2.13) have a standard form, and we therefore relied on numerical algorithms to estimate the posterior. We used Markov Chain Monte Carlo (MCMC) methods to approximate the marginal posterior distributions. The idea of the MCMC approach is to simulate a Markov Chain whose limiting state distribution is equal to the posterior and from this obtain samples of the parameters. For computational purposes, eq. 2.7 and 2.9 can be simplified by expanding $P_{S,\mathbf{T}}^{(k)}(\mathbf{T}_\omega^{(\tau)}|\Theta_S, \mathbf{W})$ in eq. 2.5. Since most of the $\mathbf{T}_{\omega,\delta}^{(\tau)} = \emptyset$, i.e. we have not observed any shipments between counties ω and δ for the majority of days τ , we can omit the corresponding terms in the probability vector in the multinomial distribution since they are equal to one. Therefore,

$$L_{\mathbf{T}}^{(k)}(\mathbf{T}|\Theta) = \prod_{S \in \mathbf{U}} P_{S,\mathbf{T}}^{(k)}(\mathbf{T}_S|\Theta_S, \mathbf{W}) = \prod_{S \in \mathbf{U}} \text{Poisson}(|\mathbf{T}_S| | \hat{\lambda}_{\mathbf{E}_S} | \tau | z_S^{(k)}) \text{MN} \left(\begin{pmatrix} |\mathbf{T}_{1,1}^{(1)}| \\ \vdots \\ |\mathbf{T}_{|\mathbf{E}_S|,|\mathbf{C} \setminus \mathbf{E}_S|}^{(\tau)}| \end{pmatrix} \middle| |\mathbf{T}_S|, \begin{pmatrix} p_{1|1}^{(1,k)} / (z_S^{(k)} | \tau |) \\ \vdots \\ p_{|\mathbf{E}_S|,|\mathbf{C} \setminus \mathbf{E}_S|}^{(|\tau|,k)} / (z_S^{(k)} | \tau |) \end{pmatrix} \right) \propto \prod_{S \in \mathbf{U}} \left(\text{Poisson}(|\mathbf{T}_S| | \hat{\lambda}_{\mathbf{E}_S} | \tau | z_S^{(k)}) \prod_{t \in \mathbf{T}_S} \begin{pmatrix} p_{\delta_t | \omega_t}^{(k)} \\ q_{\delta_t | \omega_t}^{(k)} \end{pmatrix} \right) \quad (2.14)$$

and

$$L_{\mathbf{T},\rho}^{(k)}(\mathbf{T}, \rho | \Theta, \mathbf{W}, \xi) \propto \prod_{S \in \mathbf{U}} \left(\text{Poisson}(|\mathbf{T}_S| | \hat{\lambda}_{\mathbf{E}_S} | \tau | z_S^{(k)}) \prod_{t \in \mathbf{T}_S} \begin{pmatrix} p_{\delta_t | \omega_t}^{(k)} \\ q_{\delta_t | \omega_t}^{(k)} \end{pmatrix} \prod_{r \in \mathbf{r}_S} \text{Logit-Normal}(\rho_{r_S} | \hat{z}_S^{(k)}, \xi_S^{-1}) \right). \quad (2.15)$$

This can be interpreted as the likelihood of observing $|\mathbf{T}_S|$ number of shipments from every origin state $S \in \mathbf{U}$, multiplied with the likelihood of every observed shipment $t, \forall t \in \mathbf{T}_S$ to have origin $\omega_t \in \mathbf{E}_S$ and destination county $\delta_t \in \mathbf{C} \setminus \mathbf{E}_S$, conditional on the shipment leaving the state. Additionally, eq. 2.15 includes the likelihood of expert responses ρ_{r_S} . In our implementation, the latter forms of the likelihoods in eq. 2.14 and 2.15 are used. For most of our model parameters, the conditional distribution is not of a standard form, and we implemented Metropolis-Hastings updates of these parameters [30]. This involves proposing candidates from a proposal distribution $Q(y_{prop}|y_{acc})$ based on the current state of the Markov Chain. Here, y_{prop} and y_{acc} denotes the proposed and latest accepted state (i.e. parameter values) of the Markov Chain, respectively. The proposed values are accepted with probability

$$\Upsilon(y_{prop}|y_{acc}) = \min \left(1, \frac{Q(y_{acc}|y_{prop})P^{(k)}(y_{prop})}{Q(y_{prop}|y_{acc})P^{(k)}(y_{acc})} \right), \quad (2.16)$$

where $P^{(k)}$ denotes the posterior of the models in eq. 2.11-2.13. This gives us a Markov Chain with transition distribution

$$\Pi(y_{prop}|y_{acc}) = Q(y_{prop}|y_{acc})\Upsilon(y_{prop}|y_{acc}), \quad (2.17)$$

which has the target density $P^{(k)}$ as limiting state distribution [30]. The state specific kernel parameters (d_S, R_S) were updated jointly with a bivariate normal random walk on the log-transform of the parameters $(d_S, R_S - 1)$. Similarly, the weight parameters \mathbf{W} (excluding $W_{Missouri}$, which was fixed to one for identifiability) were updated jointly with a 47-dimensional multivariate normal random walk on the log-transform of the parameters. For the model with expert data only, we performed the random walk on $y = \text{logit}(\nu_s)$ and proposals of y were drawn from a normal distribution. All hyperparameters (m_x, κ_x) were updated with a bivariate normal random walk distribution on the log-scale. To avoid manually tuning the proposal distributions to achieve acceptable mixing, we implemented an optimized version of the Robbins-Monro process as introduced by [31]. This estimates the mean and covariance of the parameters sets that are updated jointly, along with a scaling factor that facilitates a chosen long-term acceptance rate, here set to 0.234 as proposed in [32]. For the low dimensional model with experts only, we however found it sufficient to get good mixing by letting the standard deviation of the normally distributed proposals for $\text{logit}(\nu_S)$ be fixed at 0.01. The choice of gamma prior for λ and ξ provides conjugacy, and we sampled directly from their conditional distributions as

$$\lambda_S \sim \text{Gamma}(\alpha_\lambda |\mathbf{T}_s|, \beta_\lambda + |\boldsymbol{\tau}| z_S^{(k)}) \quad (2.18)$$

and

$$\xi_S \sim \text{Gamma}\left(\alpha_\xi + \frac{|\mathbf{r}_S|}{2}, \beta_\xi + \frac{\sum_{r \in \mathbf{r}_S} (\text{logit}(\rho_r) - \text{logit}(\hat{z}_S^{(k)}))}{2}\right). \quad (2.19)$$

In eq. 2.18 and 2.19, the parameters $(\alpha_\lambda, \beta_\lambda)$ and (α_ξ, β_ξ) are the shape and rate parameters of the Gamma prior distribution for λ and ξ , respectively, given from our parameterization by mean and coefficient of variation. Their relationship is specified in section (g). For each model, we ran 3 simulations with over-dispersed start values to assure convergence to high posterior density regions. We ran each simulation for 4,000,000 iterations and discarded the first 2,000,000 iterations as burn in. Further, because the chains typically showed a high degree of autocorrelation, we thinned our chains by 90% to avoid excessively large output files. Thus, for every model we obtained 600,000 draws from the posterior distribution. To verify our code, we repeatedly analyzed data simulated with known parameters to ensure the model correctly retrieved the parameters used for simulation. As a measure of the efficacy of the algorithm, we estimated the number independent samples as given by Gelman [33]. These calculations are based on the assumptions of approximately normally distributed samples, and we therefore analyzed the logarithm of $\mathbf{d}, \mathbf{R} - 1, \lambda, \mathbf{W}$ and hyperparameters m_x, κ_x , except for m_R which was log transformed as $m_R - 1$. Further, we calculated the potential scale reduction factor (PSRF, [34]), which provides a measure of convergence among chains. When using the adjusted hyperprior in the sensitivity analysis together with kernel one and beef data, some states had two high posterior density regions (HDR) of (d_S, R_S) on the log-scale (on which the random walk was performed). This led to poor mixing of the chains and subsequently few independent draws for these states. We therefore applied an ad-hoc solution by adjusting the proposal distributions for the considered parameters. From preliminary runs, we divided the samples of $(\log(d_S), \log(R_S))$ for each considered state into one of the two HDRs by comparing the parameter value to some limits of our choice that separated the two HDRs. We continued by calculating the mean and the covariance of the MCMC-chains of $(\log(d_S), \log(R_S))$ for samples from the two HDRs. We constructed two bivariate log-normal proposal distributions with the mean taken from the HDRs and with covariance as calculated above, multiplied by four. Our aim was to construct proposal distributions for the two HDRs that were wide enough to sample from the entire posterior with good mixing. For every iteration, one of the two proposals was used with probability 0.5. That is, the proposals of $(\log(d_S), \log(R_S))$ was obtained by alternately sampling from the two distributions estimated from the two HDRs. Note that this computational technique does not change estimates; it was only used to enhance mixing for states where our original methods performed poorly. This update technique was used for Florida, Kentucky and Mississippi

regardless of the use of additional expert data. When incorporating expert data, Arkansas was added to the list of states where bimodal proposals were used.

14

(k) Model selection and validation

To rank our proposed models by their level of parsimony, we used two types of information criteria: Deviance Information Criterion (DIC) [35] and Widely Applicable Information Criterion (WAIC) [36]. Both criteria are derived from estimating the out-of-sample predictive accuracy using within-sample data and a penalty term for the overestimation of the accuracy this leads to [37]. Both criteria have similar interpretation; a lower score indicates the preferred model. DIC estimates the fit from the log-likelihood (or log predictive density) of the data conditional on the posterior means of the parameters and the penalty term used in DIC is equal to the effective number of parameters [35]. We chose however to use the median of the parameters (as proposed in [35]) since it proved to increase numerical stability in our study because posterior densities typically exhibited high skewness. In practice, DIC is easily estimated in MCMC algorithms and is therefore a convenient tool for model selection. However, DIC is not fully Bayesian since the log predictive density conditions on point estimates (in our case the median), and concerns has been raised about a tendency for DIC to favor complex models [38]. We therefore used WAIC as an additional measure for model selection, which is a more fully Bayesian approach. WAIC uses the (computed) log pointwise posterior predictive density to estimate the out-of-sample predictive accuracy. This pointwise approach means that WAIC better capture the posterior uncertainty than DIC. Consequently, WAIC relies on a partitioning of the data into well-defined data points (in our case $\mathbf{T}_{\omega,\delta}^{(\tau)}$). This is why we use the model definition of the form as in 2.7 and 2.9 instead of the more concise form as in 2.14 and 2.15; zero observed shipments between two counties is also an observation. For WAIC, two types of penalty terms have been proposed. We chose to follow the advice of Gelman et al. [37] and used the posterior sample variance of the log predictive density. For computational reasons, we used a subsample of the iterations in the MCMC algorithm when calculating WAIC and we chose 20,000 random samples of our parameters from the last 2,000,000 iterations. The measures above provide information on the accuracy of our model's predictive ability and we therefore implemented an additional strategy for validation. We compared predictions of the models to ICVI data, including other years' data where available, by network summary statistics to ensure our model re-captures relative features of the original data. For these purposes, we simulated 1,000 networks from each model based on random draws from the posterior distribution. We compared distributions of shipment distances in the ICVI data to corresponding distributions based on posterior predictive simulations. The latter were down-sampled to 10% interstate shipments and no intrastate shipments to make them comparable to the former. Further, we calculated for each state the correlation between ICVI data and simulated networks in terms of destination states.

3. Results

Our models for the ICVI data included four state-level parameters describing the shipment pattern: one modeling shipment rate describing the rate at which premises in each state generate shipments (λ_S), one modeling propensity to attract shipments (\hat{W}_S), and two kernel parameters modeling how the probability of shipments decay with distance (scale, d_S , and shape, R_S). The analyses revealed a heterogeneous shipment pattern across the US, with large variation in all estimated parameters across states and production systems, as exemplified for four selected states in Figure 2. Parameters λ_S and \hat{W}_S varied among states by more than an order of magnitude, and estimates were similar across the three kernels. The kernel parameters d_S and R_S also varied substantially among states, pinpointing the importance of accounting for state level heterogeneity when assessing shipment distances. These estimates however depended heavily on the choice of functional form. Whereas parameters for kernel two and three showed great similarity, estimates for kernel one consistently differed from the other two. Notably, kernel one estimates for d_S ,

rsos.royalsocietypublishing.org R. Soc. open sci. 0000000

Figure 2. 95% credibility intervals for selected states and all implemented kernels. The rows show from top to bottom, d (km), shape parameter R (unitless), propensity to attract shipments \hat{W} (unitless), and shipment rate λ (shipment \cdot day $^{-1}$). Left and right columns show credibility intervals for beef and dairy, respectively. Results are shown for analysis including expert data.

defined as the distance where the probability of destination has dropped to half of the probability of an immediate neighbor, were often remarkably low, including estimates below 10^{-7} km. Estimates for all parameters and kernels and are shown in supplementary material S1. Estimates for expert precision parameters (ξ) were similar for all kernels, and the credibility intervals largely overlapped for all states where expert data were available. Detailed results for ξ are shown in supplementary material S2.

The variability in the underlying parameters elicited a heterogeneous pattern in terms of the predicted proportion of shipments moving intrastate (Figure 3), and the number of shipments

Figure 3. Predicted proportion of shipments moving interstate for beef and dairy combined based on three different functional forms of the spatial kernel. Results are shown for analyses where experts were included for available states.

moving in and out of each state varied substantially for both beef and dairy shipment (Figure 4). County-level networks are made available as csv files in the supplement S8.

(a) Model selection and the effect of experts

Independent of selection criteria (Table 2), model selection consistently disfavored the kernel functional form one. The choice between functional forms two and three varied between data sets (beef or dairy), but selection criteria were considerably more similar than either of these two was to functional form one. Further, the posterior predictive distributions of shipment distances showed high similarity between functional forms two and three and fit better with the distances observed in the ICVI data than did functional form one (Figure 5). The choice of kernel functional form only had minor effect on the large-scale contact pattern, providing near identical median estimates of in- and out-degree (Figure 4). Thus, we primarily focus on functional form three and present equivalent analyses for the other kernels in the supplementary material (S1,S2,S5,S7).

The inclusion of experts did not change ranking of kernels in terms of model selection (Table 2). Figure 6 further shows that the posterior predictive estimates of p using experts only data typically provided wide intervals of expected proportion of shipments moving intrastate. Consequently, their contribution to the analysis of both experts and ICVI data was moderate, shifting only slightly the corresponding distributions of p . The experts also had only marginal effect on the models' ability to recapture between state link strengths (Figure 7). Because of the low impact of experts on the estimates, we primarily focus on estimates including expert and provide non-expert analyses in the supplementary material (S1,S3,S4,S5,S7).

(b) Validation

We performed both within- and out-of-sample validation. The posterior predictions recaptured the relevant large-scale structure of the 2009 ICVI data, with the number of incoming and

Figure 4. Number of incoming (A, Beef, and B, Dairy) and outgoing (C, Beef, and D, Dairy) shipments for U.S. states as given by the ICVI data and the corresponding median prediction from 1,000 realizations with each implemented kernel. Simulated shipments were down-sampled by 90% to correspond to the ICVI data. Results are shown for analyses where experts were included for available states.

Commodity	Data	Kernel	Δ WAIC	Δ DIC	Minimum number of independent samples
Beef	ICVI	1	625.0	630.0	6873 (W VT)
		2	14.2	11.5	2445 (d OH)
		3	0.0	0.0	3805 (d OH)
	ICVI+Experts	1	626.4	630.1	9505 (W OH)
		2	21.0	10.4	2348 (λ OH)
		3	0.0	0.0	6071 (R FL)
Dairy	ICVI	1	530.0	531.0	4724 (σ_d)
		2	0.0	0.0	4876 (W IN)
		3	45.2	45.6	6050 (W IA)
	ICVI+Experts	1	510.0	512.1	6631 (W WI)
		2	0.0	0.0	6792 (W OR)
		3	45.3	46.0	11047 (W MT)

Table 2. Differences from the preferred kernel functional form for two model selection criteria (Δ WAIC and Δ DIC) for each considered data set. The minimum number of independent samples includes in parenthesis for which (hyper)parameter (and state where applicable) the minimum value was estimated for.

outgoing shipments showing striking similarity to the ICVI data (Figure 4). The analysis of between state link strengths showed a high correlation between posterior predicted networks and the ICVI data (Figure 7).

Importantly, we were also able to utilize additional ICVI data from 2010 and 2011 for selected states to perform out-of-sample validation. Independent of kernel functional form, we found the link strength from these states correlated well with USAMM predictions (Figure 8), with the

Figure 5. Distributions of interstate shipment distances for Arizona, Kentucky and Mississippi, comparing ICVI data (red) to posterior predictions (green) to models with different functional forms for the kernel. Results are shown for dairy and beef shipments combined for analyses where experts were included for available states.

exception 2010 and 2011 shipments from North Carolina and 2010 shipments from Nebraska. In all instances, correlations of 2010 and 2011 ICVI data with USAMM predictions were comparable to correlations of 2010 and 2011 ICVI data with correlations 2009 ICVI data, indicated as red diamonds in Figure 8. This result and the overall high correlation in Figure 7 show that regardless of choice of kernel, the model captures the shipping patterns. That is, both within- and out-of-sample validation verifies our models at the state level.

Comparing distance distribution for these states (Figure 9) revealed similar shipment distances across years, with the exception of Iowa and Minnesota. The 95% posterior predictive bands typically do not fully envelope the observed distribution curves for all distances, but USAMM captured the broad pattern of shipment distances. The sensitivity analysis showed that using a wider hyperprior only had a minor influence on parameter estimates and did not change ranking of models. A comprehensive presentation of the results of the sensitivity analysis is provided in the supplementary material (S6).

(c) Computation

PSRF investigates convergence for MCMC chains, and a value close to one is expected if all chains have converged. For the models considered here, all PSRF were less than 1.001, indicating sufficient convergence. We further estimated the number of independent samples and Table 2 shows that the lowest value across models and data was estimated at 2,348.

Figure 6. Median and 95% credibility interval of posterior distributions of proportion of movements moving interstate for analysis including only experts (red), ICVI and experts together (green) and ICVI data only (blue). Results are shown for analyses with kernel functional form three.

4. Discussion

Understanding between premises contact patterns is essential for epidemiological modeling of domestic animal diseases [12,39] and can be used to inform preparedness [40,41] and surveillance [4,42]. Animal shipments are of particular importance because of their high risk [3] and potentially large spatial scales [18,43]. However, information on complete shipments is not always accessible, necessitating methods that scale up from available information.

The version of USAMM presented here includes several important improvements from previous versions. We focused on improving state level estimates, specifically targeting the proportion of shipments moving within-state. Building on the framework of Lindström et al. [17], we made three important improvements for predicting inter- and intrastate animal shipments. First, we estimated the propensity to attract shipments (W) in the model, rather than a priori specifying it as a fixed constant, as was done by Lindström et al. This advancement improved the correlation in between state link strength to >0.9 (Figure 7) compared to 0.8 in the model of Lindström et al., indicating the models presented here captures the overall state pattern better than previous versions of USAMM.

Secondly, we changed the model structure to model rate of shipments per premises (λ) rather than number of shipments per state. This change offers a more intuitive sense of the parameter and is more likely to be reported in other studies since is interpreted as premises level rate rather than a state level quantity, which depends on the number of premises in the state. For the same reasons, it also facilitated prior elicitation and fits better in a hierarchical model structure. Further,

Figure 7. Boxplots of correlations between the 2009 ICVI data and 1000 predicted network with three different kernel functional forms, with and without expert data, in terms of number of shipments between all contiguous states. Boxes show inter-quartile range and horizontal line within boxes indicate median. Whiskers show highest and lowest values within 1.5 times inter-quartile range and dots represents outliers, Results are shown for networks with beef and dairy shipments combined.

the changes improved computation since the rate parameter could be Gibbs-sampled through the conditional distribution, given a conjugate gamma prior. Importantly, it also allowed for inclusion of shipment rate in model selection based on WAIC.

Thirdly, we implemented three different functional forms for the spatial kernel that model how the probability of destination decays with distance. In countries where complete data is available, shipments typically occur at shorter distances [16,44,45], and it is therefore important when modeling outbreaks of infectious diseases to have accurate estimations of the kernel’s behavior at distances where the majority of shipments occurs. Yet, it is equally important to accurately predict shipments at long distances since these can spread pathogens to previously uninfected areas and spark new local outbreaks [3,46]. Kernels two and three take on a plateau-shape at short distances, even when the estimated shape encompasses a fat tail. Conversely, kernel one exhibits a peaked behavior at short distances when the kernel tail is fat. It is particularly essential to investigate the effect of these differences since the behavior at short distances determines how many shipments are estimated to move within-state. The model selection consistently disfavored kernel one (Table 2), which has the functional form used by Lindström et al. (2013). This illustrates that this kernel did worse than the other two in representing shipment distances at both short and long distances. Further, kernel one estimates of scale parameter d included values below 10^{-7} km (Figure 2). These are exceptionally short distances considering the large spatial scales of the U.S.; most of the drop-in probability occurs at shorter distances than possible to observe

Figure 8. Pearson correlations for number of interstate shipments between networks (1000 realizations, boxplots) generated with USAMM with different kernel functional forms, fit to ICVI data from 2009, and ICVI data for 2010 and 2011. Corresponding correlations between the 2009 ICVI data and ICVI data for 2010 and 2011 are indicated as red diamonds. Beef and dairy combined.

in the data. As such, the functional form of kernel one is unsuitable for the system. Further, the large differences in kernel parameter estimates between states and kernels show the importance of considering different kernels and accounting for spatial heterogeneity when modeling the distance dependence, especially since animal shipments can play an important role in spreading infectious diseases [3].

The predictions of shipments given by Lindström et al. (2013) have been used to investigate efficiency of potential surveillance strategies of bTB [4,47] and control strategies of FMD [18]. Here we show that the functional form used by Lindström et al. is the least preferred. This raises an important question: are conclusions based on previous shipment predictions unreliable? Large-scale predictions were shown here to be similar across all kernel functional forms in terms of the proportion of shipment moving intrastate (Figure 3) as well as number of incoming and outgoing shipments per state (Figure 4). Figure 5 further shows that even though kernel two and three are better predictors of observed shipment distances than kernel one, simulated shipment distances are typically similar. Thus, we argue that studies based on previous versions are not invalid. However, the improvements made in this study are substantial, and we advise future studies that rely on estimates of U.S. cattle shipments to use these updated predictions. For this purpose, we provide 1000 realizations of shipments with kernel three in the supplement S8.

To further inform the choice of kernel functional form, we included expert opinion data in the analysis for states where these were available. Expert elicitation questions must be formulated so that they are clear to all the targeted experts. In the survey used here, questions were designed to

Figure 9. Distance distributions of movement distances in ICVI data of 2009, 2010, and 2011 for nine states and 95% predictive band of model of movement distances predicted by USAMM (kernel three, including experts).

promote inclusion of experts with varying levels of statistical expertise, and it was not possible to elicit information about the four state level parameters (rate λ_S , propensity to attract shipments W_S , scale d_S , and shape R_S). Instead we used responses about quantities experts could have an intuitive sense about and recalculated responses into quantities that could be included in the statistical model, specifically the proportion of shipments moving interstate. The inclusion of experts in the analysis did not alter the ranking of kernel functional forms (Table 2) and had minimal effect on estimates (Figure 6, Figure 7). The reason is a mixture of two properties of the data. First, the expert responses were disparate, leading to low precision (ξ_s) of the logit-normal distribution modeling expert responses. Consequently, the contribution of the experts to the posterior was low. Secondly, in instances where expert estimates were less disparate, such as for Texas beef shipments, the estimates coincided with the estimates from the ICVI data only. Thus, we can at least conclude that the estimates of the statistical model for ICVI data does not contradict the collective expectations of the experts. The disparate answers from the included experts further demonstrates that intuitive expectations about basic shipment patterns in the U.S. are challenging, even for insightful practitioners. This highlights the need for quantitative approaches such as those presented in this study.

With the improvements to the existing framework, the analysis recaptured essential state-level features of the training data (Figure 4, Figure 5, Figure 7) as well as validation data (Figure 8). Figure 5 and Figure 9 further illustrate that USAMM predictions produce shipment distances comparable to the observed shipments. However, there are instances where the posterior predictive bands do not overlap the observed distribution curves. The reason is likely because of county level differences beyond number of premises that lead to a high number of incoming

and/or outgoing shipments for specific counties. Future developments of the model could aim at identifying industry structures that produce these differences and thereby improve estimates at the county scale.

Nevertheless, we believe the spatially explicit model structure of our model is an appropriate framework to build on. Other methods of inferring links in incomplete networks typically rely on expectations of the nodes [48] or network structure [49,50], neither of which are readily available for the considered system. The spatial component is essential for most topics where estimates of cattle shipments are of interest, such as investigations of surveillance, control strategies, and disease preparedness.

The advancement here focused on state level differences in shipment patterns and revealed substantial differences in shipment patterns across the U.S. Median estimates of per premises rate of shipments (λ) and propensity to receive shipments (W) varied by more than an order of magnitude between states (Figure 2), a pattern that was consistent across models and data. Similarly, the kernel parameter varied substantially among states, indicating that a single set of parameters is insufficient to recapture the shipment pattern of a heterogeneous livestock industry. This is consistent with the findings of Brommesson et al. [16], who found that parameters of the spatial kernel describing Swedish cattle shipments varied geographically. These considerations are of even greater importance when considering the highly heterogeneous structure of the U.S. cattle industry.

The estimates provided by this study currently offer the most reliable depiction of national scale cattle shipments in the U.S., accounting for state level differences in the industry structure. For the first time, we were able to validate the predictions with out-of-sample data from additional years and found that the broad predictions fit well with the observations. To make our findings available for the broad range of research that rely on estimates of cattle shipments in the U.S., we make realizations of county level shipment networks available (see supplementary material). We expect these will benefit future agency planning efforts and studies of infectious livestock diseases where shipment of animals constitutes an important route of transmission.

Data Accessibility. We make available the code to allow a replication of our study, as well as indicate in the manuscript sources of supporting data (for instance, expert opinion data [28]). In addition to the code, the study required the use of three types of data/input; namely, Interstate Certificate of Veterinary Inspection (ICVI) data, FLAPS model predictions of farm densities based on data from the USDA National Agricultural Statistics Service, and expert opinion estimates. The underlying data for county-level densities is accessible via the NASS homepage, and we share the FLAPS predictions for county-level totals in the cases where NASS has not published data, and the aforementioned expert opinion data. These come from a previously published survey study with experts from several US states. We use all respondent data, but do not disclose individual expert responses for states with two or less respondents. This is because the premise of the survey was that individual experts' responses wouldn't be disclosed. To the extent that they can be shared, data is already available. The ICVI datasets analyzed during this study are not publically available. Requests for these data may be available from USDA authors, Dr. Katie Portacci (Katie.Portacci@aphis.usda.gov) and Dr. Ryan Miller (Ryan.S.Miller@aphis.usda.gov) upon reasonable request, in compliance with Federal regulations, and under agreements with the United States Department of Agriculture.

Authors' Contributions. Conceived and designed the experiments: P.B., C.T.W., T.L. Collection and curation of data: L.B.-J., C.H., D.M., R.S.M., K.P. Implementation: P.B., S.S. Visualization: P.B., C.H. Analysis: P.B., S.S., L.B.-J., C.H., D.M., R.S.M., C.T.W., T.L. All authors contributed in writing the paper and gave final approval for publication.

Competing Interests. The authors declare they have no competing interests.

Funding. This work is supported by funding provided by the U.S. Department of Homeland Security Science and Technology Directorate under contract number HSHQDC-13-C-B0028.

1
2
3
4
5
6
7
8
9
10
11
12
13
14
15
16
17
18
19
20
21
22
23
24
25
26
27
28
29
30
31
32
33
34
35
36
37
38
39
40
41
42
43
44
45
46
47
48
49
50
51
52
53
54
55
56
57
58
59
60

Acknowledgements. All analyses were performed on resources provided by the Swedish National Infrastructure for Computing (SNIC).

rsos.royalsocietypublishing.org R. Soc. open sci. 0000000
.....

References

1. Tälle M, Wiréhn L, Ellström D, Hjerpe M, Hüge-Brodin M, Jensen P, Lindström T, Neset TS, Wennergren U, Metson G. 2019 Synergies and Trade-Offs for Sustainable Food Production in Sweden: An Integrated Approach. *Sustainability* **11**, 601.
2. Manual on the preparation of national animal disease emergency preparedness plans. .
3. Fèvre EM, Bronsvoort BMdC, Hamilton KA, Cleaveland S. 2006 Animal movements and the spread of infectious diseases. *Trends in Microbiology* **14**, 125–131.
4. Gorsich EE, McKee CD, Gear DA, Miller RS, Portacci K, Lindström T, Webb CT. 2018 Model-guided suggestions for targeted surveillance based on cattle shipments in the U.S.. *Preventive Veterinary Medicine* **150**, 52–59.
5. Gibbens JC, Wilesmith JW, Sharpe CE, Mansley LM, Michalopoulou E, Ryan JBM, Hudson M. 2001 Descriptive epidemiology of the 2001 foot-and-mouth disease epidemic in Great Britain: the first five months. *Veterinary Record* **149**, 729–743.
6. Meat Animals Production, Disposition, and Income - Final Estimates. .
7. The United States Meat Industry at a Glance. .
8. USDA - National Agricultural Statistics Service - Pennsylvania - Survey Results. .
9. USDA - NASS, Census of Agriculture - Publications - 2012. .
10. Paarlberg PL, Lee JG, Seitzinger AH. 2002 Potential revenue impact of an outbreak of foot-and-mouth disease in the United States. *Journal of the American Veterinary Medical Association* **220**, 988–992.
11. Okafor CC, Grooms DL, Bruning-Fann CS, Averill JJ, Kaneene JB. 2011 Descriptive Epidemiology of Bovine Tuberculosis in Michigan (1975–2010): Lessons Learned. .
12. Brooks-Pollock E, de Jong MCM, Keeling MJ, Klinkenberg D, Wood JLN. 2015 Eight challenges in modelling infectious livestock diseases. *Epidemics* **10**, 1–5.
13. . 2000 Regulation (EC) No 1760/2000 of the European Parliament and of the Council of 17 July 2000 establishing a system for the identification and registration of bovine animals and regarding the labelling of beef and beef products and repealing Council Regulation (EC) No 820/97. *OFFICIAL JOURNAL- EUROPEAN COMMUNITIES LEGISLATION L* **43**, 1–10.
14. Anderson DP. 2010 The U.S. Animal Identification Experience. *Journal of Agricultural and Applied Economics* **42**, 543–550.
15. Buhnerkempe MG, Gear DA, Portacci K, Miller RS, Lombard JE, Webb CT. 2013 A national-scale picture of U.S. cattle movements obtained from Interstate Certificate of Veterinary Inspection data. *Preventive Veterinary Medicine* **112**, 318–329.
16. Brommesson P, Wennergren U, Lindström T. 2016 Spatiotemporal Variation in Distance Dependent Animal Movement Contacts: One Size Doesn't Fit All. *PLOS ONE* **11**, e0164008.
17. Lindström T, Gear DA, Buhnerkempe M, Webb CT, Miller RS, Portacci K, Wennergren U. 2013 A Bayesian Approach for Modeling Cattle Movements in the United States: Scaling up a Partially Observed Network. *PLoS ONE* **8**, e53432.
18. Buhnerkempe MG, Tildesley MJ, Lindström T, Gear DA, Portacci K, Miller RS, Lombard JE, Werkman M, Keeling MJ, Wennergren U, Webb CT. 2014 The Impact of Movements and Animal Density on Continental Scale Cattle Disease Outbreaks in the United States. *PLoS ONE* **9**, e91724.
19. Garabed RB, Perez AM, Johnson WO, Thurmond MC. 2009 Use of expert opinion for animal disease decisions: An example of foot-and-mouth disease status designation. *Preventive Veterinary Medicine* **92**, 20–30.
20. Horst HS, Dijkhuizen AA, Huirne RBM, De Leeuw PW. 1998 Introduction of contagious animal diseases into The Netherlands: elicitation of expert opinions. *Livestock Production Science* **53**, 253–264.
21. Albert I, Donnet S, Guihenneuc-Jouyaux C, Low-Choy S, Mengersen K, Rousseau J. 2012 Combining Expert Opinions in Prior Elicitation. *Bayesian Analysis* **7**, 503–532.
22. Martin TG, Kuhnert PM, Mengersen K, Possingham HP. 2005 The Power of Expert Opinion in Ecological Models Using Bayesian Methods: Impact of Grazing on Birds. *Ecological Applications* **15**, 266–280.

23. Choy SL, O'Leary R, Mengersen K. 2009 Elicitation by design in ecology: using expert opinion to inform priors for Bayesian statistical models. *Ecology* **90**, 265–277.
24. Swart A, Ibañez-Justicia A, Buijs J, van Wieren SE, Hofmeester TR, Sprong H, Takumi K. 2014 Predicting Tick Presence by Environmental Risk Mapping. *Frontiers in Public Health* **2**.
25. Portacci K, Miller RS, Riggs PD, Buhnerkempe MG, Abrahamsen LM. 2013 Assessment of paper interstate certificates of veterinary inspection used to support disease tracing in cattle. *Journal of the American Veterinary Medical Association* **243**, 555–560.
26. Burdett CL, Kraus BR, Garza SJ, Miller RS, Bjork KE. 2015 Simulating the Distribution of Individual Livestock Farms and Their Populations in the United States: An Example Using Domestic Swine (*Sus scrofa domestica*) Farms. *PLOS ONE* **10**, e0140338.
27. USDA/NASS QuickStats Ad-hoc Query Tool. .
28. Beck-Johnson LM, Hallman C, Miller RS, Portacci K, Gorsich EE, Grear DA, Hartmann K, Webb CT. 2019 Estimating and exploring the proportions of inter- and intrastate cattle shipments in the United States. *Preventive Veterinary Medicine* **162**, 56–66.
29. Nadarajah S. 2005 A generalized normal distribution. *Journal of Applied Statistics* **32**, 685–694.
30. Hastings WK. 1970 Monte Carlo Sampling Methods Using Markov Chains and Their Applications. *Biometrika* **57**, 97–109.
31. Garthwaite PH, Fan Y, Sisson SA. 2010 Adaptive Optimal Scaling of Metropolis-Hastings Algorithms Using the Robbins-Monro Process. *arXiv:1006.3690 [stat]*. arXiv: 1006.3690.
32. Roberts GO, Gelman A, Gilks WR. 1997 Weak convergence and optimal scaling of random walk Metropolis algorithms. *The Annals of Applied Probability* **7**, 110–120.
33. Gelman A, Carlin J, Stern H, Rubin D. 2003 *Bayesian Data Analysis, Second Edition*. Chapman & Hall/CRC Texts in Statistical Science. Taylor & Francis.
34. Brooks SP, Gelman A. 1998 General Methods for Monitoring Convergence of Iterative Simulations. *Journal of Computational and Graphical Statistics* **7**, 434–455.
35. Spiegelhalter DJ, Best NG, Carlin BP, Van Der Linde A. 2002 Bayesian measures of model complexity and fit. *Journal of the Royal Statistical Society: Series B (Statistical Methodology)* **64**, 583–639.
36. Watanabe S. 2010 Asymptotic equivalence of Bayes cross validation and widely applicable information criterion in singular learning theory. *Journal of Machine Learning Research* **11**, 3571–3594. 3571.
37. Gelman A, Hwang J, Vehtari A. 2014 Understanding predictive information criteria for Bayesian models. *Statistics and Computing* **24**, 997–1016.
38. Plummer M. 2008 Penalized loss functions for Bayesian model comparison. *Biostatistics* **9**, 523–539.
39. Lindström T, Sisson SA, Lewerin SS, Wennergren U. 2011 Bayesian analysis of animal movements related to factors at herd and between herd levels: Implications for disease spread modeling. *Preventive Veterinary Medicine* **98**, 230–242.
40. Westergaard JM. 2008 Contingency Planning: Preparation of Contingency Plans. *Zoonoses and Public Health* **55**, 42–49.
41. Nöremark M, Frössling J, Lewerin SS. 2013 A survey of visitors on Swedish livestock farms with reference to the spread of animal diseases. *BMC Veterinary Research* **9**, 184.
42. Ribeiro-Lima J, Enns EA, Thompson B, Craft ME, Wells SJ. 2015 From network analysis to risk analysis—An approach to risk-based surveillance for bovine tuberculosis in Minnesota, US. *Preventive Veterinary Medicine* **118**, 328–340.
43. Ferguson NM, Donnelly CA, Anderson RM. 2001 Transmission intensity and impact of control policies on the foot and mouth epidemic in Great Britain. *Nature* **413**, 542–548.
44. Natale F, Giovannini A, Savini L, Palma D, Possenti L, Fiore G, Calistri P. 2009 Network analysis of Italian cattle trade patterns and evaluation of risks for potential disease spread. *Preventive Veterinary Medicine* **92**, 341–350.
45. (PDF) Spatial analysis of cattle movement patterns in Portugal. .
46. Keeling M, Woolhouse M, Shaw D, Matthews L, Chase-Topping M, Haydon D, Cornell S, Kappey J, Wilesmith J, Grenfell B. 2001 Dynamics of the 2001 UK foot and mouth epidemic: Stochastic dispersal in a heterogeneous landscape. *Science* **294**, 813–817. 813.

47. Kao SYZ, VanderWaal K, Enns EA, Craft ME, Alvarez J, Picasso C, Wells SJ. 2018 Modeling cost-effectiveness of risk-based bovine tuberculosis surveillance in Minnesota. *Preventive Veterinary Medicine* **159**, 1–11.

48. Guimerà R, Sales-Pardo M. 2009 Missing and spurious interactions and the reconstruction of complex networks. *Proceedings of the National Academy of Sciences* **106**, 22073–22078.

49. Håkansson N, Jonsson A, Lennartsson J, Lindström T, Wennergren U. 2010 Generating structure specific networks. *Advances in Complex Systems* **13**, 239–250.

50. Gates MC, Woolhouse MEJ. 2015 Controlling infectious disease through the targeted manipulation of contact network structure. *Epidemics* **12**, 11–19.

.....rsos.royalsocietypublishing.org R. Soc. open sci. 0000000

1
2
3
4
5
6
7
8
9
10
11
12
13
14
15
16
17
18
19
20
21
22
23
24
25
26
27
28
29
30
31
32
33
34
35
36
37
38
39
40
41
42
43
44
45
46
47
48
49
50
51
52
53
54
55
56
57
58
59
60

5. Supplementary

(a) S1 Marginal posterior estimates of d , R , \hat{W} and $\hat{\lambda}$

(i) S1.1 Estimates including expert data

Figure 10. Marginal posterior estimates obtained by using kernel one, indicated by their median (dots) and 95% credibility interval (error bars) of state level model parameters for analysis of beef ICVI data, including experts for available states.

rsos.royalsocietypublishing.org R. Soc. open sci. 0000000

rsos.royalsocietypublishing.org R. Soc. open sci. 0000000

Figure 11. Marginal posterior estimates obtained by using kernel two, indicated by their median (dots) and 95% credibility interval (error bars) of state level model parameters for analysis of beef ICVI data, including experts for available states.

30

rsos.royalsocietypublishing.org R. Soc. open sci. 0000000

Figure 12. Marginal posterior estimates obtained by using kernel three, indicated by their median (dots) and 95% credibility interval (error bars) of state level model parameters for analysis of beef ICVI data, including experts for available states.

Figure 13. Marginal posterior estimates obtained by using kernel one, indicated by their median (dots) and 95% credibility interval (error bars) of state level model parameters for analysis of dairy ICVI data, including experts for available states.

Figure 14. Marginal posterior estimates obtained by using kernel two, indicated by their median (dots) and 95% credibility interval (error bars) of state level model parameters for analysis of dairy ICVI data, including experts for available states.

Figure 15. Marginal posterior estimates obtained by using kernel three, indicated by their median (dots) and 95% credibility interval (error bars) of state level model parameters for analysis of dairy ICVI data, including experts for available states.

Figure 16. Marginal posterior estimates obtained by using kernel one, indicated by their median (dots) and 95% credibility interval (error bars) of state level model parameters for analysis of beef ICVI data, excluding experts for available states.

rsos.royalsocietypublishing.org R. Soc. open sci. 0000000

Figure 17. Marginal posterior estimates obtained by using kernel two, indicated by their median (dots) and 95% credibility interval (error bars) of state level model parameters for analysis of beef ICVI data, excluding experts for available states.

Figure 18. Marginal posterior estimates obtained by using kernel three, indicated by their median (dots) and 95% credibility interval (error bars) of state level model parameters for analysis of beef ICVI data, excluding experts for available states.

rsos.royalsocietypublishing.org R. Soc. open sci. 0000000

Figure 19. Marginal posterior estimates obtained by using kernel one, indicated by their median (dots) and 95% credibility interval (error bars) of state level model parameters for analysis of dairy ICVI data, excluding experts for available states.

Figure 20. Marginal posterior estimates obtained by using kernel two, indicated by their median (dots) and 95% credibility interval (error bars) of state level model parameters for analysis of dairy ICVI data, excluding experts for available states.

Figure 21. Marginal posterior estimates obtained by using kernel three, indicated by their median (dots) and 95% credibility interval (error bars) of state level model parameters for analysis of dairy ICVI data, excluding experts for available states.

(b) S2 Marginal posterior estimates of expert precision parameter ξ

Figure 22. Marginal posterior estimates indicated by their median (dots) and 95% credibility interval (error bars) of state level (where applicable) expert precision parameter (ξ), for models using expert data only and expert data combined with ICVI data and three different kernels. Estimates for beef are shown in top row and estimates for dairy in the bottom row.

rsos.royalsocietypublishing.org R. Soc. open sci. 0000000

(c) S3 Predicted proportion of interstate shipments excluding experts

rsos.royalsocietypublishing.org R. Soc. open sci. 0000000

Figure 23. Predicted proportion of shipments moving interstate for beef and dairy combined based on three different functional forms of the spatial kernel. Results are shown for analyses excluding expert data.

(d) S4 Number of in- and outgoing shipments

Figure 24. Predicted proportion of shipments moving interstate for beef and dairy combined based on three different functional forms of the spatial kernel. Results are shown for analyses excluding expert data.

(e) S5 Proportion interstate shipments

Figure 25. Median and 95% credibility interval of posterior distributions of proportion of shipments moving interstate for analysis including only experts (red), ICVI data only (blue) and ICVI and experts together (green). Results are shown for analyses for beef.

rsos.royalsocietypublishing.org R. Soc. open sci. 0000000

Figure 26. Median and 95% credibility interval of posterior distributions of proportion of shipments moving interstate for analysis including only experts (red), ICVI data only (blue) and ICVI and experts together (green). Results are shown for analyses for dairy.

(f) S6 Sensitivity analysis

Hyperparameters whose marginal posteriors that had more than 1% density of either tail outside the interval of the hyperprior containing 95% of the density were subject for further investigation in the sensitivity analysis. If the first percentile of the marginal posterior distribution was lower than the 2.5th percentile of the hyperprior, we investigated the effect of the elicited hyperprior by decreasing the lower bound of the 95% interval by which it was defined by 50%. Analogously, we adjusted the hyperprior if the 99th percentile of the marginal posterior was greater than the 97.5 percentile by doubling this upper bound. That is, the hyperpriors were made wider to be less restrictive. Figures of parameters subject to this modification and figures of the effect on the other parameters are shown below.

(i) S6.1 Marginal posterior estimates including expert data

45

rsos.royalsocietypublishing.org R. Soc. open sci. 0000000

Figure 27. Solid lines show scaled density distributions of m_d using original hyperprior (red) and wide hyperprior (blue). Results are shown for analysis using beef data including expert opinions and using kernel one. Dashed lines show the corresponding hyperprior used (scaled so the maximum value attained by any of the hyperpriors within the given parameter values is equal to one). x-axis is truncated at 1000 and does not show the entire tail of the density distributions

Figure 28. Solid lines show scaled density distributions of cv_d using original hyperprior (red) and wide hyperprior (blue). Results are shown for analysis using beef data including expert opinions and using kernel one. Dashed lines show the corresponding hyperprior used (scaled so the maximum value attained by any of the hyperpriors within the given parameter values is equal to one). Note that hyperprior distributions (dashed lines) have most of their density at small values of cv_d , and therefore appear as vertical lines at zero.

Figure 29. Solid lines show scaled density distributions of m_R using original hyperprior (red) and wide hyperprior (blue). Results are shown for analysis using beef data including expert opinions and using kernel one. Dashed lines show the corresponding hyperprior used (scaled so the maximum value attained by any of the hyperpriors within the given parameter values is equal to one).

Figure 30. Solid lines show scaled density distributions of cv_R using original hyperprior (red) and wide hyperprior (blue). Results are shown for analysis using beef data including expert opinions and using kernel one. Dashed lines show the corresponding hyperprior used (scaled so the maximum value attained by any of the hyperpriors within the given parameter values is equal to one).

Figure 31. Marginal posterior estimates obtained by using kernel one, indicated by their median (dots) and 95% credibility interval (error bars) of state level model parameters for analysis of beef ICVI data, including experts for available states. Estimates using original and wide hyperpriors are shown in red and blue, respectively.

Figure 32. Solid lines show scaled density distributions of cv_d using original hyperprior (red) and wide hyperprior (blue). Results are shown for analysis using beef data including expert opinions and using kernel two. Dashed lines show the corresponding hyperprior used (scaled so the maximum value attained by any of the hyperpriors within the given parameter values is equal to one).

Figure 33. Solid lines show scaled density distributions of cv_R using original hyperprior (red) and wide hyperprior (blue). Results are shown for analysis using beef data including expert opinions and using kernel two. Dashed lines show the corresponding hyperprior used (scaled so the maximum value attained by any of the hyperpriors within the given parameter values is equal to one).

Figure 34. Marginal posterior estimates obtained by using kernel two, indicated by their median (dots) and 95% credibility interval (error bars) of state level model parameters for analysis of beef ICVI data, including experts for available states. Estimates using original and wide hyperpriors are shown in red and blue, respectively.

Figure 35. Solid lines show scaled density distributions of cv_d using original hyperprior (red) and wide hyperprior (blue). Results are shown for analysis using beef data including expert opinions and using kernel three. Dashed lines show the corresponding hyperprior used (scaled so the maximum value attained by any of the hyperpriors within the given parameter values is equal to one).

Figure 36. Solid lines show scaled density distributions of cv_R using original hyperprior (red) and wide hyperprior (blue). Results are shown for analysis using beef data including expert opinions and using kernel three. Dashed lines show the corresponding hyperprior used (scaled so the maximum value attained by any of the hyperpriors within the given parameter values is equal to one).

Figure 37. Marginal posterior estimates obtained by using kernel three, indicated by their median (dots) and 95% credibility interval (error bars) of state level model parameters for analysis of beef ICVI data, including experts for available states. Estimates using original and wide hyperpriors are shown in red and blue, respectively.

Figure 38. Solid lines show scaled density distributions of m_d using original hyperprior (red) and wide hyperprior (blue). Results are shown for analysis using dairy data including expert opinions and using kernel one. Dashed lines show the corresponding hyperprior used (scaled so the maximum value attained by any of the hyperpriors within the given parameter values is equal to one).

Figure 39. Solid lines show scaled density distributions of cv_d using original hyperprior (red) and wide hyperprior (blue). Results are shown for analysis using dairy data including expert opinions and using kernel one. Dashed lines show the corresponding hyperprior used (scaled so the maximum value attained by any of the hyperpriors within the given parameter values is equal to one).

Figure 40. Solid lines show scaled density distributions of m_R using original hyperprior (red) and wide hyperprior (blue). Results are shown for analysis using dairy data including expert opinions and using kernel one. Dashed lines show the corresponding hyperprior used (scaled so the maximum value attained by any of the hyperpriors within the given parameter values is equal to one).

Figure 41. Solid lines show scaled density distributions of CV_R using original hyperprior (red) and wide hyperprior (blue). Results are shown for analysis using dairy data including expert opinions and using kernel one. Dashed lines show the corresponding hyperprior used (scaled so the maximum value attained by any of the hyperpriors within the given parameter values is equal to one).

Figure 42. Solid lines show scaled density distributions of cv_{ξ} using original hyperprior (red) and wide hyperprior (blue). Results are shown for analysis using dairy data including expert opinions and using kernel one. Dashed lines show the corresponding hyperprior used (scaled so the maximum value attained by any of the hyperpriors within the given parameter values is equal to one).

Figure 43. Marginal posterior estimates obtained by using kernel one, indicated by their median (dots) and 95% credibility interval (error bars) of state level model parameters for analysis of dairy ICVI data, including experts for available states. Estimates using original and wide hyperpriors are shown in red and blue, respectively.

Figure 44. Solid lines show scaled density distributions of cv_d using original hyperprior (red) and wide hyperprior (blue). Results are shown for analysis using dairy data including expert opinions and using kernel two. Dashed lines show the corresponding hyperprior used (scaled so the maximum value attained by any of the hyperpriors within the given parameter values is equal to one).

Figure 45. Solid lines show scaled density distributions of m_R using original hyperprior (red) and wide hyperprior (blue). Results are shown for analysis using dairy data including expert opinions and using kernel two. Dashed lines show the corresponding hyperprior used (scaled so the maximum value attained by any of the hyperpriors within the given parameter values is equal to one).

Figure 46. Solid lines show scaled density distributions of cv_{ξ} using original hyperprior (red) and wide hyperprior (blue). Results are shown for analysis using dairy data including expert opinions and using kernel two. Dashed lines show the corresponding hyperprior used (scaled so the maximum value attained by any of the hyperpriors within the given parameter values is equal to one).

Figure 47. Marginal posterior estimates obtained by using kernel two, indicated by their median (dots) and 95% credibility interval (error bars) of state level model parameters for analysis of dairy ICVI data, including experts for available states. Estimates using original and wide hyperpriors are shown in red and blue, respectively.

Figure 48. Solid lines show scaled density distributions of cv_d using original hyperprior (red) and wide hyperprior (blue). Results are shown for analysis using dairy data including expert opinions and using kernel three. Dashed lines show the corresponding hyperprior used (scaled so the maximum value attained by any of the hyperpriors within the given parameter values is equal to one).

Figure 49. Solid lines show scaled density distributions of cv_{ξ} using original hyperprior (red) and wide hyperprior (blue). Results are shown for analysis using dairy data including expert opinions and using kernel three. Dashed lines show the corresponding hyperprior used (scaled so the maximum value attained by any of the hyperpriors within the given parameter values is equal to one).

Figure 50. Marginal posterior estimates obtained by using kernel three, indicated by their median (dots) and 95% credibility interval (error bars) of state level model parameters for analysis of dairy ICVI data, including experts for available states. Estimates using original and wide hyperpriors are shown in red and blue, respectively.

(ii) S6.2 Marginal posterior estimates excluding expert data

Figure 51. Solid lines show scaled density distributions of m_d using original hyperprior (red) and wide hyperprior (blue). Results are shown for analysis using beef data excluding expert opinions and using kernel one. Dashed lines show the corresponding hyperprior used (scaled so the maximum value attained by any of the hyperpriors within the given parameter values is equal to one). x-axis is truncated at 1000 and does not show the entire tail of the density distributions.

rsos.royalsocietypublishing.org R. Soc. open sci. 0000000

70
rsos.royalsocietypublishing.org R. Soc. open sci. 0000000

Figure 52. Solid lines show scaled density distributions of cv_d using original hyperprior (red) and wide hyperprior (blue). Results are shown for analysis using beef data excluding expert opinions and using kernel one. Dashed lines show the corresponding hyperprior used (scaled so the maximum value attained by any of the hyperpriors within the given parameter values is equal to one). Note that hyperprior distributions (dashed lines) have most of their density at small values of cv_d , and therefore appear as vertical lines at zero.

Figure 53. Solid lines show scaled density distributions of m_R using original hyperprior (red) and wide hyperprior (blue). Results are shown for analysis using beef data excluding expert opinions and using kernel one. Dashed lines show the corresponding hyperprior used (scaled so the maximum value attained by any of the hyperpriors within the given parameter values is equal to one).

Figure 54. Solid lines show scaled density distributions of cv_R using original hyperprior (red) and wide hyperprior (blue). Results are shown for analysis using beef data excluding expert opinions and using kernel one. Dashed lines show the corresponding hyperprior used (scaled so the maximum value attained by any of the hyperpriors within the given parameter values is equal to one).

Figure 55. Marginal posterior estimates obtained by using kernel one, indicated by their median (dots) and 95% credibility interval (error bars) of state level model parameters for analysis of beef ICVI data, excluding experts for available states. Estimates using original and wide hyperpriors are shown in red and blue, respectively.

Figure 56. Solid lines show scaled density distributions of cv_d using original hyperprior (red) and wide hyperprior (blue). Results are shown for analysis using beef data excluding expert opinions and using kernel two. Dashed lines show the corresponding hyperprior used (scaled so the maximum value attained by any of the hyperpriors within the given parameter values is equal to one).

Figure 57. Solid lines show scaled density distributions of cv_R using original hyperprior (red) and wide hyperprior (blue). Results are shown for analysis using beef data excluding expert opinions and using kernel two. Dashed lines show the corresponding hyperprior used (scaled so the maximum value attained by any of the hyperpriors within the given parameter values is equal to one).

Figure 58. Marginal posterior estimates obtained by using kernel two, indicated by their median (dots) and 95% credibility interval (error bars) of state level model parameters for analysis of beef ICVI data, excluding experts for available states. Estimates using original and wide hyperpriors are shown in red and blue, respectively.

Figure 59. Solid lines show scaled density distributions of cv_d using original hyperprior (red) and wide hyperprior (blue). Results are shown for analysis using beef data excluding expert opinions and using kernel three. Dashed lines show the corresponding hyperprior used (scaled so the maximum value attained by any of the hyperpriors within the given parameter values is equal to one).

Figure 60. Solid lines show scaled density distributions of cv_R using original hyperprior (red) and wide hyperprior (blue). Results are shown for analysis using beef data excluding expert opinions and using kernel three. Dashed lines show the corresponding hyperprior used (scaled so the maximum value attained by any of the hyperpriors within the given parameter values is equal to one).

Figure 61. Marginal posterior estimates obtained by using kernel three, indicated by their median (dots) and 95% credibility interval (error bars) of state level model parameters for analysis of beef ICVI data, excluding experts for available states. Estimates using original and wide hyperpriors are shown in red and blue, respectively.

Figure 62. Solid lines show scaled density distributions of m_d using original hyperprior (red) and wide hyperprior (blue). Results are shown for analysis using dairy data excluding expert opinions and using kernel one. Dashed lines show the corresponding hyperprior used (scaled so the maximum value attained by any of the hyperpriors within the given parameter values is equal to one).

Figure 63. Solid lines show scaled density distributions of cv_d using original hyperprior (red) and wide hyperprior (blue). Results are shown for analysis using dairy data excluding expert opinions and using kernel one. Dashed lines show the corresponding hyperprior used (scaled so the maximum value attained by any of the hyperpriors within the given parameter values is equal to one).

Figure 64. Solid lines show scaled density distributions of m_R using original hyperprior (red) and wide hyperprior (blue). Results are shown for analysis using dairy data excluding expert opinions and using kernel one. Dashed lines show the corresponding hyperprior used (scaled so the maximum value attained by any of the hyperpriors within the given parameter values is equal to one).

Figure 65. Solid lines show scaled density distributions of cv_R using original hyperprior (red) and wide hyperprior (blue). Results are shown for analysis using dairy data excluding expert opinions and using kernel one. Dashed lines show the corresponding hyperprior used (scaled so the maximum value attained by any of the hyperpriors within the given parameter values is equal to one).

84
rsos.royalsocietypublishing.org R. Soc. open sci. 0000000

Figure 66. Marginal posterior estimates obtained by using kernel one, indicated by their median (dots) and 95% credibility interval (error bars) of state level model parameters for analysis of dairy ICVI data, excluding experts for available states. Estimates using original and wide hyperpriors are shown in red and blue, respectively.

Figure 67. Solid lines show scaled density distributions of cv_d using original hyperprior (red) and wide hyperprior (blue). Results are shown for analysis using dairy data excluding expert opinions and using kernel two. Dashed lines show the corresponding hyperprior used (scaled so the maximum value attained by any of the hyperpriors within the given parameter values is equal to one).

Figure 68. Solid lines show scaled density distributions of m_R using original hyperprior (red) and wide hyperprior (blue). Results are shown for analysis using dairy data excluding expert opinions and using kernel two. Dashed lines show the corresponding hyperprior used (scaled so the maximum value attained by any of the hyperpriors within the given parameter values is equal to one).

Figure 69. Marginal posterior estimates obtained by using kernel two, indicated by their median (dots) and 95% credibility interval (error bars) of state level model parameters for analysis of dairy ICVI data, excluding experts for available states. Estimates using original and wide hyperpriors are shown in red and blue, respectively.

88

rsos.royalsocietypublishing.org R. Soc. open sci. 0000000

Figure 70. Solid lines show scaled density distributions of cv_d using original hyperprior (red) and wide hyperprior (blue). Results are shown for analysis using dairy data excluding expert opinions and using kernel three. Dashed lines show the corresponding hyperprior used (scaled so the maximum value attained by any of the hyperpriors within the given parameter values is equal to one).

Figure 71. Marginal posterior estimates obtained by using kernel three, indicated by their median (dots) and 95% credibility interval (error bars) of state level model parameters for analysis of dairy ICVI data, excluding experts for available states. Estimates using original and wide hyperpriors are shown in red and blue, respectively.

(iii) S6.3 Model selection

Table 3 shows that the ranking of the models when using the hyper prior from the sensitivity analysis is overall consistent with the findings using original hyper prior. The only alternation in ranking was in the WAIC scores for beef data using ICI and Expert data. In that case, kernel two and three had similar ranking with negligible difference in WAIC scores, whereas kernel three was identified with highest WAIC score in the original analysis. Kernel one showed in this case, as in the original analysis, lowest WAIC score.

(g) S7 Distance distributions

Figures below show distance distributions of generated interstate networks down sampled to 10% for beef and dairy combined. Top row show results based on models including expert data and bottom row excluding expert data. The different kernels are depicted column wise. Distance

Commodity	Data	Kernel	Δ WAIC		Δ DIC		Minimum number of independent samples	
			Main analysis	Sensitivity Analysis	Main analysis	Sensitivity Analysis	Main analysis	Sensitivity Analysis
Beef	ICVI	1	625.0	462.1	630.1	469.1	6873 (W VT)	6873 (R AZ)
		2	14.2	12.3	11.5	17.3	2445 (d OH)	5318 (R OH)
		3	0.0	0.0	0.0	0.0	3805 (d OH)	1971 (R OH)
	ICVI+ Experts	1	626.4	485.8	630.1	496.2	9505 (W OH)	2114 (R AZ)
		2	21.0	0.0	10.4	21.5	2348 (λ OH)	2813 (d OH)
		3	0.0	0.2	0.0	0.0	6071 (R FL)	1733 (d OH)
Dairy	ICVI	1	530.0	524.5	531.0	522.3	4724 (σ_d)	6203 (W WY)
		2	0.0	0.0	0.0	0.0	4876 (W IN)	6408 (W VT)
		3	45.2	45.2	45.6	44.1	6050 (W IA)	8453 (W IN)
	ICVI+ Experts	1	510.0	506.1	512.1	502.0	6631 (W WI)	6118 (σ_R)
		2	0.0	0.0	0.0	0.0	6792 (W OR)	9052 (W AR)
		3	45.3	45.7	46.0	43.1	11047 (W MT)	5966 (W NH)

Table 3. Differences from the preferred kernel functional form for two model selection criteria (Δ WAIC and Δ DIC) for each considered data set. The minimum number of independent samples includes in parenthesis for which (hyper)parameter (and state where applicable) the minimum value was estimated for.

distributions of movement distances in ICVI data of 2009, 2010, and 2011 (where available) are depicted as red, green and blue lines, respectively.

rsos.royalsocietypublishing.org R. Soc. open sci. 0000000

Figure 72. Distance distributions of generated interstate networks down sampled to 10% for beef and dairy combined. Top row shows results based on models including expert data and bottom row shows results excluding expert data. The different kernels are depicted column wise. Distance distributions of movement distances in ICVI data of 2009, 2010, and 2011 are depicted as red, green and blue lines, respectively.

rsos.royalsocietypublishing.org R. Soc. open sci. 0000000

Figure 73. Distance distributions of generated interstate networks down sampled to 10% for beef and dairy combined. Top row shows results based on models including expert data and bottom row shows results excluding expert data. The different kernels are depicted column wise.

rsos.royalsocietypublishing.org R. Soc. open sci. 0000000

Figure 74. Distance distributions of generated interstate networks down sampled to 10% for beef and dairy combined. Top row shows results based on models including expert data and bottom row shows results excluding expert data. The different kernels are depicted column wise.

Figure 75. Distance distributions of generated interstate networks down sampled to 10% for beef and dairy combined. Top row shows results based on models including expert data and bottom row shows results excluding expert data. The different kernels are depicted column wise.

Figure 76. Distance distributions of generated interstate networks down sampled to 10% for beef and dairy combined. Top row shows results based on models including expert data and bottom row shows results excluding expert data. The different kernels are depicted column wise. Distance distributions of movement distances in ICVI data of 2009, 2010, and 2011 are depicted as red, green and blue lines, respectively.

rsos.royalsocietypublishing.org R. Soc. open sci. 0000000

Figure 77. Distance distributions of generated interstate networks down sampled to 10% for beef and dairy combined. Top row shows results based on models including expert data and bottom row shows results excluding expert data. The different kernels are depicted column wise.

rsos.royalsocietypublishing.org R. Soc. open sci. 0000000

Figure 78. Distance distributions of generated interstate networks down sampled to 10% for beef and dairy combined. Top row shows results based on models including expert data and bottom row shows results excluding expert data. The different kernels are depicted column wise.

Figure 79. Distance distributions of generated interstate networks down sampled to 10% for beef and dairy combined. Top row shows results based on models including expert data and bottom row shows results excluding expert data. The different kernels are depicted column wise.

rsos.royalsocietypublishing.org R. Soc. open sci. 0000000

Figure 80. Distance distributions of generated interstate networks down sampled to 10% for beef and dairy combined. Top row shows results based on models including expert data and bottom row shows results excluding expert data. The different kernels are depicted column wise.

Figure 81. Distance distributions of generated interstate networks down sampled to 10% for beef and dairy combined. Top row shows results based on models including expert data and bottom row shows results excluding expert data. The different kernels are depicted column wise.

1
2
3
4
5
6
7
8
9
10
11
12
13
14
15
16
17
18
19
20
21
22
23
24
25
26
27
28
29
30
31
32
33
34
35
36
37
38
39
40
41
42
43
44
45
46
47
48
49
50
51
52
53
54
55
56
57
58
59
60

Figure 82. Distance distributions of generated interstate networks down sampled to 10% for beef and dairy combined. Top row shows results based on models including expert data and bottom row shows results excluding expert data. The different kernels are depicted column wise.

rsos.royalsocietypublishing.org R. Soc. open sci. 0000000

Figure 83. Distance distributions of generated interstate networks down sampled to 10% for beef and dairy combined. Top row shows results based on models including expert data and bottom row shows results excluding expert data. The different kernels are depicted column wise. Distance distributions of movement distances in ICVI data of 2009, 2010, and 2011 are depicted as red, green and blue lines, respectively.

1
2
3
4
5
6
7
8
9
10
11
12
13
14
15
16
17
18
19
20
21
22
23
24
25
26
27
28
29
30
31
32
33
34
35
36
37
38
39
40
41
42
43
44
45
46
47
48
49
50
51
52
53
54
55
56
57
58
59
60

rsos.royalsocietypublishing.org R. Soc. open sci. 0000000
.....

Figure 84. Distance distributions of generated interstate networks down sampled to 10% for beef and dairy combined. Top row shows results based on models including expert data and bottom row shows results excluding expert data. The different kernels are depicted column wise.

104

rsos.royalsocietypublishing.org R. Soc. open sci. 0000000

Figure 85. Distance distributions of generated interstate networks down sampled to 10% for beef and dairy combined. Top row shows results based on models including expert data and bottom row shows results excluding expert data. The different kernels are depicted column wise.

Figure 86. Distance distributions of generated interstate networks down sampled to 10% for beef and dairy combined. Top row shows results based on models including expert data and bottom row shows results excluding expert data. The different kernels are depicted column wise.

Figure 87. Distance distributions of generated interstate networks down sampled to 10% for beef and dairy combined. Top row shows results based on models including expert data and bottom row shows results excluding expert data. The different kernels are depicted column wise. Distance distributions of movement distances in ICVI data of 2009, 2010, and 2011 are depicted as red, green and blue lines, respectively.

Figure 88. Distance distributions of generated interstate networks down sampled to 10% for beef and dairy combined. Top row shows results based on models including expert data and bottom row shows results excluding expert data. The different kernels are depicted column wise.

rsos.royalsocietypublishing.org R. Soc. open sci. 0000000

Figure 89. Distance distributions of generated interstate networks down sampled to 10% for beef and dairy combined. Top row shows results based on models including expert data and bottom row shows results excluding expert data. The different kernels are depicted column wise. Distance distributions of movement distances in ICVI data of 2009, 2010, and 2011 are depicted as red, green and blue lines, respectively.

Figure 90. Distance distributions of generated interstate networks down sampled to 10% for beef and dairy combined. Top row shows results based on models including expert data and bottom row shows results excluding expert data. The different kernels are depicted column wise.

rsos.royalsocietypublishing.org R. Soc. open sci. 0000000

Figure 91. Distance distributions of generated interstate networks down sampled to 10% for beef and dairy combined. Top row shows results based on models including expert data and bottom row shows results excluding expert data. The different kernels are depicted column wise.

Figure 92. Distance distributions of generated interstate networks down sampled to 10% for beef and dairy combined. Top row shows results based on models including expert data and bottom row shows results excluding expert data. The different kernels are depicted column wise.

rsos.royalsocietypublishing.org R. Soc. open sci. 0000000

Figure 93. Distance distributions of generated interstate networks down sampled to 10% for beef and dairy combined. Top row shows results based on models including expert data and bottom row shows results excluding expert data. The different kernels are depicted column wise.

Figure 94. Distance distributions of generated interstate networks down sampled to 10% for beef and dairy combined. Top row shows results based on models including expert data and bottom row shows results excluding expert data. The different kernels are depicted column wise.

Figure 95. Distance distributions of generated interstate networks down sampled to 10% for beef and dairy combined. Top row shows results based on models including expert data and bottom row shows results excluding expert data. The different kernels are depicted column wise.

Figure 96. Distance distributions of generated interstate networks down sampled to 10% for beef and dairy combined. Top row shows results based on models including expert data and bottom row shows results excluding expert data. The different kernels are depicted column wise.

Figure 97. Distance distributions of generated interstate networks down sampled to 10% for beef and dairy combined. Top row shows results based on models including expert data and bottom row shows results excluding expert data. The different kernels are depicted column wise.

rsos.royalsocietypublishing.org R. Soc. open sci. 0000000

Figure 98. Distance distributions of generated interstate networks down sampled to 10% for beef and dairy combined. Top row shows results based on models including expert data and bottom row shows results excluding expert data. The different kernels are depicted column wise.

Figure 99. Distance distributions of generated interstate networks down sampled to 10% for beef and dairy combined. Top row shows results based on models including expert data and bottom row shows results excluding expert data. The different kernels are depicted column wise.

rsos.royalsocietypublishing.org R. Soc. open sci. 0000000

Figure 100. Distance distributions of generated interstate networks down sampled to 10% for beef and dairy combined. Top row shows results based on models including expert data and bottom row shows results excluding expert data. The different kernels are depicted column wise.

Figure 101. Distance distributions of generated interstate networks down sampled to 10% for beef and dairy combined. Top row shows results based on models including expert data and bottom row shows results excluding expert data. The different kernels are depicted column wise.

Figure 102. Distance distributions of generated interstate networks down sampled to 10% for beef and dairy combined. Top row shows results based on models including expert data and bottom row shows results excluding expert data. The different kernels are depicted column wise.

Figure 103. Distance distributions of generated interstate networks down sampled to 10% for beef and dairy combined. Top row shows results based on models including expert data and bottom row shows results excluding expert data. The different kernels are depicted column wise.

rsos.royalsocietypublishing.org R. Soc. open sci. 0000000

Figure 104. Distance distributions of generated interstate networks down sampled to 10% for beef and dairy combined. Top row shows results based on models including expert data and bottom row shows results excluding expert data. The different kernels are depicted column wise.

Figure 105. Distance distributions of generated interstate networks down sampled to 10% for beef and dairy combined. Top row shows results based on models including expert data and bottom row shows results excluding expert data. The different kernels are depicted column wise.

Figure 106. Distance distributions of generated interstate networks down sampled to 10% for beef and dairy combined. Top row shows results based on models including expert data and bottom row shows results excluding expert data. The different kernels are depicted column wise.

Figure 107. Distance distributions of generated interstate networks down sampled to 10% for beef and dairy combined. Top row shows results based on models including expert data and bottom row shows results excluding expert data. The different kernels are depicted column wise. Distance distributions of movement distances in ICVI data of 2009, 2010, and 2011 are depicted as red, green and blue lines, respectively.

Figure 108. Distance distributions of generated interstate networks down sampled to 10% for beef and dairy combined. Top row shows results based on models including expert data and bottom row shows results excluding expert data. The different kernels are depicted column wise.

rsos.royalsocietypublishing.org R. Soc. open sci. 0000000

Figure 109. Distance distributions of generated interstate networks down sampled to 10% for beef and dairy combined. Top row shows results based on models including expert data and bottom row shows results excluding expert data. The different kernels are depicted column wise.

Figure 110. Distance distributions of generated interstate networks down sampled to 10% for beef and dairy combined. Top row shows results based on models including expert data and bottom row shows results excluding expert data. The different kernels are depicted column wise.

Figure 111. Distance distributions of generated interstate networks down sampled to 10% for beef and dairy combined. Top row shows results based on models including expert data and bottom row shows results excluding expert data. The different kernels are depicted column wise.

rsos.royalsocietypublishing.org R. Soc. open sci. 0000000

Figure 112. Distance distributions of generated interstate networks down sampled to 10% for beef and dairy combined. Top row shows results based on models including expert data and bottom row shows results excluding expert data. The different kernels are depicted column wise.

Figure 113. Distance distributions of generated interstate networks down sampled to 10% for beef and dairy combined. Top row shows results based on models including expert data and bottom row shows results excluding expert data. The different kernels are depicted column wise.

Figure 114. Distance distributions of generated interstate networks down sampled to 10% for beef and dairy combined. Top row shows results based on models including expert data and bottom row shows results excluding expert data. The different kernels are depicted column wise.

Figure 115. Distance distributions of generated interstate networks down sampled to 10% for beef and dairy combined. Top row shows results based on models including expert data and bottom row shows results excluding expert data. The different kernels are depicted column wise.

rsos.royalsocietypublishing.org R. Soc. open sci. 0000000

Figure 116. Distance distributions of generated interstate networks down sampled to 10% for beef and dairy combined. Top row shows results based on models including expert data and bottom row shows results excluding expert data. The different kernels are depicted column wise. Distance distributions of movement distances in ICVI data of 2009, 2010, and 2011 are depicted as red, green and blue lines, respectively.

Figure 117. Distance distributions of generated interstate networks down sampled to 10% for beef and dairy combined. Top row shows results based on models including expert data and bottom row shows results excluding expert data. The different kernels are depicted column wise.

Figure 118. Distance distributions of generated interstate networks down sampled to 10% for beef and dairy combined. Top row shows results based on models including expert data and bottom row shows results excluding expert data. The different kernels are depicted column wise.

Figure 119. Distance distributions of generated interstate networks down sampled to 10% for beef and dairy combined. Top row shows results based on models including expert data and bottom row shows results excluding expert data. The different kernels are depicted column wise.

Appendix B

Reviewers' Comments to Author:

Reviewer: 1

Comments to the Author(s)

I found this manuscript very difficult to read. This is in part because the statistical model is quite complicated, but I think much more could be done to help the reader.

We appreciate this concern and have substantially restructured the method section in line with the suggestions offered by the reviewers.

At a trivial level, there are a few notational errors or inconsistencies that need to be fixed. With a model as complicated as this, the notation should be carefully designed to assist understanding; having the same symbol used for different things is a bad idea (for example, ω is used for two quite different things), and notational errors are confusing and frustrating.

We have thoroughly gone through the paper and hope no further notation errors are included. Specifically for the case of omega, it should be noted that this in fact was correctly implemented, and we believe this raised concern was likely due to a similarity between the lower-case omega and the italic w, which look quite similar. Still, to avoid such confusion for readers, we have changed the notation for omega to the variant pi (ϖ), which is more distinctly different from the italic w. Other alternatives we considered led to other issues in terms of clarity.

At a more structural level, my view is that the paper needs some fundamental restructuring to make it easier on the reader.

While having a table like Table 1 to define notation is a good idea, putting it up front rather than in an appendix is a bad idea in my view. Without knowing the model intimately, the reader can't possibly absorb everything in the table. By the time I had been through the table I was a bit dazed, knowing I had not absorbed all of it, and wondering how I was possibly going to follow what came next without holding the whole table in my head.

We have followed the suggestion to move the table to an appendix.

Here's a suggestion for an alternative structure, in which the central components of the model are presented to give a broad-brush picture of the model, followed by the details of each component later, by which time the reader has the big picture and the context:

1. Put Eqn (2.2) front and centre, at the start of the model description. Say what each of its components is but don't give the detail of $p_{\{\delta|\omega\}}$, just forward-reference the subsection that gives this detail (which should also contain details of the kernels, since these appear only as a component of $p_{\{\delta|\omega\}}$).

2. Replace lines 50-60 on page 9, and lines 1-16 on page 10 with the statement "We factorise Eqn (2.2) into the probability of total shipments $|T\{\tau\}_\omega|$ leaving county ω , and the probabilities of the observed incoming shipments at each destination county outside the state, conditional on this total shipment". Then give the maths of Eqn (2.4) - but try to do it in a simpler way! Then just say that the probability for all days and all origin counties is just the product over counties and days of this probability. I struggle to see the purpose of Eqn (2.5) beyond this simple statement, and as it stands (2.5) and all the text and math around it just muddies the waters (even if it is mathematically rigorous - if you really want it, put it in an appendix).

The above then specifies the model for the observed data, in a much simpler and much more understandable way than currently.

We have substantially restructured the methods in line with these suggestions and moved large chunks to appendices. As a slight downside to this restructuring, which in contrast to the former structure, starts at the higher level and builds down to the details, there are repeated instances where we have to refrain from exact definitions when concepts are first mentioned and instead reference appendices or subsequent sections. Yet, we agree with the reviewer that this new structure likely make the model easier to grasp.

3. Put all the detail of the hierarchical Bayes model (in particular, lines 20 to 43 on page 12) into an appendix.

We have followed this suggestion and have moved details of the hierarchical prior structure to the supplementary material.

4. Put as much of section J "Computation" as possible into an appendix.

We have followed this suggestion and have moved most of the computational details to the supplementary material.

Finally, the writing style is really poor throughout. Sentences are often too long, there are too many parenthetical clauses, and a mixture of mathematics and English that makes them very difficult to read. Use shorter, simpler sentences and separate the maths from the text. To give a random example, you say this :

"we modeled the total probability of observing $|T^{\{\tau\}}_{\{\omega\delta\}}|$ number of interstate shipments from county ω in E_S to all possible interstate destination counties δ in $C \setminus E_S$ (i.e. all contiguous counties not in S) on day τ , as a product of their individual probabilities."

This is shorter, simpler and clearer:

"we modeled the probability of the total observed interstate shipments on a day, from one county to any other outside its state, as a product of their individual probabilities." Then give the maths. Much easier to follow.

We have changed these specific suggestions and have aimed to generally incorporate more text along the mathematical notations. Yet, we cannot accurately and unambiguously present the model without the exact specifications, which requires mathematical notations. Hopefully, the tradeoff is perceived better balanced in our revised manuscript.

There are a handful of minor comments on notation and such, in the attached pdf file.

We have made changes according to these suggestions.

Reviewer: 2

Comments to the Author(s)

The combination of elicited expert opinion and quantitative data is an interesting and potentially very useful approach to reconstructing networks. I will not profess high levels of expertise in understanding how to interpret expert opinion data such as these, but the approach appears sensible. It is of course impossible to directly evaluate the computational aspects of the model but the approach to obtaining the posteriors is sensible and careful and would seem to be appropriate.

I have a few general comments. First of all, the authors largely seem to ignore the temporal aspects of the livestock network, in particular whether or not there is variation in the tempo particularly when comparing across states. The assumption of Poisson distributed frequencies may be poor, and this is likely to be a critical issue for the transmission of some disease, in particular where disease transmission is close to the threshold for persistence. It would be useful to be given some confidence that temporal variation (be it weekly, monthly or seasonal) will not impact the predicted results.

This comment made us conflicted. On the one hand, we agree with the reviewer; seasonality could play an important role in shipment patterns. On the other hand, thorough mapping of seasonality

patterns is well worth a study of its own and would venture outside of the scope of this study. We have therefore limited the studies of seasonality to the scope of the sensitivity analysis to investigate if seasonal (here quarters of the year) differences could explain observed patterns. Because there was little difference in the effect of kernel functional forms on the broad patterns and we encountered additional computational challenges in this pursuit, we limited these analyses to functional form 3. Note also that the available expert data only contained annual estimates, and we therefore excluded experts from these analyses. The analyses show that some differences are observed, but the broad patterns are consistent across seasons. To limit the amount of figures and text, we show some of these results in the main text and include additional information in the supplementary material. We also provide seasonal networks (1000 realization).

Examining the maps comparing the distribution (by state) of proportions of interstate inward and outward movements (Figures 3 and 4) there seems to be little difference amongst the 3 kernels, and it is difficult to discern how much is being added to the information by the processes generated the 3 fitted outcomes, as there is no comparison to a null model. This comparison is essential to determining the value of the fit. I would suggest taking a very simple, easy to calculate model as a point of comparison to determine how much value is being added by the authors' methods. **We agree with the reviewer that comparison to a null model is useful, yet it not always straightforward to define a relevant null model. We did however include a map of the proportion of premises (per commodity) located in each state, which would be mirrored in the results if all premises were equally likely to ship to each other, independent of location. To reduce the volume of the paper, we however included this as supplementary information.**

Also the expert opinion information seems to add little to the model. For example in terms of the model selection in Table 2, or predicting the proportion of interstate movements in Fig. 6, or in Fig 7. The authors are transparent about this and the reasons why. However, the paper is written as if this inclusion is a major aim of the study, and yet there is little here to validate the approach they have developed. It would be helpful to have some better evidence that the methods they have proposed are really worth the effort (which appears to be considerable). At the very least some insight as to what conditions would one expect these opinions to matter would be very useful to have. **We agree with the reviewer, and we also expected the inclusion of experts to have more pronounced effect on the results. We have tried to emphasize in our revised manuscript that the reason for this is because, when the experts provide disparate estimates, they do not substantially inform the model. However, rather than disregarding experts from the analysis, we believe it is important to publish examples of when experts do not improve analyses, thereby avoiding a publication bias of success stories.**

In figure 9, there are some strong bi-modalities in the CVI data for some states, which are not present in the USAMM predictions - this could do with some explanation.

The exact reason for this is unclear, but as we now bring up in the discussion, it is likely due to infrastructure and/or premises characteristics present in some counties that promotes high number of shipments between specific pairs. We suggest that future studies should aim to clarify what these drivers are, which could improve predictions at the county scale.

Appendix C

To the RSOS Editorial Team,

Please consider for publication our revised manuscript “Assessing intrastate shipments from interstate data and expert opinion.”

Reviewer comments to Author:

Reviewer: 2

Comments to the Author(s)

The authors have largely addressed my comments and those of the other reviewer. I would note that they have misunderstood my query regarding temporality of movements. While it is true that seasonality may play a role, and the additional analysis presented is useful, I was referring to the role that the Poisson distribution may play in allowing for the transmission of short infectious period diseases. Two papers that examine this question are Nickbakhsh et al (doi: 10.1016/j.epidem.2013.03.001), and Colman et al. (doi: 10.1186/s12879-018-3117-6). For slowly transmitting diseases this won't matter, but a Poisson distribution assumption is likely to make a rapidly transmitting disease spread more easily - therefore influencing the epidemic thresholds. I don't think its necessary that the authors do more work to address this in this manuscript - they have done plenty for a single publication already. It is worth noting though, that the assumption they've made (and given that livestock movements are a commercial enterprise, they almost certainly have much more regularly scheduled tempos) could have an influence on epidemic thresholds.

We have followed the suggestion of the reviewer and included a sentence about the potential importance of deviations from a Poisson process in our discussion (page 21, line 25).

A minor point (which I did not pick up on before) is that the authors state on line 55 onwards of page 23 that "Further, kernel one estimates of scale parameter d included values below $10E-7$ km (Figure 2). These are exceptionally short distances considering the large spatial scales of the U.S." This is true - and indeed is very short on any national scale. However, in practice, one would imagine that anyone using the model to assign movements to farms would simply devise some method to handle this - e.g. assign all probability below a certain threshold to either a non-movement, or to a local movement to a near neighbour. The important question is whether or not the probability of moving these very short distances (once the issue has been handled by some such method) seriously impacts the predictive power of the model in terms of disease transmission - if this is the case then the authors are correct that it detracts from the model in a practical sense - but otherwise it just necessitates an approximation.

This appears to be a slight misunderstanding. Our model does not predict shipment distances independent of recipient locations and then allocate them to nearby premises. Rather, the distance dependent kernels describe the relative probability of destinations, summarized over all possible destinations (see page 5, line 36). As this appears to not be an objection on the reviewer's behalf, we have not made any changes following this comment.